# ONLINE LEARNING AND EQUILIBRIUM COMPUTATION WITH RANKING FEEDBACK

**Mingyang Liu[1], Yongshan Chen[2,3], Zhiyuan Fan[1], Gabriele Farina[1], Asuman E. Ozdaglar[1], Kaiqing Zhang[2]**
[1] Massachusetts Institute of Technology
[2] University of Maryland, College Park
[3] Northeastern University

## ABSTRACT

Online learning in arbitrary, and possibly adversarial, environments has been extensively studied in sequential decision-making, and it is closely connected to equilibrium computation in game theory. Most existing online learning algorithms rely on *numeric* utility feedback from the environment, which may be unavailable in human-in-the-loop applications and/or may be restricted by privacy concerns. In this paper, we study an online learning model in which the learner only observes a *ranking* over a set of proposed actions at each timestep. We consider two ranking mechanisms: rankings induced by the *instantaneous* utility at the current timestep, and rankings induced by the *time-average* utility up to the current timestep, under both *full-information* and *bandit* feedback settings. Using the standard external-regret metric, we show that sublinear regret is impossible with instantaneous-utility ranking feedback in general. Moreover, when the ranking model is relatively deterministic, *i.e.*, under the Plackett-Luce model with a temperature that is sufficiently small, sublinear regret is also impossible with time-average utility ranking feedback. We then develop new algorithms that achieve sublinear regret under the additional assumption that the utility sequence has sublinear total variation. Notably, for full-information time-average utility ranking feedback, this additional assumption can be removed. As a consequence, when all players in a normal-form game follow our algorithms, repeated play yields an approximate coarse correlated equilibrium. We also demonstrate the effectiveness of our algorithms in an online large-language-model routing task.

## 1 INTRODUCTION

Online learning has been extensively studied as a model for sequential decision-making in arbitrary and possibly adversarial environments (Shalev-Shwartz et al., 2012; Hazan et al., 2016). At each round, the agent commits to a strategy, takes an action, and then receives *feedback* from the environment, often in *numeric* form, such as the utility vector (in the *full-information* feedback setting) or the realized utility value (in the *bandit* feedback setting). Numerous algorithms achieve no-regret guarantees, *i.e.*, they ensure that the agent's *external regret* grows sublinearly with time (Shalev-Shwartz et al., 2012; Hazan et al., 2016). Moreover, online learning is closely connected to equilibrium computation in game theory: when all players employ no-regret learning in the repeated play of a normal-form game (NFG), their time-average play approaches a coarse correlated equilibrium (CCE) (Cesa-Bianchi & Lugosi, 2006).

However, numeric utility feedback may not always be available in real-world applications. For instance, when the feedback is elicited from an environment with humans in the loop, it is typically much easier for them to *compare* or *rank* candidate actions than to provide calibrated numerical *scores*. This phenomenon has been widely recognized and is exemplified by the success of reinforcement learning from human feedback (RLHF) in fine-tuning language models (Ouyang et al., 2022). Moreover, even when well-defined numeric utilities exist, they may be inaccessible to the learning agent due to privacy or security constraints. As a concrete example, consider an online platform (see Figure 1 (a)) that recommends commodities to a stream of customers over time, where

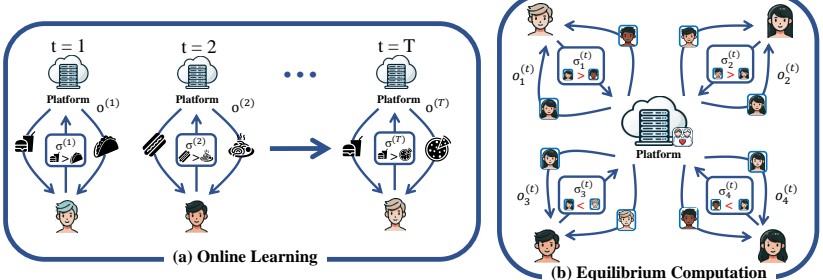

Figure 1: Two examples of online learning and equilibrium computation with ranking feedback. In (a), an online platform recommends food options to a customer at each timestep and receives a ranking over the proposed items, which it uses to improve recommendation quality. In (b), an online dating app recommends potential matches; users rank the suggested candidates, and the platform leverages these rankings to learn matching equilibria over time.

customers arriving at different timesteps may have different preferences. While the platform aims to improve its recommendations, customers may be unable or unwilling to reveal their true valuations. Depending on whether customers are *one-shot* (arrive, rank, and leave forever) or *long-lived* with memory, the ranking feedback may be induced either by the *instantaneous* utility at each timestep or by the *time-average* utility aggregated over historical utility vectors. The platform thus seeks to minimize the *regret* of its recommendations under ranking-only feedback. Yet, the fundamental limits and algorithmic solutions for this online learning setting remain elusive.

Ranking feedback becomes particularly relevant in game-theoretic settings, where multiple humans repeatedly interact, and the goal is to compute an equilibrium of the underlying game. For example, consider an online dating platform that recommends potential matches (see Figure 1 (b)). In each round, each user may provide only a ranking over the recommended candidates, and the platform aims to compute an equilibrium outcome, *i.e.*, a matching between users, that (approximately) respects everyone's preferences. Related scenarios arise in other matching platforms, *e.g.*, ride-sharing services that match drivers and passengers based on their preferences, such as drivers' preferences over trip lengths and riders' preferences over driving styles (*e.g.*, punctuality or cautiousness). Although these applications may seem reminiscent of the classical stable matching model (Gale & Shapley, 1962), our setting is fundamentally different. See Appendix A for a detailed comparison.

In this paper, we systematically study online learning and equilibrium computation with ranking feedback in a nonstochastic environment, where the loss vectors may be generated arbitrarily, and potentially even adversarially. This setting can also be viewed as a generalization of the stochastic bandits with ranking feedback studied recently in Maran et al. (2024). See Appendix A for a detailed comparison. Our goal is to understand when regret minimization in this setting is possible, and to develop new algorithms that provide both regret-minimization and equilibrium-approximation guarantees. We summarize our contributions as follows.

**Contributions.** We consider two types of ranking feedback, categorized by how the rankings are generated: one based on the *instantaneous utility* at each timestep (**InstUtil Rank**), and one based on the *time-average utility* up to the current timestep (**AvgUtil Rank**). We establish the following results: (i) sublinear regret is impossible under **InstUtil Rank** feedback. Moreover, under **AvgUtil Rank** feedback, sublinear regret remains unattainable (up to logarithmic factors) when the ranking model is overly *deterministic* (*i.e.*, when the temperature parameter $\tau > 0$ in (PL) is small); (ii) we propose new algorithms that achieve sublinear regret under an additional assumption that the utility vectors have sublinear variation; (iii) we show that this variation assumption can be removed under full-information **AvgUtil Rank** feedback when $\tau$ is a constant; and (iv) when all players run our no-regret learning algorithms in the repeated play of a normal-form game, the induced play yields an approximate coarse correlated equilibrium. Our contributions are summarized in Table 1. We present an application to large-language-model routing in Section 8, and provide experimental validation of our algorithms in Appendix C.

## 2   PRELIMINARIES

The basic notation and a brief introduction to normal-form games are deferred to Appendix B.

| Lower Bound | Full-Information | Bandit |
|---|---|---|
| **InstUtil Rank** | $\Omega(T)$ for $\tau \leq \mathcal{O}(1)$ | |
| **AvgUtil Rank** | $\widetilde{\Omega}(T)$ for $\tau \leq \mathcal{O}\left(\frac{1}{T \log T}\right)$ | $\Omega(T)$ for $\tau \leq \mathcal{O}\left(\frac{1}{\log T}\right)$ |

| Upper Bound ($\tau = \mathcal{O}(1)$, Sublinear Regret) | Full-Information | Bandit |
|---|---|---|
| **InstUtil Rank** | Assumption 4.2 | |
| **AvgUtil Rank** | ✓ | Assumption 4.2 ($q < \frac{1}{3}$) |

Table 1: Summary of our contributions, including the *negative results* (top) and the *positive results* (bottom). The bottom table lists the minimal assumptions required to achieve sublinear regret in each setting (✓ indicates that no additional assumptions are needed). Here, $\tau > 0$ denotes the temperature parameter of the ranking model in (PL).

## 2.1 ONLINE LEARNING

We study online learning in a non-stochastic and potentially adversarial environment, where an agent interacts with the environment over multiple timesteps by selecting an action and receiving feedback at each timestep. The agent's action set is finite and denoted as $\mathcal{A} := \left\{a^1, a^2, \ldots, a^{|\mathcal{A}|}\right\}$ with $|\mathcal{A}| > 1$. At each timestep $t$, the agent commits to a mixed strategy $\pi^{(t)} \in \Delta^{\mathcal{A}}$. In the classical online learning setting (with numeric feedback), with *full-information* feedback, the agent observes a utility vector $\boldsymbol{u}^{(t)} \in [-1, 1]^{\mathcal{A}}$; with *bandit* feedback, the agent observes only the realized utility $u^{(t)}(a^{(t)})$ for the sampled action $a^{(t)} \sim \pi^{(t)}$.

The agent's goal is to minimize her (external) *regret*, defined as the difference between her cumulative utility and that of the best fixed action in hindsight. Formally, for any integer $T > 0$, the regret is defined as

$$R^{(T),\text{external}} := \max_{\widehat{\pi} \in \Delta^{\mathcal{A}}} \sum_{t=1}^{T} \left\langle \boldsymbol{u}^{(t)}, \widehat{\pi} - \pi^{(t)} \right\rangle. \tag{2.1}$$

Since our goal is to minimize regret, and regret is unchanged if the utility vector $\boldsymbol{u}^{(t)}$ is shifted by an additive constant at each timestep $t$, we may assume without loss of generality that $u^{(t)}(a^{|\mathcal{A}|}) = 0$, *i.e.*, the last action always receives zero utility for any $t \in [T]$.

## 2.2 ONLINE LEARNING ALGORITHMS WITH NUMERIC FEEDBACK

Our main results later will be *modular*, in the sense that *any* standard online learning algorithm with (full-information) numeric feedback, including projected gradient descent (PGD), multiplicative-weights update (MWU), and Follow-The-Regularized-Leader (FTRL) (Hazan et al., 2016), can be used as a deterministic *black-box* oracle in the algorithms we develop.

As a preliminary, we formally define such deterministic oracles here. Let $\text{Alg} \colon \bigcup_{t=0}^{\infty} \left(\mathbb{R}^{\mathcal{A}}\right)^t \to \Delta^{\mathcal{A}}$ denote an online learning algorithm, which can be viewed as a mapping from a sequence of utility vectors to a distribution over the action set $\mathcal{A}$. Thus, given utility vectors $\left(\boldsymbol{u}^{(s)}\right)_{s=1}^{t}$ from timesteps 1 through $t$, the algorithm outputs the next strategy $\pi^{(t+1)} = \text{Alg}\left(\left(\boldsymbol{u}^{(s)}\right)_{s=1}^{t}\right)$ to be played at timestep $t + 1$. Note that this formulation implicitly assumes that $\text{Alg}$ is deterministic. Finally, we denote the (external) regret incurred by $\text{Alg}$ as

$$R^{(T),\text{external}}\left(\text{Alg}, \left(\boldsymbol{u}^{(t)}\right)_{t=1}^{T}\right) := \max_{\widehat{\pi} \in \Delta^{\mathcal{A}}} \sum_{t=1}^{T} \left\langle \boldsymbol{u}^{(t)}, \widehat{\pi} - \text{Alg}\left(\left(\boldsymbol{u}^{(s)}\right)_{s=1}^{t-1}\right) \right\rangle, \tag{2.2}$$

which can be made sublinear in $T$ for any sequence of utility vectors $\left(\boldsymbol{u}^{(t)}\right)_{t=1}^{T}$ (Hazan et al., 2016).

## 3 ONLINE LEARNING WITH RANKING FEEDBACK

In online learning with ranking feedback, at each timestep $t$, the agent does not have direct access to $\boldsymbol{u}^{(t)}$, nor the realized utility (the utility of the realized action at timestep $t$). Instead, at timestep $t$, she can propose a multiset (which may include repeated elements) of actions $o^{(t)}$, and receive

a permutation $\sigma^{(t)} \in \Sigma \left( o^{(t)} \right)$ ($\Sigma(\cdot)$ returns the set of all permutations) from the environment, representing a *ranking* of those actions in $o^{(t)}$. In the full-information setting, $o^{(t)} = \mathcal{A}$, *i.e.*, the whole action set is proposed. In the bandit setting, $\left| o^{(t)} \right| = K \leq |\mathcal{A}|$. Suppose the agent's strategy at timestep $t$ is $\pi^{(t)} \in \Delta^{\mathcal{A}}$, then we assume that in this bandit setting, the actions in $o^{(t)}$ are proposed by sampling from $\pi^{(t)}$ independently (with replacement). This way, the empirical average of the utilities will be an *unbiased* estimator of the expected utility, *i.e.*, $\mathbb{E} \left[ \frac{\sum_{a \in o^{(t)}} u^{(t)}(a)}{K} \right] = \left\langle \boldsymbol{u}^{(t)}, \pi^{(t)} \right\rangle$.

We will adopt this proposal mechanism throughout the paper, which was also used in the literature of adversarial dueling bandits (Saha et al., 2021).

Let $\sigma^{(t)}(k) \in \mathcal{A}$ be the $k^{th}$ element of the permutation for any $k \in [K]$. Then, for any $k_1 < k_2 \in [K]$, action $\sigma^{(t)}(k_1)$ is preferred over action $\sigma^{(t)}(k_2)$. For notational simplicity, we define $a^i \overset{\sigma}{>} a^j$ if action $a^i$ appears ahead of $a^j$ ($a^i$ is preferred over $a^j$) in a permutation $\sigma$.

For the ranking model, we consider the standard Plackett-Luce (PL) model (Luce, 1959; Plackett, 1975), where at each timestep $t$, conditioned on the proposed action set $o^{(t)}$, the ranking $\sigma^{(t)}$ is sampled according to

$$\mathbb{P} \left( \sigma^{(t)} \, | \, o^{(t)} \right) = \prod_{k_1=1}^{K} \frac{\exp \left( \frac{1}{\tau} r^{(t)} \left( \sigma^{(t)} (k_1) \right) \right)}{\sum_{k_2=k_1}^{K} \exp \left( \frac{1}{\tau} r^{(t)} (\sigma^{(t)} (k_2)) \right)}, \tag{PL}$$

where $\boldsymbol{r}^{(t)} \in \mathbb{R}^{\mathcal{A}}$ is some utility vector based on which the ranking is determined, $\tau > 0$ is the *temperature* parameter that determines how uncertain the ranking model is: when $\tau \to 0^+$, the model is absolutely certain, and the action with a larger utility in $\boldsymbol{r}^{(t)}$ will always be ranked in front of the actions with a smaller utility in the permutation (ties are broken in favor of the smaller index). The utility vector $\boldsymbol{r}^{(t)}$ depends on the problem setting, which we will introduce next.

We consider two types of ranking feedback throughout the paper, based on the choice of $\boldsymbol{r}^{(t)}$ in (PL): (i) ranking by the *instantaneous* utility (**InstUtil Rank**); (ii) ranking by the *time-average* utility (**AvgUtil Rank**). These two feedback types may be motivated by different applications (cf. Section 1), and have been studied in Yue et al. (2012); Saha & Gaillard (2022); Maran et al. (2024). Both feedback types can also be further separately defined for the *full-information* and *bandit* settings, respectively, as follows.

**InstUtil Rank: Ranking with Instantaneous Utility.**

The first type of ranking feedback we consider is based on the *instantaneous* utility function, *i.e.*, $\boldsymbol{r}^{(t)} = \boldsymbol{u}^{(t)}$ in (PL). This type is relevant when the feedback provider is oblivious or one-shot. For example, a stream of customers arrive in an online fashion, each of whom arrives, ranks, and then leaves, see *e.g.*, Mansour et al. (2015). When the environment is stationary and stochastic, the classical dueling-bandits model likewise uses instantaneous utilities for comparison and ranking (Yue et al., 2012; Du et al., 2020).

**Full-information setting.** *All* actions are proposed and evaluated at each timestep $t$, *i.e.*, $o^{(t)} = \mathcal{A}$. Hence, the agent's performance can be *evaluated* by $\left\langle \boldsymbol{u}^{(t)}, \pi^{(t)} \right\rangle$. Note that this does not imply the agent has access to the full vector $\boldsymbol{u}^{(t)}$, as that would defeat the purpose of our ranking-feedback setting. Accordingly, we evaluate performance using the standard external regret $R^{(T),\text{external}}$ defined in (2.1).

**Bandit setting.** Only the *proposed actions* at each timestep $t$ can be evaluated and ranked, with the associated elements in the vector $\boldsymbol{u}^{(t)}$. In particular, the proposed actions are *evaluated* by the average utility of $\frac{1}{K} \sum_{a \in o^{(t)}} u^{(t)}(a)$, leading to the following performance metric of regret:

$$R^{(T)} := \max_{\widehat{\pi} \in \Delta^{\mathcal{A}}} \sum_{t=1}^{T} \left( \left\langle \boldsymbol{u}^{(t)}, \widehat{\pi} \right\rangle - \frac{1}{K} \sum_{a \in o^{(t)}} u^{(t)}(a) \right). \tag{3.1}$$

Note that such a definition is an external regret, which differs from the regret studied in (multi-)dueling-bandits (Yue et al., 2012; Du et al., 2020; Saha et al., 2021; Saha & Gaillard, 2022). Details can be found in Appendix A.

**AvgUtil Rank: Ranking with Time-average Utility.**

The second type of ranking-feedback is based on the *time-average* utility, which differs for the full-information and bandit settings, as detailed below. This type is relevant when the feedback provider has *memory* and can use the history of utilities for ranking. For example, the customers are long-lived in the platform, see *e.g.*, Küçükgül et al. (2022) and Baldwin (2009). Notably, under bandit feedback, when $\tau \to 0^+$ and the environment is stationary and stochastic, such a model aligns with the one studied in the recent work of Maran et al. (2024).

**Full-information setting.** The time-average utility vector of $\boldsymbol{u}_{\text{avg}}^{(t)} := \frac{1}{t} \sum_{s=1}^{t} \boldsymbol{u}^{(s)}$ will be used as the $\boldsymbol{r}^{(t)}$ in (PL), and the same (external-)regret $R^{(T),\text{external}}$ from (2.1) will be used as the metric.
**Bandit setting.** Only the proposed actions will be given to the environment to evaluate. For instance, the platform (learning agent) may recommend $K$ restaurants among all possibilities to the user (environment) to try out, so that the user will only know her evaluations of those $K$ restaurants. As a result, the average utility is now defined as the *empirical mean* of the utility vectors over time. Formally, for each timestep $t > 0$ and action $a \in \mathcal{A}$, we define

$$u_{\text{empirical}}^{(t)}(a) := \frac{\sum_{s=1}^{t} u^{(s)}(a) \sum_{a' \in o^{(s)}} \mathbb{1}\left(a = a'\right)}{\sum_{s=1}^{t} \sum_{a' \in o^{(s)}} \mathbb{1}\left(a = a'\right)}, \tag{3.2}$$

and $u_{\text{empirical}}^{(0)}(a) = 0$. This $\boldsymbol{u}_{\text{empirical}}^{(t)}$ will then be used as the $\boldsymbol{r}^{(t)}$ in (PL) for ranking. The regret metric will still be the one in (3.1).
Due to space constraints, we defer the background and formalism of equilibrium computation (with ranking feedback) in the game-theoretic setting to Appendix B.

## 4 HARDNESS RESULTS

In this section, we present hard instances to show that online learning in non-stochastic and potentially adversarial environments can be hard in general, under both **InstUtil Rank** and **AvgUtil Rank**, even when there are only two actions.

Theorem 4.1 in the following shows that for any temperature $\tau$ in (PL) no larger than a *constant*, there exists a sequence of utility vectors such that the expected regret is linear under **InstUtil Rank**, for both full-information and bandit feedback settings.

**Theorem 4.1.** *Consider **InstUtil Rank**. For any $T > 0$, temperature $0 < \tau \leq 0.1$, and online learning algorithm, there exists a sequence of utilities $\left(\boldsymbol{u}^{(t)}\right)_{t=1}^{T}$ such that $\min\left\{\mathbb{E}\left[R^{(T),\text{external}}\right], \mathbb{E}\left[R^{(T)}\right]\right\} \geq \Omega\left(T\right)$ in both full-information and bandit feedback settings. The expectation is taken over the randomness of the algorithm and the ranking.*

To prove Theorem 4.1, we need to construct two sequences of utility vectors, which yield the same ranking under **InstUtil Rank** in expectation. However, being no-regret in one of them will result in linear regret in the other. The detailed proof can be found in Appendix D.

The key challenge in achieving no-regret in the hard instance above is that the utility vectors $\left(\boldsymbol{u}^{(t)}\right)_{t=1}^{T}$ change arbitrarily fast, *i.e.*, the accumulated variation grows *linearly* in time. Hence, to obtain positive results, we may need to restrict how fast they change over time, as quantitatively characterized by the following assumption.

**Assumption 4.2** (Sublinear variation of utility vectors). *The utility vectors $\left(\boldsymbol{u}^{(t)}\right)_{t=1}^{T}$ have a sublinear variation over time,* i.e.*, for some $q < 1$,*

$$P^{(T)} := \sum_{t=2}^{T} \left\|\boldsymbol{u}^{(t)} - \boldsymbol{u}^{(t-1)}\right\| \leq \mathcal{O}(T^q). \tag{4.1}$$

Our result stated in Section 5 next will show that with Assumption 4.2, we can achieve sublinear regret, and thus close the gap. Moreover, note that in a game where the opponents all run common no-regret learning algorithms such as Follow-The-Regularized-Leader (cf. Definition L.2), Assumption 4.2 will be satisfied (cf. Lemma L.3).

Next, we show in Theorem 4.3 that when **AvgUtil Rank** is used, and $\tau$ is small enough, the minimal regret is still at least linear in $T$ (up to logarithmic terms).

**Theorem 4.3.** *Consider* **AvgUtil Rank** *with full-information feedback. For any $T > 0$, temperature $0 < \tau \leq \mathcal{O}\left(\frac{1}{T \log T}\right)$, and online learning algorithm, there exists $T' \geq T$ and a sequence of utilities $\left(\boldsymbol{u}^{(t)}\right)_{t=1}^{T'}$ such that $\mathbb{E}\left[R^{(T'),\text{external}}\right] \geq \widetilde{\Omega}\left(T'\right)$. The expectation is taken over the randomness of the algorithm and the ranking.*

To prove Theorem 4.3, we construct $\log T$ sequences of utility vectors that induce identical ranking feedback when $\tau$ is small. We then show that at least one of these sequences incurs average regret $\widetilde{\Omega}(1)$. Given Theorem 4.3, it is impossible to achieve $\widetilde{o}(T)$ with **AvgUtil Rank** when $\tau$ is very small. However, in Section 6, we will mitigate the gap by showing that when $\tau$ is a *constant* (*i.e.*, $\mathcal{O}(1)$), we can achieve sublinear regret with **AvgUtil Rank**, even without Assumption 4.2.

Due to the different instantiations of $\boldsymbol{r}^{(t)}$ in the full-information and the bandit feedback settings under **AvgUtil Rank**, we have a separate hardness result for the latter, stronger than Theorem 4.3, as it allows a larger $\tau$ and avoids logarithmic terms. The result can be viewed as strengthening the hardness result for the adversarial bandit setting in Maran et al. (2024), which corresponds to the case in our model with $\tau \to 0^+$.

**Theorem 4.4.** *Consider* **AvgUtil Rank** *with bandit feedback. For any $T > 0$, temperature $0 < \tau \leq \mathcal{O}\left(\frac{1}{\log T}\right)$, and online learning algorithm, there exists a sequence of utilities $\left(\boldsymbol{u}^{(t)}\right)_{t=1}^{4T}$ such that $\mathbb{E}\left[R^{(4T)}\right] \geq \Omega\left(T\right)$. The expectation is taken over the randomness of the algorithm and the ranking.*

To prove Theorem 4.4, we need to construct two utility sequences such that achieving sublinear regret in the first utility sequence will lead to insufficient exploration for the second sequence. As a result, when $\tau$ is small, those two sequences cannot be distinguished, and a linear regret must be incurred in one of them. Details of the proof can be found in Appendix D.

## 5 ONLINE LEARNING WITH **InstUtil Rank** FEEDBACK

We start by introducing a new utility estimation oracle to be used in our later algorithms.

### 5.1 UTILITY ESTIMATION

A natural approach to learning from ranking feedback is to use the observed permutations to *estimate* the underlying numeric utility vectors. At each timestep $t$, we form an estimate of the current utility vector $\boldsymbol{u}^{(t)}$ using a sliding window of the most recent $m$ rounds of observations: when $t \geq m$, we use the past $m$ permutations $\left\{\sigma^{(s)}\right\}_{s=t-m+1}^{t}$ to construct an estimate $\widetilde{\boldsymbol{u}}^{(t)}$. Due to the non-convexity of (PL), the key step is to decompose a length-$K$ ranking into pairwise comparisons. This reduction allows us to exploit the logistic structure of pairwise comparison probabilities and, via the monotonicity (and invertibility) of the logistic map, translate the estimation error of ranking probabilities back into an error bound on estimating the utilities. The full procedure is given in Algorithm 1, and we state its guarantee next.

**Theorem 5.1.** *Consider* **InstUtil Rank** *and Algorithm 1. Suppose each action is proposed independently with probability at least $p > 0$ at each timestep $t \in [T]$ and let $\widetilde{\boldsymbol{u}}^{(t)} = \text{Estimate}\left(\left\{\sigma^{(s)}\right\}_{s=t-m'+1}^{t}\right)$. Then, for any $\delta \in (0,1)$ and $t \geq m'$, when $m' p^4 \geq 2 \log\left(\frac{2}{\delta}\right)$, with probability at least $1 - \delta$, the estimate $\widetilde{\boldsymbol{u}}^{(t)}$ satisfies,*

$$\left\|\widetilde{\boldsymbol{u}}^{(t)} - \boldsymbol{u}^{(t)}\right\|_{\infty} \leq \frac{\tau \left(e^{\frac{1}{\tau}} + 1\right)^2}{p} \sqrt{\frac{\log\left(\frac{4|\mathcal{A}|}{\delta}\right)}{m'}} + \sum_{s=t-m'+1}^{t-1} \left\|\boldsymbol{u}^{(s+1)} - \boldsymbol{u}^{(s)}\right\|_{\infty}.$$

Treating $\delta$, $p$, and $\tau$ as constants, the accumulated estimation error $\sum_{t=1}^{T} \left\|\widetilde{\boldsymbol{u}}^{(t)} - \boldsymbol{u}^{(t)}\right\|_{\infty}$ will be bounded by $\mathcal{O}\left(\frac{T}{\sqrt{m'}} + m' P^{(T)}\right)$, which implies that a sublinear accumulated estimation error can be achievable when $P^{(T)}$ is sublinear (Assumption 4.2). Moreover, Theorem 5.1 shows that the upper bound diverges as $\tau \to 0^+$. This aligns with intuition: when $\tau \to 0^+$, the highest-utility action

is ranked first almost deterministically, and thus the feedback carries essentially no information about the utility gaps between actions. At the other extreme, when $\tau \to +\infty$, the bound also diverges because the ranking is nearly sampled uniformly and hence largely independent of the underlying utility vector.

The full proof of Theorem 5.1 is deferred to Appendix E. Next, we show how to achieve sublinear regret in both full-information and bandit settings with **InstUtil Rank**, based on such an estimator.

## 5.2 SUBLINEAR REGRET WITH **InstUtil Rank**

This section shows that for any online learning algorithm that can achieve sublinear external regret with numeric utility feedback, we can construct an online learning algorithm with **InstUtil Rank** feedback based on it, in a black-box way.

With full-information feedback, the learning agent proposes the full action set $\mathcal{A}$ at each timestep. In this case, we can obtain $\widetilde{\boldsymbol{u}}^{(t)}$, an estimate of $\boldsymbol{u}^{(t)}$, by Algorithm 1, and obtain guarantees using Theorem 5.1 with $p = 1$, since all actions are proposed at each timestep.

With bandit feedback, the utility of the learning agent at timestep $t$ is $\frac{1}{K} \sum_{k=1}^{K} u^{(t)} \left( \sigma^{(t)}(k) \right)$, *i.e.*, the average utility of the proposed actions. To achieve sublinear $R^{(T)}$ (as defined in (3.1)), each proposed action will be sampled from $\pi^{(t)}$ independently *with replacement*. In other words, an action may be proposed multiple times at a single timestep. To ensure sufficient exploration, namely, every action is proposed with positive probability, we impose a uniform lower bound $\pi^{(t)}(a) \geq \frac{\gamma}{|\mathcal{A}|}$ for some $\gamma > 0$ and all $a \in \mathcal{A}$. We enforce this by updating $\pi^{(t+1)} = (1-\gamma)\mathrm{Alg}\left( \left( \widetilde{\boldsymbol{u}}^{(s)} \right)_{s=1}^{t} \right) + \gamma \frac{\mathbf{1}(\mathcal{A})}{|\mathcal{A}|}$, *i.e.*, a convex combination of the strategy generated by the no-regret learning algorithm Alg and a uniform probability distribution over $\mathcal{A}$. A diagram of the algorithm can be found in Figure 10, and the details are tabulated in Algorithm 2. Then, we have the following theorem.

**Theorem 5.2.** *Consider* **InstUtil Rank** *with constant* $\tau > 0$. *By running Algorithm 2, for any* $\delta \in (0,1)$, $T > 0$, *and any full-information no-regret learning algorithm with numeric utility feedback,* Alg, *by choosing the window size* $m$ *and* $\gamma$ *properly, we have that with probability at least* $1 - \delta$, *the following hold:*

$$R^{(T),\text{external}} \leq R^{(T),\text{external}} \left( \mathrm{Alg}, \left( \widetilde{\boldsymbol{u}}^{(t)} \right)_{t=1}^{T} \right) + \mathcal{O}\left( \left( P^{(T)} \right)^{\frac{1}{3}} T^{\frac{2}{3}} \left( \log\left( \frac{T}{\delta} \right) \right)^{\frac{1}{3}} \right) \qquad \text{(Full-Info)}$$

$$R^{(T)} \leq R^{(T),\text{external}} \left( \mathrm{Alg}, \left( \widetilde{\boldsymbol{u}}^{(t)} \right)_{t=1}^{T} \right) + \mathcal{O}\left( \left( P^{(T)} \right)^{\frac{1}{5}} T^{\frac{4}{5}} \log\left( \frac{T}{\delta} \right) \right). \qquad \text{(Bandit)}$$

The proof can be found in Appendices G and H. Theorem 5.2 implies that when $P^{(T)}$, the variation of utility vectors, is sublinear (cf. Assumption 4.2), the regret of Algorithm 2 will be sublinear.

# 6 ONLINE LEARNING WITH **AvgUtil Rank** FEEDBACK

## 6.1 UTILITY ESTIMATION

Since $\sigma^{(t)}$ is generated based on $\boldsymbol{u}_{\text{avg}}^{(t)} := \frac{1}{t} \sum_{s=1}^{t} \boldsymbol{u}^{(s)}$, we will estimate $\boldsymbol{u}_{\text{avg}}^{(t)}$ instead. We will still apply Algorithm 1, which generates $\widetilde{\boldsymbol{u}}_{\text{avg}}^{(t)}$, an estimate of $\boldsymbol{u}_{\text{avg}}^{(t)}$, when the permutation is sampled under **AvgUtil Rank** feedback. Moreover, notice that

$$\left\| \boldsymbol{u}_{\text{avg}}^{(t)} - \boldsymbol{u}_{\text{avg}}^{(t-1)} \right\|_{\infty} = \left\| \frac{\boldsymbol{u}^{(t)} + (t-1)\boldsymbol{u}_{\text{avg}}^{(t-1)}}{t} - \boldsymbol{u}_{\text{avg}}^{(t-1)} \right\|_{\infty} \leq \frac{1}{t} \left( \left\| \boldsymbol{u}^{(t)} \right\|_{\infty} + \left\| \boldsymbol{u}_{\text{avg}}^{(t-1)} \right\|_{\infty} \right) \leq \frac{2}{t}.$$

Thus, $\sum_{s=t-m'+1}^{t-1} \left\| \boldsymbol{u}_{\text{avg}}^{(s+1)} - \boldsymbol{u}_{\text{avg}}^{(s)} \right\|_{\infty}$, the counterpart of $\sum_{s=t-m'+1}^{t-1} \left\| \boldsymbol{u}^{(s+1)} - \boldsymbol{u}^{(s)} \right\|_{\infty}$ in Theorem 5.1, can be bounded by $\sum_{s=t-m'+1}^{t-1} \frac{1}{s+1}$, irrelevant of the accumulated utility variation $P^{(T)}$.

## 6.2 FULL-INFORMATION SETTING

Unlike **InstUtil Rank** feedback, where we can plug the utility estimates into essentially any (full-information) no-regret learning oracle, the **AvgUtil Rank** setting requires an update rule that is

*stable* with respect to the perturbations in the accumulated (averaged) utilities, such as Follow-The-Regularized-Leader (Hazan et al., 2016). The reason is that we would like the strategies produced from the estimated averages $\widetilde{\boldsymbol{u}}_{\mathrm{avg}}^{(1)}, \ldots, \widetilde{\boldsymbol{u}}_{\mathrm{avg}}^{(t)}$ to remain close to those that would have been produced from the ground-truth sequence $\boldsymbol{u}_{\mathrm{avg}}^{(1)}, \ldots, \boldsymbol{u}_{\mathrm{avg}}^{(t)}$. With such stability, the regret of our learner can be controlled by the regret of this "ideal" learner, and hence remains sublinear up to the additional error induced by estimation.

**Assumption 6.1.** *The (full-information) online learning algorithm* Alg *satisfies the following condition: for any $T > 0$, $t \in [T]$, and sequences of utilities $\left(\boldsymbol{u}^{(s)}\right)_{s=1}^{t}, \left(\boldsymbol{u}'^{(s)}\right)_{s=1}^{t} \in \left(\mathbb{R}^{\mathcal{A}}\right)^{t}$, we have*

$$\left\| \mathrm{Alg}\left( \left(\boldsymbol{u}^{(s)}\right)_{s=1}^{t} \right) - \mathrm{Alg}\left( \left(\boldsymbol{u}'^{(s)}\right)_{s=1}^{t} \right) \right\| \leq L \left\| \sum_{s=1}^{t} \boldsymbol{u}^{(s)} - \sum_{s=1}^{t} \boldsymbol{u}'^{(s)} \right\|,$$

*where $L = \Theta\left(T^{-c}\right)$ for some constant $c \in (0,1)$.*

It can be verified that FTRL with any strongly convex regularizer satisfies this assumption (cf. Lemma L.3). Then, similar to Section 5.2, any online learning algorithm satisfying Assumption 6.1 can achieve a sublinear regret when equipped with the utility estimator in Algorithm 1. The overall procedure is summarized in Algorithm 3 and Figure 11, with the following guarantee.

**Theorem 6.2.** *Consider* **AvgUtil Rank** *with constant $\tau > 0$ and full-information feedback. By running Algorithm 3, for any $\delta \in (0, 1)$, $T > 0$, and any full-information no-regret learning algorithm with numeric utility feedback,* Alg, *that satisfies Assumption 6.1, by choosing $m$ properly, we have that with probability at least $1 - \delta$, $R^{(T),\mathrm{external}}$ satisfies*

$$R^{(T),\mathrm{external}} \leq R^{(T),\mathrm{external}}\left( \mathrm{Alg}, \left(\boldsymbol{u}^{(t)}\right)_{t=1}^{T} \right) + \mathcal{O}\left( LT^{\frac{5}{3}} \log\left( \frac{T}{\delta} \right) \right).$$

Theorem 6.2 implies that if we choose the stability parameter $L = \Theta\left(T^{-c}\right)$ with some $c > 2/3$, then the external regret $R^{(T),\mathrm{external}}$ becomes sublinear in $T$. This choice of $L$ can be realized by instantiating Alg as FTRL with suitable parameters (see Lemma L.3). Hence, the choice of $L$ will also affect $R^{(T),\mathrm{external}}\left( \mathrm{Alg}, \left(\boldsymbol{u}^{(t)}\right)_{t=1}^{T} \right)$. The formal statement of Theorem 6.2 and its proof can be found in Appendix I.

## 6.3 BANDIT SETTING

By applying Algorithm 1, we can only obtain an estimate of $\boldsymbol{u}_{\mathrm{empirical}}^{(t)}$ (cf. (3.2)) instead of $\boldsymbol{u}_{\mathrm{avg}}^{(t)}$, since it is the former that is used for ranking. However, almost all no-regret learning algorithms made decisions according to the accumulated utility, such as mirror descent (Hazan et al., 2016), FTRL, and regret matching (Zinkevich et al., 2007). Let $n^{(t)}(a) := \sum_{s=1}^{t} \#_{o^{(s)}}(a)$ for any $a \in \mathcal{A}$ be the number of times action $a$ has been proposed up to timestep $t$, where $\#_{o^{(s)}}(a)$ is the number of occurrences of $a$ in the proposed choices $o^{(s)}$ at timestep $s$. A natural idea is to compute $n^{(t)}(a)u_{\mathrm{empirical}}^{(t)}(a) - n^{(t-1)}(a)u_{\mathrm{empirical}}^{(t-1)}(a)$ to get an estimate of $u^{(t)}(a)$. Nonetheless, the variance of this estimator will be too large due to the multiplication of $n^{(t)}(a) \propto t$.

To address this issue, we divide the timesteps $\{1, 2, \ldots, t\}$ into $\lceil t/M \rceil$ blocks, with each block containing $M$ timesteps except for the last one. Then, for each block $\{s \cdot M + 1, s \cdot M + 2, \ldots, (s+1)M\}$ (for $s \leq \left\lfloor \frac{t}{M} \right\rfloor - 1$) and $a \in \mathcal{A}$, we estimate $\frac{1}{M} \sum_{s'=s \cdot M+1}^{(s+1)M} u^{(s')}(a)$ by computing

$$\frac{\widetilde{u}_{\mathrm{empirical}}^{((s+1) \cdot M)}(a) n^{((s+1) \cdot M)}(a) - \widetilde{u}_{\mathrm{empirical}}^{(s \cdot M)}(a) n^{(s \cdot M)}(a)}{n^{((s+1) \cdot M)}(a) - n^{(s \cdot M)}(a)}.$$

In this way, the coefficient multiplying $\widetilde{u}_{\mathrm{empirical}}^{((s+1) \cdot M)}(a)$ is now $\frac{n^{((s+1) \cdot M)}(a)}{n^{((s+1) \cdot M)}(a) - n^{(s \cdot M)}(a)} \propto \frac{t}{M}$. The choice of $M$ therefore induces a bias-variance trade-off: increasing $M$ reduces the variance, but the resulting quantity estimates the average utility of action $a$ *conditional on $a$ being selected* in that block, which can differ substantially from the block-average utility of $a$. This discrepancy becomes more pronounced for large $M$, since more utility variation can accumulate within a longer block. The full algorithm is introduced in Algorithm 3 and Figure 11.

**Theorem 6.3.** *Consider* **AvgUtil Rank** *with constant $\tau > 0$ and bandit feedback. By running Algorithm 3, for any $\delta \in (0,1)$, $T > 0$, and any full-information no-regret learning algorithm with numeric utility feedback,* Alg*, that satisfies Assumption 6.1, by choosing $m, \gamma$, and $M$ properly, when $P^{(T)} \leq \mathcal{O}(T^q)$ for some $q < \frac{1}{3}$ and $L = \Theta(T^{-c})$ with $c \in \left(\frac{5}{6} + \frac{q}{2}, 1\right)$, with probability at least $1 - \delta$, $R^{(T)}$ satisfies*

$$R^{(T)} \leq R^{(T),\text{external}}\left(\text{Alg}, \left(\boldsymbol{u}^{(t)}\right)_{t=1}^T\right) + \widetilde{\mathcal{O}}\left(\left(\log\left(\frac{1}{\delta}\right)\right)^2 L^{\frac{1}{3}} T^{\frac{23}{18}} \left(P^{(T)}\right)^{\frac{1}{6}}\right),$$

*where $\widetilde{\mathcal{O}}$ hides logarithmic dependence on $T$.*

In Theorem 6.3, we require $c < 1$ because, for many algorithms, one typically has an external-regret bound of the form $R^{(T),\text{external}}\left(\text{Alg}, \left(\boldsymbol{u}^{(t)}\right)_{t=1}^T\right) \leq \mathcal{O}\left(\frac{1}{L} + LT\right)$. See, *e.g.*, FTRL with any strongly convex regularizer as an example (cf. Lemma L.3 and Hazan et al. (2016)). In particular, taking $c = 1$ would make the $\frac{1}{L}$ term linear in $T$.

## 7 EQUILIBRIUM COMPUTATION WITH RANKING FEEDBACK

For a normal-form game $\left(N, \{\mathcal{A}_i\}_{i=1}^N, \{\mathcal{U}_i\}_{i=1}^N\right)$, the external regret of player $i \in [N]$ is defined as $R_i^{(T),\text{external}} := \max_{\widehat{\pi}_i \in \Delta^{\mathcal{A}_i}} \sum_{t=1}^T \left\langle \boldsymbol{u}_i^{(t)}, \widehat{\pi}_i - \pi_i^{(t)} \right\rangle$, where $\pi_i^{(t)} \in \Delta^{\mathcal{A}_i}$ is the strategy of player $i$ at timestep $t$ and $\boldsymbol{u}_i^{(t)}(a_i) = \sum_{\boldsymbol{a}' \in \times_{j=1}^N \mathcal{A}_j} \mathcal{U}_i(\boldsymbol{a}') \mathbb{1}(a_i' = a_i) \prod_{j' \neq i} \pi_{j'}^{(t)}(a_{j'}')$ for any $a_i \in \mathcal{A}_i$. Then, it is known that the time-average joint strategy $\pi_{\text{avg}}^{(T)}$, where $\pi_{\text{avg}}^{(T)}(\boldsymbol{a}) := \frac{1}{T} \sum_{t=1}^T \prod_{i \in [N]} \pi_i^{(t)}(a_i)$ for any $\boldsymbol{a} \in \times_{i=1}^N \mathcal{A}_i$, is an $\epsilon$-CCE, with $\epsilon := \max_{i \in [N]} \left\{\frac{1}{T} R_i^{(T),\text{external}}\right\}$.

Applying the algorithm in Section 5 (for **InstUtil Rank** feedback) or Section 6 (for **AvgUtil Rank** feedback), we achieve sublinear $R_i^{(T),\text{external}}$ for each player $i \in [N]$. Note that $P^{(T)}$ in Assumption 4.2 can be bounded by the summation of all players' strategy variation (see Lemma K.1). Thus, to ensure $P^{(T)}$ is sublinear in $T$, Alg needs to additionally satisfy the following assumption.

**Assumption 7.1** (Sublinear variation of strategies). *The (full-information) online learning algorithm* Alg *needs to satisfy the following condition: for any $T > 0$, $t \in [T-1]$, and sequence of utility vectors $\left(\boldsymbol{u}^{(s)}\right)_{s=1}^{t+1} \in \left([-1,1]^{\mathcal{A}}\right)^{t+1}$, we have $\left\|\text{Alg}\left(\left(\boldsymbol{u}^{(s)}\right)_{s=1}^t\right) - \text{Alg}\left(\left(\boldsymbol{u}^{(s)}\right)_{s=1}^{t+1}\right)\right\| \leq \eta$, where $\eta = \Theta(T^{-w})$ for some constant $w \in (0,1)$.*

Mirror descent (cf. Wei et al. (2021, Lemma 1) and Liu et al. (2023, Lemma C.5)) and FTRL with any strongly convex regularizer both satisfy this property, see Lemma L.3 for the proof. When Assumption 7.1 is satisfied, one can achieve sublinear regret with **InstUtil Rank**, under both full-information and bandit feedback. The formal statement is as follows.

**Theorem 7.2.** *Consider* **InstUtil Rank** *with constant $\tau > 0$ and Algorithm 2. For any $\delta \in (0,1)$, $T > 0$, and any full-information no-regret learning algorithm with numeric utility feedback,* Alg*, that satisfies Assumption 7.1, by choosing $M, m, \gamma$ according to Theorem 5.2, we have that with probability at least $1 - \delta$, the algorithm finds an $\epsilon$-CCE, with*

$$\epsilon \leq \max_{i \in [N]} \left\{\frac{1}{T} R_i^{(T),\text{external}}\left(\text{Alg}, \left(\widetilde{\boldsymbol{u}}_i^{(t)}\right)_{t=1}^T\right)\right\} + \mathcal{O}\left(\eta^{\frac{1}{3}} \left(\log\left(\frac{T}{\delta}\right)\right)^{\frac{1}{3}}\right) \qquad \text{(Full-Information)}$$

$$\epsilon \leq \max_{i \in [N]} \left\{\frac{1}{T} R_i^{(T),\text{external}}\left(\text{Alg}, \left(\widetilde{\boldsymbol{u}}_i^{(t)}\right)_{t=1}^T\right)\right\} + \mathcal{O}\left(\eta^{\frac{1}{5}} \log\left(\frac{T}{\delta}\right)\right). \qquad \text{(Bandit)}$$

With **AvgUtil Rank** feedback, when all the players apply Algorithm 3 and both Assumption 6.1 and Assumption 7.1 are satisfied by the oracle Alg being used, the external regret of each player will be sublinear in $T$ according to Theorem 6.2. Finally, we have the statement below.

**Theorem 7.3.** *Consider* **AvgUtil Rank** *with constant $\tau > 0$ and Algorithm 3. For any $\delta \in (0,1)$, $T > 0$, and any full-information no-regret learning algorithm with numeric utility feedback,* Alg*,*

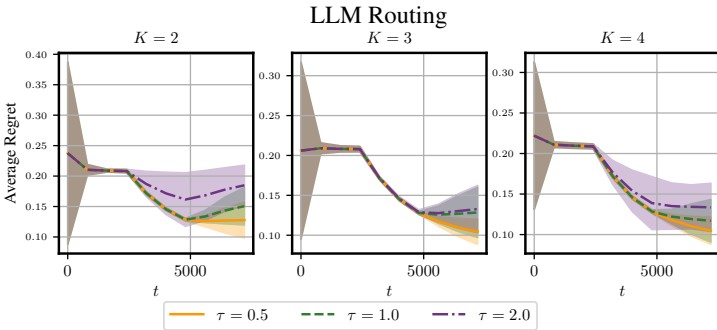

Figure 2: Regret of Algorithm 3 with **AvgUtil Rank** under bandit feedback for different temperatures $\tau$ and numbers of proposed actions $K$, in the online learning setting.

*that satisfies Assumption 6.1, by choosing $M, m, \gamma$ according to Theorem 6.2, we have that with probability at least $1 - \delta$, the algorithm finds an $\epsilon$-CCE under full-information feedback, with*

$$\epsilon \leq \max_{i \in [N]} \left\{ \frac{1}{T} R_i^{(T),\text{external}} \left( \text{Alg}, \left( \boldsymbol{u}_i^{(t)} \right)_{t=1}^T \right) \right\} + \mathcal{O}\left( LT^{\frac{5}{3}} \log\left( \frac{T}{\delta} \right) \right). \qquad \text{(Full-Information)}$$

*When $M$, $m$, and $\gamma$ are chosen as in Theorem 6.3, and both Assumption 6.1 and Assumption 7.1 hold, we have that with probability at least $1 - \delta$, the algorithm finds an $\epsilon$-CCE under bandit feedback, with*

$$\epsilon \leq \max_{i \in [N]} \left\{ \frac{1}{T} R_i^{(T),\text{external}} \left( \text{Alg}, \left( \boldsymbol{u}_i^{(t)} \right)_{t=1}^T \right) \right\} + \widetilde{\mathcal{O}}\left( \left( \log\left( \frac{1}{\delta} \right) \right)^2 \left( L^{\frac{1}{3}} \eta^{\frac{1}{6}} + L^{\frac{1}{2}} \right) T^{\frac{4}{9}} \right). \qquad \text{(Bandit)}$$

Lastly, we would like to remark that although the online learning setting can be hard with a small $\tau$ (cf. the hardness results in Theorem 4.3 and Theorem 4.4), computing an equilibrium is still possible even when $\tau \to 0^+$. A detailed discussion can be found in Remark K.3.

## 8 EXPERIMENTS

To demonstrate the practical merit of online learning with ranking feedback, we consider the following scenario as an application example, which we term as *online large-language-model routing*. A service provider hosts multiple language models, each possessing distinct strengths. For instance, GPT-4o (Hurst et al., 2024) may excel in creative writing, whereas GPT-5 specializes in coding. Consequently, users may exhibit different preferences for different models.

We cast personalized large language model (LLM) routing as an online learning problem with ranking feedback. Fix a particular user and view each candidate model as an action. At each timestep $t$, the user submits a query, and the server samples $K$ candidate responses by drawing models according to a mixed strategy $\pi^{(t)}$ and querying the selected models on that prompt. The user then returns a ranking over the proposed responses, providing only relative-preference feedback rather than calibrated scores. This feedback is induced by an underlying, time-varying utility vector $\boldsymbol{u}^{(t)}$ that captures the user's instantaneous (or time-aggregated) satisfaction with each model. The server's objective is to update $\pi^{(t)}$ online to minimize external regret with respect to the best fixed model in hindsight, while remaining responsive to nonstationary preferences. For instance, a user may prioritize creative writing for a period and later shift toward coding or mathematical problem solving.

We simulate this process in Figure 2. Specifically, we generate responses for the HH-RLHF (Bai et al., 2022) dataset using Qwen3-32B (Yang et al., 2025), Phi-4 (Abdin et al., 2024), GPT-4o (Hurst et al., 2024), and Llama-3.1-70B (Dubey et al., 2024). We evaluate the responses using a reward model[1] and execute Algorithm 3 with **AvgUtil Rank** under bandit feedback. At each time step, a prompt is sampled from the dataset along with responses from different models, and the resulting scores form the reward vector. As shown in Figure 2, the average regret decreases over time, suggesting that our router quickly approaches the performance of always serving the best fixed model in hindsight, *i.e.*, the model with the highest cumulative reward under the user's preferences. More experiments about online learning and equilibrium computation can be found in Appendix C.

---

[1]https://huggingface.co/OpenAssistant/reward-model-deberta-v3-large-v2.

## ACKNOWLEDGEMENT

The authors thank the anonymous reviewers and area chair for the value feedback. M.L. was supported by the MathWorks Fellowship. A.O. and G.F. were supported in part by the ONR grant N000142512296. G.F. was additionally supported by CCF-2443068 and an AI2050 Early Career Fellowship. K.Z. was supported by the ARO grant W911NF-24-1-0085, the NSF CAREER Award 2443704, and the AFOSR YIP Award FA9550-25-1-0258. Y.C.'s work was done while visiting UMD.

## 9 ETHICS STATEMENT

This research does not involve human participants, personally identifiable information, or sensitive data. All experiments were conducted under hypothetical and simulated environments. No animals or humans were harmed or involved in this study. The authors affirm that the work complies with ethical standards of the research community.

Furthermore, we have carefully considered the potential societal impacts of our research. While the proposed methods could be applied in various real-world settings, we acknowledge that any misuse, such as in surveillance or decision-making without fairness considerations, may raise ethical concerns. We strongly encourage the responsible use of our work and emphasize that it should not be deployed in contexts that may cause harm or reinforce social biases.

## 10 REPRODUCIBILITY STATEMENT

All assumptions are listed in Table 1 and the proofs are presented in the appendices, from Appendix D to Appendix L.

## 11 USE OF LARGE LANGUAGE MODELS

In this paper, we use large language models (LLMs) to improve writing (*e.g.*, by correcting grammatical errors), creating illustrative figures (Figure 1), and generating code.

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

## A  RELATED WORK

**Dueling Bandits.**  Using comparison and/or ranking feedback for sequential decision-making has mostly been studied under the framework of dueling bandits (Yue et al., 2012; Saha & Gaillard, 2022; Saha & Gopalan, 2019; Du et al., 2020; Saha et al., 2021; Dudík et al., 2015), where the agent takes two (or multiple) actions at each timestep, and receives a ranking of them as feedback. Different from our setting, the ranking feedback in these works was only based on the *instantaneous* utility at that timestep, while our results can address *both* settings with instantaneous and time-average utilities for ranking.  More importantly, the regret notions studied in these works were particularly designed for the dueling-bandit setting, and thus different from the classical external regret we focus on here. Finally, dueling bandits mostly focused on environments that are *stationary* and *stochastic* (Yue et al., 2012; Saha & Gaillard, 2022; Saha & Gopalan, 2019; Du et al., 2020), while we focus on the *non-stochastic* setting where the environment is arbitrary and potentially *adversarial*, as in online learning (Shalev-Shwartz et al., 2012; Hazan et al., 2016).  Due to the last two differences, the implication of these dueling-bandit algorithms in the game-theoretic setting is unclear, while our algorithms find an approximate CCE of the game, as a corollary of the no-(external-)regret guarantee.

**Reinforcement Learning from Human Feedback (RLHF) and Preference-Based RL.**  Inspired by the successes in aligning large language models (LLMs) (Ouyang et al., 2022), reinforcement learning from human feedback has received increasing attention.  RLHF is usually instantiated as *preference-based* learning, where humans rank the model outputs based on their preferences, and a reward model is then estimated from the feedback, which will be further used for model fine-tuning. This way, RLHF is oftentimes implemented in an *offline* fashion, where batch feedback data are used for reward model estimation (Ziegler et al., 2019; Bai et al., 2022; Ouyang et al., 2022; Zhu et al., 2023; Park et al., 2024).  Recently, *online* versions of RLHF have also been developed (Dwaracherla et al., 2024; Du et al., 2024; Xie et al., 2025; Cen et al., 2025; Zhang et al., 2025), where the *exploration* issue was addressed with online feedback.  In fact, beyond fine-tuning LLMs, preference-based RL has also been studied in the classical Markov decision process model with online feedback (Novoseller et al., 2020; Saha et al., 2023; Xu et al., 2020).  However, the utility/reward functions in these works are again stationary, and the regret notions extend those in the dueling-bandits literature, which are thus different from ours. Hence, these results do not apply to our adversarial online learning and game-theoretic settings.

**Learning of Stable Matchings.**  Some of our motivating scenarios for the game-theoretic setting may also be modeled as the *stable matching* problem (Gale & Shapley, 1962), which has been extensively studied when the agents have full knowledge of their preferences.  Recently, growing efforts have been devoted to *learning* in stable matching markets with unknown preferences, and through interactions between the agents (Liu et al., 2020; 2021; Basu et al., 2021; Jagadeesan et al., 2021; Etesami & Srikant, 2025; Shah et al., 2024b;a).  Notably, Etesami & Srikant (2025); Shah et al. (2024b;a) also took a *game-theoretic* perspective, by developing learning dynamics for finding matchings in a decentralized, uncoordinated fashion. However, one key difference is that the learning agents (*e.g.*, the proposers or the platform) can still receive *numeric* feedback of the utilities each round, based on the matching result, while in our model, they can only receive the ranking feedback. Moreover, the learning dynamics in Etesami & Srikant (2025); Shah et al. (2024a;b) were specific to the matching model, while ours aim to address general normal-form games.

**Recent Work by Maran, Bacchiocchi, Stradi, Castiglioni, Gatti, and Restelli (2024).**  The work closest to ours is the recent one by Maran et al. (2024), which studied multi-armed bandits with ranking feedback, also under the standard (external-)regret metric. Different from the ranking model in dueling bandits, the model of Maran et al. (2024) is based on time-average utilities, a setting also considered in our paper. More importantly, in contrast to our paper, Maran et al. (2024) focused on the *stochastic* bandits setting where the utility functions are stationary, while our focus is on the *adversarial/online* and game-theoretic settings, with *both* instantaneous and time-average utility-based rankings. Furthermore, the ranking model in Maran et al. (2024) corresponds to the case of $\tau \to 0^+$ in our framework. Finally and notably, Maran et al. (2024) also provided a hardness result for the adversarial bandit setting (with $\tau \to 0^+$ in our framework), while our hardness results (with

different hard instances) are stronger in the sense that they allow a wider range of $\tau$ for the bandit setting, and also cover the full-information setting (cf. Table 1).

# B    ADDITIONAL NOTATION AND PRELIMINARIES

**Notation.**    For any integer $N > 0$, we define $[N] \coloneqq \{1, ..., N\}$ to denote the set of positive integers no larger than $N$. We use bold notation $\boldsymbol{x}$ to denote a finite-dimensional vector, and $x_i$ to denote the $i^{th}$ element of the vector. For any discrete set $\mathcal{S}$, let $|\mathcal{S}|$ denote its cardinality, $\Delta^{\mathcal{S}} \coloneqq \big\{ \boldsymbol{x} \in \mathbb{R}^{\mathcal{S}} \colon \sum_{s \in \mathcal{S}} x_s = 1, \ x_s \geq 0 \text{ for all } s \in \mathcal{S} \big\}$ be the probability simplex over $\mathcal{S}$, and $\mathbf{1}(\mathcal{S})$ be an all-one vector with each index being elements in $\mathcal{S}$. For any ordered discrete set $\mathcal{S}$, we use $\mathbb{R}^{\mathcal{S}}$ to denote the $|\mathcal{S}|$ dimensional real space, where the $s^{th} \in \mathcal{S}$ element of any $\boldsymbol{x} \in \mathbb{R}^{\mathcal{S}}$ is denoted as $x_s$ or $x(s)$. For any vector $\boldsymbol{x} \in \mathbb{R}^m$, let $\|\boldsymbol{x}\|_p$ be its $L_p$-norm and we use $\|\boldsymbol{x}\|$ to denote the $L_2$-norm by default. For any convex compact set $\mathcal{C} \subseteq \mathbb{R}^n$ and $\boldsymbol{x} \in \mathbb{R}^n$, let $\mathrm{Proj}_{\mathcal{C}}(\boldsymbol{x}) = \mathrm{argmin}_{\boldsymbol{x}' \in \mathcal{C}} \|\boldsymbol{x} - \boldsymbol{x}'\|$. For any event $e$, let $\mathbb{1}(e)$ be its indicator, which is equal to one when $e$ holds and zero otherwise. Additionally, for any discrete set $\mathcal{S}$, let $\Sigma(\mathcal{S})$ be the set containing all the permutations of the elements in $\mathcal{S}$. We will use $\mathrm{sig}(x) \coloneqq \frac{\exp(x)}{1+\exp(x)} \colon \mathbb{R} \to \mathbb{R}$ to denote the logistic function.

## B.1    NORMAL-FORM GAMES

An $N$-player normal-form game can be characterized by a tuple $\left(N, \{\mathcal{A}_i\}_{i=1}^{N}, \{\mathcal{U}_i\}_{i=1}^{N}\right)$, where $\mathcal{A}_i \coloneqq \left\{a_i^1, a_i^2, \ldots, a_i^{|\mathcal{A}_i|}\right\}$ is the (finite) action set for player $i \in [N]$; $\mathcal{U}_i \colon \bigtimes_{i=1}^{N} \mathcal{A}_i \to [-1, 1]$ ($\bigtimes$ denotes the Cartesian product of sets) is the utility function of player $i$, where $\mathcal{U}_i(a_1, a_2, ..., a_N)$ is the utility of player $i$ when player $j \in [N]$ takes action $a_j$. We call $\boldsymbol{a} \colon = (a_1, a_2, ..., a_N)$ the *joint action* and let $\boldsymbol{a}_{-i} \colon = (a_1, ..., a_{i-1}, a_{i+1}, ..., a_N)$. Player $i \in [N]$ can choose a strategy $\pi_i \in \Delta^{\mathcal{A}_i}$, and we call $\bigtimes_{i=1}^{N} \Delta^{\mathcal{A}_i} \ni \boldsymbol{\pi} = (\pi_1, \pi_2, \ldots, \pi_N)$ a *strategy profile*. When a strategy profile $\boldsymbol{\pi}$ is implemented, each player $i \in [N]$ has an expected utility of $\sum_{\boldsymbol{a} \in \bigtimes_{j=1}^{N} \mathcal{A}_j} \mathcal{U}_i(\boldsymbol{a}) \prod_{j \in [N]} \pi_j(a_j)$. Lastly, we use the unbold notation $\pi \in \Delta^{\bigtimes_{i=1}^{N} \mathcal{A}_i}$ to denote the (possibly correlated) joint strategy of all the players, where $\pi(\boldsymbol{a})$ is the probability of choosing the joint action $\boldsymbol{a} \in \bigtimes_{i=1}^{N} \mathcal{A}_i$.

In this paper, we focus on finding an $\epsilon$-approximate *coarse correlated equilibrium* ($\epsilon$-CCE) of the NFG, which is a probability distribution over the joint action set. It is formally defined as follows:

**Definition B.1** ($\epsilon$-CCE). *For any joint strategy $\pi \in \Delta^{\bigtimes_{i=1}^{N} \mathcal{A}_i}$, it is an $\epsilon$-CCE for $\epsilon \geq 0$ if*

$$\max_{i \in [N]} \max_{\widehat{\pi}_i \in \Delta^{\mathcal{A}_i}} \sum_{\boldsymbol{a} \in \bigtimes_{j=1}^{N} \mathcal{A}_j} \mathcal{U}_i(\boldsymbol{a}) \left( \widehat{\pi}_i(a_i) \sum_{a_i' \in \mathcal{A}_i} \pi(a_i', \boldsymbol{a}_{-i}) - \pi(\boldsymbol{a}) \right) \leq \epsilon. \qquad (\epsilon\text{-CCE})$$

When $\epsilon = 0$, we refer to it as a(n exact) CCE. We also refer to $\epsilon$ as the *exploitability*.

## B.2    EQUILIBRIUM COMPUTATION WITH RANKING FEEDBACK

There is a mediator (platform) in the game that computes strategies for the players, (*e.g.*, Uber recommends the candidate drivers and users to each other), but with only access to the ranking feedback from the players, *e.g.*, humans. Specifically, when the strategy profile $\boldsymbol{\pi}$ is implemented by the players, player $i$'s utility of taking action $a_i \in \mathcal{A}_i$ is $u_i^{\boldsymbol{\pi}}(a_i) \coloneqq \sum_{\boldsymbol{a}' \in \bigtimes_{j=1}^{N} \mathcal{A}_j} \mathcal{U}_i(\boldsymbol{a}') \mathbb{1}(a_i' = a_i) \prod_{j \neq i} \pi_j(a_j')$. However, instead of observing the utility directly, the mediator can only observe the ranking based on it. Therefore, at each timestep $t$, the mediator will choose a strategy profile $\boldsymbol{\pi}$ and propose each player $i \in [N]$ a multiset $o_i^{(t)} = \left\{a_i^{(t),k}\right\}_{k=1}^{K}$ consisting of $K$ actions, and in different settings proceed differently as follows:

- **Full-information setting.** *All* the actions of each player $i \in [N]$ can be evaluated and ranked at each timestep $t$ based on some utility vector, which is $\boldsymbol{u}_i^{\boldsymbol{\pi}^{(t)}}$ under **InstUtil Rank**

and $\boldsymbol{u}_{\text{avg}}^{(t)} := \frac{1}{t} \sum_{s=1}^{t} \boldsymbol{u}_i^{\boldsymbol{\pi}^{(s)}}$ under **AvgUtil Rank**, where $\boldsymbol{\pi}^{(t)} = \left( \pi_1^{(t)}, \ldots, \pi_N^{(t)} \right)$ is the strategy profile at timestep $t$.

- **Bandit setting.** For each player $i \in [N]$, only the $K$ actions in $o_i^{(t)}$ that are proposed at timestep $t$ will be evaluated and ranked, with the associated elements in some utility vector. Specifically, under **InstUtil Rank**, $\widehat{\boldsymbol{u}}_i^{(t)}$ defined below will be used: for each $a_i \in \mathcal{A}_i$

$$\widehat{u}_i^{(t)}(a_i) := \frac{1}{|o_{-i}^{(t)}|} \sum_{\boldsymbol{a}'_{-i} \in o_{-i}^{(t)}} \mathcal{U}_i(a_i, \boldsymbol{a}'_{-i});$$

under **AvgUtil Rank**, the corresponding empirical average utility is as computed in (3.2), with the $\boldsymbol{u}^{(s)}$ therein being replaced by the $\widehat{\boldsymbol{u}}_i^{(s)}$ above. As in the online setting, we assume that the actions are proposed in an *unbiased* way, *i.e.*, $\mathbb{E} \left[ \frac{\sum_{\boldsymbol{a}_{-i} \in o_{-i}^{(t)}} \mathcal{U}_i(a_i, \boldsymbol{a}_{-i})}{|o_{-i}^{(t)}|} \right] = \left\langle \mathcal{U}_i(a_i, \cdot), \pi_{-i}^{(t)} \right\rangle$, for all $a_i \in \mathcal{A}_i$. In other words, $\widehat{\boldsymbol{u}}_i^{(t)}$ is an unbiased estimate of $\boldsymbol{u}_i^{\boldsymbol{\pi}^{(t)}}$.

The process will be repeated until the mediator finds an (approximate) equilibrium of the game, which is the average of the joint strategy over all timesteps.

## C ADDITIONAL EXPERIMENTS

To assess robustness across learning setups, we study both full-information and bandit feedback under the two ranking-feedback models in **InstUtil Rank** and **AvgUtil Rank**. The corresponding regret results are reported in Figures 3 to 6. We also evaluate our methods on randomly generated two-player general-sum games under the same feedback models. For these game experiments, we report the resulting CCE exploitability $\epsilon$ across different game parameters in Figures 7 to 9, together with 95% confidence intervals.

We consider both full-information and bandit feedback. Throughout, all experiments run for $T = 10^7$ iterations, and each player has 10 actions. We evaluate performance across the number of proposed actions $K \in \{3, 5, 10\}$, the temperature parameter $\tau \in \{0.5, 1, 2\}$ in (PL), and the accumulated utility variation $P^{(T)} = T^q$ with $q \in \{0.3, 0.5, 0.7\}$. For **AvgUtil Rank** under full-information feedback, we consider only the case $q = 1.0$, which does not restrict the utility variation, since the corresponding guarantee does not rely on Assumption 4.2. Moreover, for **AvgUtil Rank** under bandit feedback, although 0.5 and 0.7 exceed the theoretical threshold $\frac{1}{3}$ required for Algorithm 3, we empirically observe that Algorithm 3 still attains sublinear regret. This suggests that Algorithm 3 is more robust in practice than the current theory indicates.

For hyper-parameter selection, we conduct a separate grid search for each experimental setting. Under **InstUtil Rank**, we tune the estimation window size $m$ and the exploration rate $\gamma$; under **AvgUtil Rank**, we additionally tune the block size $M$. Specifically, we search over $m \in \{5 \times 10^4, 10^5, 1.5 \times 10^5\}$, $\gamma \in \{0.1, 0.05, 0.01\}$, and, when applicable, $M \in \{3 \times 10^6, 5 \times 10^6, 10^7\}$. The learning rate is set to $\eta = 1/\sqrt{T}$ in all experiments except for bandit feedback under **AvgUtil Rank**, where we use $\eta = 10^{-6}$. Each hyper-parameter configuration is evaluated over 10 random seeds, namely $\{0, 1, \ldots, 9\}$. For each figure, we report the best-performing choice of $m$, $M$, and $\gamma$, namely, the one that achieves the smallest average regret or $\epsilon$ over all seeds at horizon $T$.

Utility estimation is performed using Algorithm 1. As the full-information no-regret learning oracle with numeric feedback, we instantiate Alg as PGD for **InstUtil Rank** and as FTRL with $L_2$-regularization for **AvgUtil Rank**.

In Figures 3 to 5, we report the performance of the learning algorithm for **InstUtil Rank** under both full-information and bandit feedback, and for **AvgUtil Rank** under bandit feedback. To generate utility sequences with cumulative variation bounded by $T^q$, we first allocate the total variation budget $T^q$ uniformly at random across timesteps. At each timestep $t$, we then sample a perturbation direction $\boldsymbol{n}^{(t)}$ uniformly at random and use binary search to choose a scaling factor $\alpha^{(t)}$ such that $\left\| \text{Proj}_{[-1,1]^A} \left( \boldsymbol{u}^{(t-1)} + \alpha^{(t)} \boldsymbol{n}^{(t)} \right) - \boldsymbol{u}^{(t-1)} \right\|$ equals the variation assigned to round $t$.

For **AvgUtil Rank** under full-information feedback, our algorithm can accommodate sequences of utility vectors with unbounded cumulative variation. We therefore additionally conduct an ablation study on the noise model in Figure 6. We first sample an initial utility vector, and at each timestep the utility vector is defined as the initial utility vector plus an independent random shift. For each coordinate, the shift is drawn from one of three noise distributions: $\mathrm{Uniform}\,(-\sigma, \sigma), \mathcal{N}\left(0, \sigma^2\right)$, or $\Gamma\left(1/\sigma^2, \sigma^2\right)$ with $\sigma = 0.3$. We report the resulting average regret over time in Figure 6.

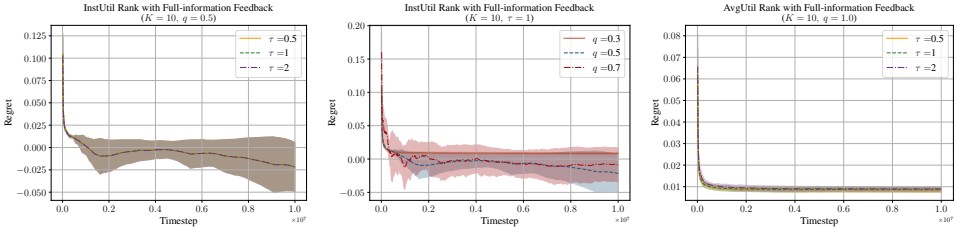

Figure 3: The exploitability for the full-information feedback setting under both **InstUtil Rank** and **AvgUtil Rank**. Performance is evaluated across different temperatures $\tau$ and cumulative utility variations $P^{(T)} = T^q$. Each parameter combination is tested 10 times with different random seeds.

Across all game settings, the exploitability decreases as $t$ increases, suggesting that the time-averaged joint strategy converges to a CCE.

# D PROOF OF SECTION 4

In this section, we will show the hardness results in Section 4.

## D.1 PROOF OF THEOREM 4.1

**Theorem 4.1.** *Consider* **InstUtil Rank**. *For any* $T > 0$, *temperature* $0 < \tau \leq 0.1$, *and online learning algorithm, there exists a sequence of utilities* $\left(\boldsymbol{u}^{(t)}\right)_{t=1}^{T}$ *such that* $\min\left\{\mathbb{E}\left[R^{(T),\mathrm{external}}\right], \mathbb{E}\left[R^{(T)}\right]\right\} \geq \Omega\left(T\right)$ *in both full-information and bandit feedback settings. The expectation is taken over the randomness of the algorithm and the ranking.*

*Proof.* Consider an online learning problem with $\mathcal{A} = \{a, b\}$, so that the utility vector can be represented as $(u(a), u(b))$. There are two instances with $\tau = 0.1$ and $K = 2$ under bandit feedback.

In the first instance, there are two types of utility vectors $(-0.5, 0)$ and $(0.15, 0)$. At each timestep, the adversary will choose $(-0.5, 0)$ with probability $\frac{4}{13}$ and the other with probability $\frac{9}{13}$.

In the second instance, there are two types of utility vectors $(-0.02, 0)$ and $(0.1, 0)$. Recall $\mathrm{sig}(x) \colon \mathbb{R} \to \mathbb{R} \coloneqq \frac{\exp(x)}{1+\exp(x)}$ is the logistic function. At each timestep, the adversary will choose

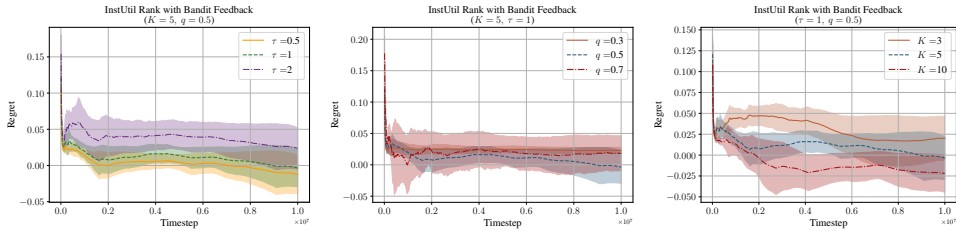

Figure 4: The regret for bandit feedback setting under **InstUtil Rank** feedback in the online learning setting. The performance is evaluated across different temperatures $\tau$, cumulative utility variations $P^{(T)} = T^q$, and numbers of proposed actions $K$. Each parameter combination is tested 10 times with different random seeds.

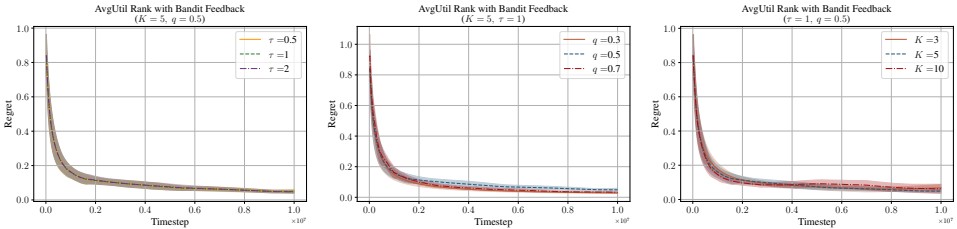

Figure 5: The regret for bandit feedback setting under **AvgUtil Rank** feedback in the online learning setting. The performance is evaluated across different temperatures $\tau$, cumulative utility variations $P^{(T)} = T^q$, and numbers of proposed actions $K$. Each parameter combination is tested 10 times with different random seeds.

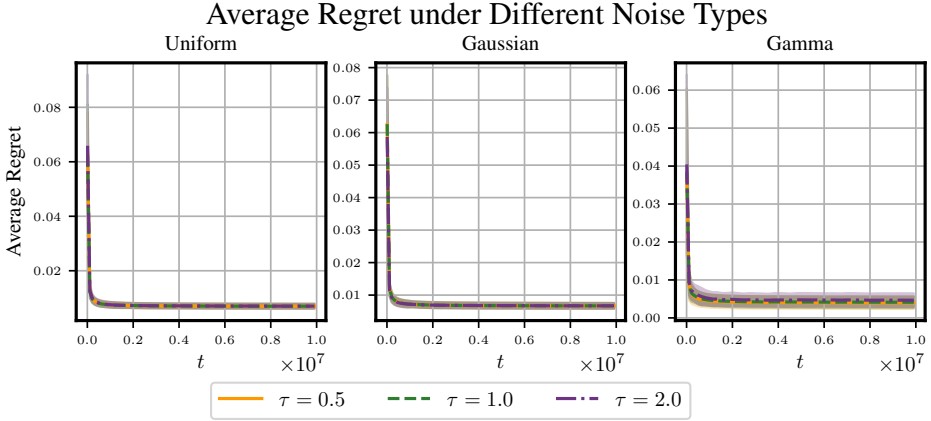

Figure 6: Regret of Algorithm 3 with **AvgUtil Rank** under full-information feedback, across different temperatures $\tau$ and noise types in the online-learning setting.

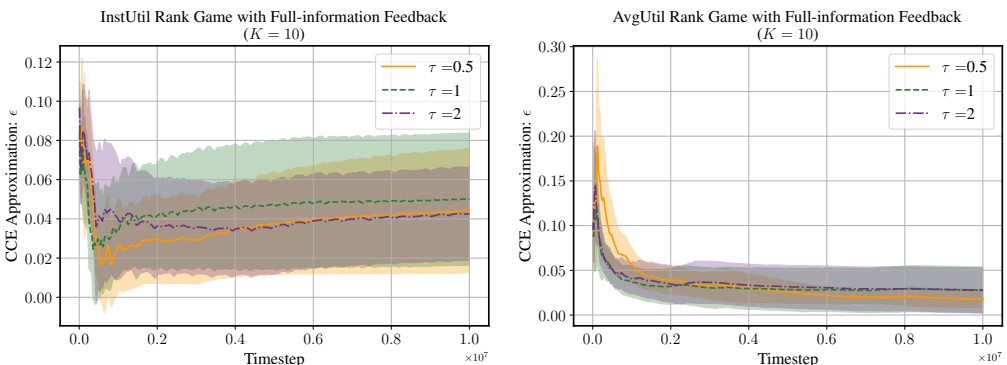

Figure 7: The exploitability for the full-information setting under both **InstUtil Rank** and **AvgUtil Rank** feedback in the game-play setting. The performance is tested under different temperatures $\tau$. Each parameter combination is tested 10 times with different random seeds.

$(-0.02, 0)$ with probability $\frac{4\text{sig}(-5)/13 + 9\text{sig}(1.5)/13 - \text{sig}(1)}{\text{sig}(-0.2) - \text{sig}(1)} \approx 0.58$ and the other with probability $1 - \frac{4\text{sig}(-5)/13 + 9\text{sig}(1.5)/13 - \text{sig}(1)}{\text{sig}(-0.2) - \text{sig}(1)}$.

The expected utility of action $b$ in both instances is 0. The expected utility of action $a$ in the first instance is $-0.05$. The expected utility of action $a$ in the second instance is 0.03.

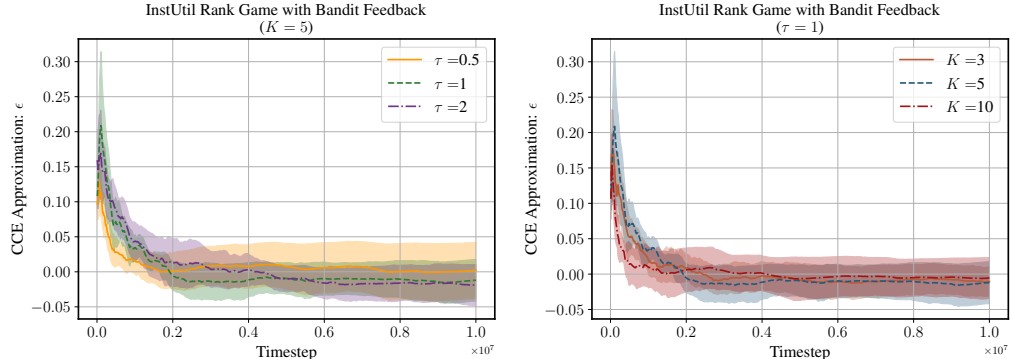

Figure 8: The exploitability for the bandit feedback setting under **InstUtil Rank** feedback in the game-play setting. Performance is evaluated across different temperatures $\tau$ and numbers of proposed actions $K$. Each parameter combination is tested 10 times with different random seeds.

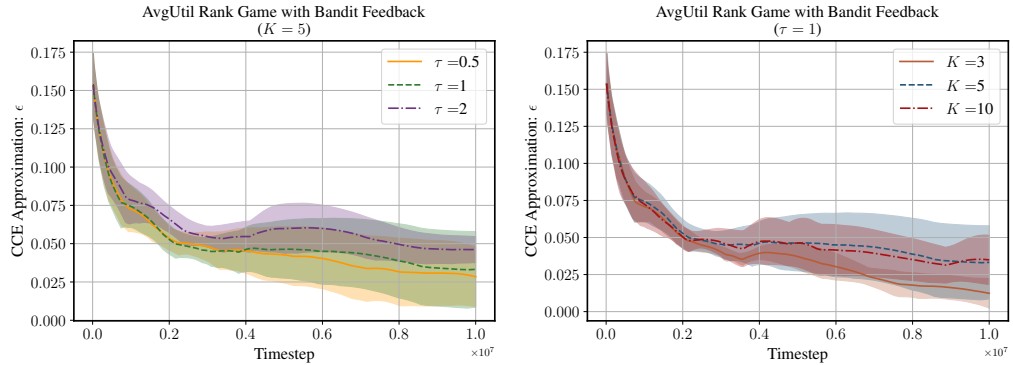

Figure 9: The exploitability for the bandit feedback setting under **AvgUtil Rank** feedback in the game-play setting. Performance is evaluated across different temperatures $\tau$ and numbers of proposed actions $K$. Each parameter combination is tested 10 times with different random seeds.

Moreover, at any timestep $t$ when both actions $a, b$ are proposed in $o^{(t)}$, in the ranking $\sigma^{(t)}$, the probability of $a$ being preferred over $b$ is

$$\Pr\left(a \overset{\sigma^{(t)}}{>} b\right) = \frac{4}{13}\mathrm{sig}(-5) + \frac{9}{13}\mathrm{sig}(1.5).$$

This coincides with the corresponding probability under the second instance,

$$\frac{4\mathrm{sig}(-5)/13 + 9\mathrm{sig}(1.5)/13 - \mathrm{sig}(1)}{\mathrm{sig}(-0.2) - \mathrm{sig}(1)}\mathrm{sig}(-0.2) + \left(1 - \frac{4\mathrm{sig}(-5)/13 + 9\mathrm{sig}(1.5)/13 - \mathrm{sig}(1)}{\mathrm{sig}(-0.2) - \mathrm{sig}(1)}\right)\mathrm{sig}(1).$$

Hence, the distribution of the ranking $\sigma^{(t)}$ is identical in the two instances. When $o^{(t)} = \{a, b\}$, the preceding equality shows that $\Pr\left(a \overset{\sigma^{(t)}}{>} b\right)$ matches across instances. When $o^{(t)} = \{a, a\}$ (or $\{b, b\}$), the ranking is deterministic, so it is trivially the same in both instances.

Therefore, under full-information feedback, for any algorithm that generates $\left(\pi^{(t)}\right)_{t=1}^{T}$, we have

$$\mathbb{E}\left[\sum_{t=1}^{T}\left\langle \boldsymbol{u}^{(t)}, \pi^{(t)}\right\rangle\right] = \sum_{t=1}^{T}\mathbb{E}\left[\left\langle \boldsymbol{u}^{(t)}, \pi^{(t)}\right\rangle\right] \overset{(i)}{=} \sum_{t=1}^{T}\left\langle \mathbb{E}\left[\boldsymbol{u}^{(t)}\right], \mathbb{E}\left[\pi^{(t)}\right]\right\rangle.$$

The equality $(i)$ holds because $\boldsymbol{u}^{(t)}$ is independent of $\pi^{(t)}$ given our process of generating both instances. Similarly, under bandit feedback,

$$\mathbb{E}\left[\sum_{t=1}^{T} \frac{1}{K} \sum_{a \in o^{(t)}} u^{(t)}(a)\right] = \sum_{t=1}^{T} \mathbb{E}\left[\frac{1}{K} \sum_{a \in o^{(t)}} u^{(t)}(a)\right] \stackrel{(i)}{=} \sum_{t=1}^{T}\left\langle\mathbb{E}\left[\boldsymbol{u}^{(t)}\right], \mathbb{E}\left[\pi^{(t)}\right]\right\rangle.$$

In $(i)$, we define $\pi^{(t)}(a) := \frac{1}{K} \sum_{a' \in o^{(t)}} \mathbb{1}\left(a = a'\right)$ for any $a \in \mathcal{A}$.

Then, for both the full-information feedback and the bandit feedback, we have

$$\mathbb{E}\left[\pi^{(t)}\right] = \sum_{\sigma^{(1)}, \ldots, \sigma^{(t-1)} \in \Sigma(\mathcal{A})} \mathbb{P}\left(\sigma^{(1)}, \ldots, \sigma^{(t-1)}\right) \mathbb{E}\left[\pi^{(t)} \mid \sigma^{(1)}, \ldots, \sigma^{(t-1)}\right].$$

The first term $\mathbb{P}\left(\sigma^{(1)}, \ldots, \sigma^{(t-1)}\right)$ is equal in the two instances according to the discussion above, and the second term $\mathbb{E}\left[\pi^{(t)} \mid \sigma^{(1)}, \ldots, \sigma^{(t-1)}\right]$ is also equal since it only depends on the algorithm. Therefore, $\mathbb{E}\left[\pi^{(t)}\right]$ is the same in both instances.

However, $\mathbb{E}\left[\boldsymbol{u}^{(t)}\right] = (-0.05, 0)$ in the first instance but $(0.03, 0)$ in the second. Therefore, whenever achieving sublinear regret in the first instance, the algorithm will suffer a linear regret in the second instance, and vice versa. □

## D.2 PROOF OF THEOREM 4.3

**Theorem 4.3.** *Consider* **AvgUtil Rank** *with full-information feedback. For any $T > 0$, temperature $0 < \tau \leq \mathcal{O}\left(\frac{1}{T \log T}\right)$, and online learning algorithm, there exists $T' \geq T$ and a sequence of utilities $\left(\boldsymbol{u}^{(t)}\right)_{t=1}^{T'}$ such that $\mathbb{E}\left[R^{(T'),\text{external}}\right] \geq \widetilde{\Omega}\left(T'\right)$. The expectation is taken over the randomness of the algorithm and the ranking.*

*Proof.* We use $(u(a), u(b))$ to denote the utility vector when the action set is $\mathcal{A} = \{a, b\}$. In the following, we will show a hard instance for $\tau \to 0^+$, *i.e.*, we always observe the action with higher utility ranks first in the permutation. Then, we will show that $\tau \leq \mathcal{O}(\frac{1}{T \log T})$ can be reduced to $\tau \to 0^+$.

The utility vector at timestep 1 is $\boldsymbol{u}^{(1)} = (0.5, 0)$. We will construct the rest of the utility vectors next.

We call the following an *action-a construction*, since, except for the last timestep, the observation is that action $a$ is always better. Let $K \in \mathbb{N}$ be the smallest integer such that $2^K \geq T$, then:

$$\begin{aligned}
\text{Sequence } 0 =&(0,1), (0,0) \\
\text{Sequence } 1 =&(1,0), (0,1), (0,1), (0,0) \\
\text{Sequence } 2 =&(1,0), (1,0), (1,0), (0,1), (0,1), (0,1), (0,1), (0,0) \\
&\cdots \\
\text{Sequence } K-1 =&\underbrace{(1,0), ..., (1,0)}_{2^{K-1}-1}, \underbrace{(0,1), ..., (0,1)}_{2^{K-1}}, (0,0) \\
\text{Sequence } K =&\underbrace{(1,0), ..., (1,0)}_{2^K-1}, (0,0).
\end{aligned} \tag{D.1}$$

Lemma D.1 in the following shows that at least one of the sequences will incur a low average utility for the algorithm.

**Lemma D.1.** *Consider* (D.1)*. For any online learning algorithm, at least one of the $K+1$ sequences satisfies that the expected average utility per timestep is less than $0.5 - \frac{1}{2(K+1)}$.*

By Lemma D.1, there exists a sequence with length $2^k$ for some $k \leq K$ such that the average utility per timestep achieved by the algorithm is less than $0.5 - \frac{1}{2(K+1)}$. We will pick this sequence as the

next $2^k$ utility vectors. If the current utility vector sequence is no less than $T$, then the hard instance is completed. Otherwise, we will establish the following *action-b construction*:

$$
\begin{aligned}
\text{Sequence } 0 =& (1,0), (0,0) \\
\text{Sequence } 1 =& (0,1), (1,0), (1,0), (0,0) \\
\text{Sequence } 2 =& (0,1), (0,1), (0,1), (1,0), (1,0), (1,0), (1,0), (0,0) \\
& \cdots \\
\text{Sequence } K-1 =& \underbrace{(0,1), ..., (0,1)}_{2^{K-1}-1}, \underbrace{(1,0), ..., (1,0)}_{2^{K-1}}, (0,0) \\
\text{Sequence } K =& \underbrace{(0,1), ..., (0,1)}_{2^K-1}, (0,0).
\end{aligned}
$$

Similarly, except for the last observation, action $b$ is the best action in all the observations. Similar to Lemma D.1, we can show that at least one of the sequences incurs average utility per timestep less than $0.5 - \frac{1}{2(K+1)}$. We will add that sequence to the end of our hard instance.

Let $T' \geq T$ be the length of the final instance. Therefore, the average regret will be at least $\frac{1}{2(K+1)} - \frac{1}{T} \geq \Omega(\frac{1}{\log T})$. Because of the construction, the best action should get at least $0.5 - \frac{1}{T}$ utility per timestep.

When $\tau \leq \frac{1}{4T \log T}$, from the construction above, the difference between the *cumulative* utility of the actions is always $0.5$. By the definition in **AvgUtil Rank**, at any given timestep the probability that an action with lower average utility is preferred over an action with higher average utility is $1 - \text{sig}\left(\frac{0.5}{T\tau}\right) \leq \mathcal{O}\left(\frac{1}{T^2}\right)$. Applying a union bound over all timesteps, it follows that with probability at least $1 - \mathcal{O}\left(\frac{1}{T}\right)$, no such misranking occurs at any timestep. Hence, every permutation ranks the higher-utility action first throughout. In other words, with high probability, the algorithm incurs linear regret. $\square$

**Lemma D.1.** *Consider* (D.1). *For any online learning algorithm, at least one of the $K+1$ sequences satisfies that the expected average utility per timestep is less than $0.5 - \frac{1}{2(K+1)}$.*

*Proof.* Note that in this online learning setting, the strategy $\pi^{(t)}$ is determined by $\boldsymbol{u}^{(1)}, \ldots, \boldsymbol{u}^{(t-1)}$. Therefore, in all the sequences in the action-$a$ construction, since action $a$ is the best in all the observations, for any two sequences $k_1 \leq k_2$, the expectation of the strategies is the same for the first $2^{k_1+1}$ utility vectors. For simplicity, we will use $x^{(t)}$ to denote the probability of choosing action-$a$ at timestep $t$.

The average utility at sequence $0$ is $\frac{1-x^{(1)}}{2}$. The average utility at sequence $1$ is $\frac{x^{(1)}}{4} + \frac{1-x^{(2)}}{4} + \frac{1-x^{(3)}}{4}$. We can see that the utility contributed by $x^{(1)}$ to all the sequences is

$$
\frac{1-x^{(1)}}{2} + \frac{x^{(1)}}{4} + \frac{x^{(1)}}{8} + ... + \frac{x^{(1)}}{2^K} + \frac{x^{(1)}}{2^K} = \frac{1}{2}.
$$

Similarly, the contribution of $x^{(2)}, x^{(3)}$ is $\frac{1}{4}$. The contribution of $x^{(4)}, x^{(5)}, \ldots, x^{(7)}$ is $\frac{1}{8}$. Therefore, the total contribution of $x^{(1)}, ..., x^{(2^K-1)}$ is $\frac{K}{2}$. There are $K+1$ sequences in total, so that at least one of the sequences has average utility per timestep less than $\frac{K}{2(K+1)} = \frac{1}{2} - \frac{1}{2K+2}$. $\square$

## D.3 Proof of Theorem 4.4

**Theorem 4.4.** *Consider* **AvgUtil Rank** *with bandit feedback. For any $T > 0$, temperature $0 < \tau \leq \mathcal{O}\left(\frac{1}{\log T}\right)$, and online learning algorithm, there exists a sequence of utilities $\left(\boldsymbol{u}^{(t)}\right)_{t=1}^{4T}$ such that $\mathbb{E}\left[R^{(4T)}\right] \geq \Omega\left(T\right)$. The expectation is taken over the randomness of the algorithm and the ranking.*

*Proof.* Consider the following two instances. Both of them satisfy $\mathcal{A} = \{a, b\}$ and $K = 1$.

$$\text{Instance } 1 = \underbrace{(0.1, 0), \ldots, (0.1, 0)}_{T}, \underbrace{(0, 0.2), \ldots, (0, 0.2)}_{T}, \underbrace{(0, 1), \ldots, (0, 1)}_{2T}$$

$$\text{Instance } 2 = \underbrace{(0.1, 0), \ldots, (0.1, 0)}_{T}, \underbrace{(0, 0.2), \ldots, (0, 0.2)}_{T}, \underbrace{(0.4, 0.2), \ldots, (0.4, 0.2)}_{2T}.$$

We call the first $T$ timesteps as the first phase, the next $T$ timesteps as the second phase, and the last $2T$ timesteps as the third phase.

For any online learning algorithm to achieve sublinear expected external regret, it must propose action $a$ for at least $0.9T$ timesteps during the first phase with probability at least $\frac{1}{2}$, since otherwise the expected external regret in the first phase is linear. With probability at least $\frac{1}{2}$, after proposing at least $0.9T$ timesteps of $a$ in the first phase, the online learner must propose action $b$ for at least $\frac{0.2T - 0.1T - 0.01T}{0.2} = 0.45T$ timesteps during the second phase to achieve sublinear regret. Then, at the end of the second phase, with probability at least $\frac{1}{4}$, $u_{\text{empirical}}^{(2T)}(b) - u_{\text{empirical}}^{(2T)}(a) \geq \frac{0.45T \cdot 0.2}{0.1T + 0.45T} - 0.1 = \frac{7}{110}$.

Intuitively, during the third phase, achieving sublinear regret on Instance 1 requires that action $a$ not be proposed too many times. However, if the learner proposes action $a$ too infrequently, it may fail to sample action $a$ sufficiently often to distinguish Instance 1 from Instance 2, which would then lead to linear regret.

Formally, during the third phase of Instance 1, the algorithm needs to propose $b$ for at least $\frac{0.2T + 2T - 0.2T - 0.1T - 0.01T}{1} = 1.89T$ timesteps with probability at least $\frac{1}{2}$ after proposing at least $0.9T$ $a$ in the first phase and $0.45T$ $b$ in the second phase. In other words, $a$ is proposed by no more than $0.11T$ times. Then, in Instance 2, throughout the third phase

$$u_{\text{empirical}}^{(t)}(b) - u_{\text{empirical}}^{(t)}(a) \overset{(i)}{\geq} u_{\text{empirical}}^{(2T+0.11T)}(b) - u_{\text{empirical}}^{(2T+0.11T)}(a)$$
$$\geq \frac{0.45T \cdot 0.2}{0.1T + 0.45T} - \frac{0.9T \cdot 0.1 + 0.11T \cdot 0.4}{0.9T + 0.11T} \geq 0.03.$$

$(i)$ follows because, among all ways of placing the $0.11T$ exploration proposals of action $a$ within the third phase, assigning them to the first $0.11T$ timesteps minimizes the gap between the empirical average utility of actions $b$ and $a$.

Therefore, when $\tau \to 0^+$, the observations of Instance 1 and Instance 2 are the same with probability at least $\frac{1}{8}$. Then, with probability at least $\frac{1}{8}$, according to the discussion above, any learning algorithm will satisfy one of the following:

- Linear regret at timestep $T$;

- Linear regret at timestep $2T$;

- Linear regret at timestep $4T$ in either Instance 1 or Instance 2.

When $\tau \leq \frac{200}{3 \log T}$, at each timestep the action with the larger empirical average utility is ranked first with probability at least $1 - \mathcal{O}\left(T^{-2}\right)$. Taking a union bound over all $T$ timesteps, we obtain that this action is ranked first at every timestep with probability at least $1 - \mathcal{O}\left(T^{-1}\right)$. Therefore, it suffices to consider the limiting case $\tau \to 0^+$, which completes the proof. $\qquad\square$

## E   PROOF OF THEOREM 5.1

In this section, we proved the high probability bound for the utility estimation error, and with that, we gave the regret upper bound of our algorithm under **InstUtil Rank** feedback. Next, we will introduce the key lemma we used for utility estimation, Lemma E.1, which shows that the ranking of $K$ actions can be decomposed into pair-wise rankings.

---

**Algorithm 1** Utility Estimation with Action Permutations: Estimate$\left( \left\{ \sigma^{(s)} \right\}_{s=1}^{m'} \right)$

---

1: **Input:** A set consisting of $m'$ permutations of actions : $\left\{ \sigma^{(s)} \right\}_{s=1}^{m'}$ with $|\sigma^{(s)}| = K$ for all $s \in [m']$, and temperature $\tau > 0$.
2: **for** $j = 1, 2, \ldots, |\mathcal{A}| - 1$ **do**
3:     **for** $s = 1, \ldots, m'$ **do**
4:        Calculate $n_{j,1}^{(s)}, n_{j,2}^{(s)}$ defined as

$$n_{j,1}^{(s)} \coloneqq \sum_{i,k \in [K]} \mathbb{1}\left( \sigma^{(s)}(i) = a^j, \sigma^{(s)}(k) = a^{|\mathcal{A}|} \text{ and } i < k \right),$$

$$n_{j,2}^{(s)} \coloneqq \sum_{i,k \in [K]} \mathbb{1}\left( \sigma^{(s)}(i) = a^j, \sigma^{(s)}(k) = a^{|\mathcal{A}|} \text{ and } i > k \right).$$

5:     **end for**
6:     Let $\mathcal{T}_j \coloneqq \left\{ s \in [m'] : n_{j,1}^{(s)} + n_{j,2}^{(s)} > 0 \right\}$
7:     Let $\text{sig}^{-1}(x) \colon (0,1) \to \mathbb{R} \coloneqq \log \frac{x}{1-x}$ be the inverse function of $\text{sig}(\cdot)$. The utility of action $a^j$ is then estimated as

$$\widetilde{u}(a^j) = \begin{cases} \text{Proj}_{[-1,1]} \left( \tau \text{sig}^{-1} \left( \frac{1}{|\mathcal{T}_j|} \cdot \sum_{s \in \mathcal{T}_j} \left( \frac{n_{j,1}^{(s)}}{n_{j,1}^{(s)} + n_{j,2}^{(s)}} \right) \right) \right) & |\mathcal{T}_j| > 0 \\ 0 & |\mathcal{T}_j| = 0. \end{cases}$$

8: **end for**
9: Return $\widetilde{\boldsymbol{u}} = \left( \widetilde{u}(a^1), \widetilde{u}(a^2), \ldots, \widetilde{u}\left(a^{|\mathcal{A}|-1}\right), 0 \right)$

---

### E.1 PAIR-WISE UTILITY ESTIMATION

Lemma Lemma E.1 shows that, when the number of proposed actions is $K > 2$, for any two distinct actions $a \neq b \in o^{(t)}$, the expected fraction of $(a, b)$-pairs (*i.e.*, pairs formed by choosing one occurrence of $a$ and one occurrence of $b$ in the multiset $o^{(t)}$) for which $a$ is ranked ahead of $b$ in $\sigma^{(t)}$ equals

$$\text{sig}\left( \frac{u^{(t)}(a) - u^{(t)}(b)}{\tau} \right).$$

Equivalently, the induced pairwise marginal is the same as in the $K = 2$ case: it matches the probability of observing the permutation $(a, b)$ when only $a$ and $b$ are proposed. Related pairwise-marginal identities appear in Hunter (2004, p. 396); Lemma Lemma E.1 extends them to multisets (allowing repeated actions).

**Lemma E.1.** *Let $\#_{\mathcal{S}}(a) \coloneqq \sum_{a' \in \mathcal{S}} \mathbb{1}(a' = a)$ represent the number of elements in a multiset $\mathcal{S}$ that are equal to $a \in \mathcal{A}$. For any utility vector $\boldsymbol{u}$, temperature $\tau > 0$, a multiset of proposed actions $\mathcal{S}$ with cardinality $|\mathcal{S}| = K$, and any two actions $a \neq b \in \mathcal{S}$, we have*

$$\frac{1}{\#_{\mathcal{S}}(a) \cdot \#_{\mathcal{S}}(b)} \mathbb{E}_{\sigma} \left[ \sum_{k_1=1}^{K} \sum_{k_2=k_1+1}^{K} \mathbb{1}(\sigma(k_1) = a) \cdot \mathbb{1}(\sigma(k_2) = b) \,\middle|\, \mathcal{S} \right] = \text{sig}\left( \frac{u(a) - u(b)}{\tau} \right).$$

*The expectation is taken over the distribution of the permutation $\sigma$ under the ranking model* (PL).

The proof can be found in Appendix E.2. With Lemma E.1, the general cases where $K > 2$ actions are proposed can be cast into the case with only two actions being proposed, by enumerating all possible action pairs. Therefore, to estimate $u^{(t)}(a^j)$ for some $a^j \in \mathcal{A}$, we will first construct an unbiased estimator of $\text{sig}\left( \frac{u^{(s)}(a^j) - u^{(s)}(a^{|\mathcal{A}|})}{\tau} \right)$ using Lemma E.1, for all timesteps $s \in [t - m + 1, t]$ when both $a^j, a^{|\mathcal{A}|} \in o^{(s)}$. Since we have assumed without loss of generality that $u^{(s)}(a^{|\mathcal{A}|}) = 0$, these values coincide with $\text{sig}\left( \frac{u^{(s)}(a^j)}{\tau} \right)$. Then, by Hoeffding's inequality and the monotonicity of the logistic function $\text{sig}(\cdot)$, with high probability, the mean of the logistic function estimators will be

bounded between the minimum and maximum of $\left\{\text{sig}\left(\frac{u^{(s)}(a^j)}{\tau}\right)\right\}_{s=t-m+1}^{t}$. By Assumption 4.2, since the utility vectors are changing slowly, that mean can be shown close to $\text{sig}\left(\frac{u^{(t)}(a^j)}{\tau}\right)$. With a good estimate of $\text{sig}\left(\frac{u^{(t)}(a^j)}{\tau}\right)$, we can then take an inverse of $\text{sig}(\cdot)$ to estimate $u^{(t)}(a^j)$. This estimation algorithm is summarized in Algorithm 1 and analyzed in Theorem 5.1 below.

In the following, we will prove Theorem 5.1, which gives the estimation error bound of the utility vector for each timestep.

**Theorem 5.1.** *Consider* **InstUtil Rank** *and Algorithm 1. Suppose each action is proposed independently with probability at least $p > 0$ at each timestep $t \in [T]$ and let $\widetilde{\boldsymbol{u}}^{(t)} = \text{Estimate}\left(\{\sigma^{(s)}\}_{s=t-m'+1}^{t}\right)$. Then, for any $\delta \in (0,1)$ and $t \geq m'$, when $m'p^4 \geq 2\log\left(\frac{2}{\delta}\right)$, with probability at least $1 - \delta$, the estimate $\widetilde{\boldsymbol{u}}^{(t)}$ satisfies,*

$$\left\|\widetilde{\boldsymbol{u}}^{(t)} - \boldsymbol{u}^{(t)}\right\|_{\infty} \leq \frac{\tau\left(e^{\frac{1}{\tau}} + 1\right)^2}{p}\sqrt{\frac{\log\left(\frac{4|\mathcal{A}|}{\delta}\right)}{m'}} + \sum_{s=t-m'+1}^{t-1}\left\|\boldsymbol{u}^{(s+1)} - \boldsymbol{u}^{(s)}\right\|_{\infty}.$$

*Proof.* Due to the symmetry of timesteps, we will only prove Theorem 5.1 for $t = m'$ for notational simplicity.

For any $j \in [|\mathcal{A}| - 1]$, we assume that the probability for action $a^j$ being chosen at each timestep is at least $p$. Let $m_1$ denote the number of timesteps (out of $m'$) in which both $a^j$ and $a^{|\mathcal{A}|}$ occur in the proposed actions. By Hoeffding's Inequality, we have that with probability at least $1 - \frac{\delta}{2}$:

$$m_1 \geq \mathbb{E}[m_1] - \sqrt{\frac{m'}{2}\log\left(\frac{2}{\delta}\right)} \geq m'p^2 - \sqrt{\frac{m'}{2}\log\left(\frac{2}{\delta}\right)}.$$

We define

$$n_{j,1}^{(s)} := \sum_{i,k\in[K]} \mathbb{1}\left(\sigma^{(s)}(i) = a^j, \sigma^{(s)}(k) = a^{|\mathcal{A}|}\text{and } i < k\right),$$

$$n_{j,2}^{(s)} := \sum_{i,k\in[K]} \mathbb{1}\left(\sigma^{(s)}(i) = a^j, \sigma^{(s)}(k) = a^{|\mathcal{A}|}\text{and } i > k\right),$$

and the estimation of $u^{(m')}(a^j)$ as

$$\widetilde{u}^{(m')}(a^j) = \text{Proj}_{[-1,1]}\left(\tau\text{sig}^{-1}\left(\frac{1}{m_1}\sum_{s=1}^{T}\mathbb{1}\left(n_{j,1}^{(s)} + n_{j,2}^{(s)} > 0\right)\frac{n_{j,1}^{(s)}}{n_{j,1}^{(s)} + n_{j,2}^{(s)}}\right)\right),$$

Hence,

$$m_1 = \sum_{s=1}^{m'}\mathbb{1}\left(n_{j,1}^{(s)} + n_{j,2}^{(s)} > 0\right).$$

By Lemma E.1, since $u^{(s)}(a^{|\mathcal{A}|}) = 0$ for every $s \in \{1, 2, \ldots, m'\}$, and noting that $n_{j,1}^{(s)} + n_{j,2}^{(s)} = \#_{\sigma^{(s)}}(a^j) \cdot \#_{\sigma^{(s)}}(a^{|\mathcal{A}|})$, we have

$$\mathbb{E}_{\sigma^{(s)}}\left[\frac{n_{j,1}^{(s)}}{n_{j,1}^{(s)} + n_{j,2}^{(s)}}\right] = \text{sig}\left(\frac{u^{(s)}(a^j) - u^{(s)}(a^{|\mathcal{A}|})}{\tau}\right) = \text{sig}\left(\frac{1}{\tau}u^{(s)}(a^j)\right).$$

By Hoeffding's Inequality, we have that with probability at least $1 - \frac{\delta}{2|\mathcal{A}|}$,

$$\left|\frac{1}{m_1}\sum_{s=1}^{T}\mathbb{1}\left(n_{j,1}^{(s)} + n_{j,2}^{(s)} > 0\right)\left(\frac{n_{j,1}^{(s)}}{n_{j,1}^{(s)} + n_{j,2}^{(s)}} - \text{sig}\left(\frac{1}{\tau}u^{(s)}(a^j)\right)\right)\right| \leq \sqrt{\frac{1}{2m_1}\log\left(\frac{4|\mathcal{A}|}{\delta}\right)}.$$

Let $u^{(m'),*}(a^j) \in [-1, 1]$ be the scalar satisfying

$$\mathrm{sig}\left(\frac{1}{\tau}u^{(m'),*}(a^j)\right) = \frac{1}{m_1}\sum_{s=1}^{m'}\mathbb{1}\left(n_{j,1}^{(s)} + n_{j,2}^{(s)} > 0\right) \cdot \mathrm{sig}\left(\frac{1}{\tau}u^{(s)}(a^j)\right).$$

Since the logistic function is monotone and continuous, $u^{(m'),*}(a^j)$ is unique and must exist. Then, with probability at least $1 - \frac{\delta}{2|\mathcal{A}|}$,

$$
\begin{aligned}
&\left|\mathrm{sig}\left(\frac{1}{\tau}\widetilde{u}^{(m')}(a^j)\right) - \mathrm{sig}\left(\frac{1}{\tau}u^{(m'),*}(a^j)\right)\right| \\
&\overset{(i)}{\leq} \left|\mathrm{sig}\left(\mathrm{sig}^{-1}\left(\frac{1}{m_1}\sum_{s=1}^{T}\mathbb{1}\left(n_{j,1}^{(s)} + n_{j,2}^{(s)} > 0\right)\frac{n_{j,1}^{(s)}}{n_{j,1}^{(s)} + n_{j,2}^{(s)}}\right)\right) - \mathrm{sig}\left(\frac{1}{\tau}u^{(m'),*}(a^j)\right)\right| \\
&= \left|\frac{1}{m_1}\sum_{s=1}^{T}\mathbb{1}\left(n_{j,1}^{(s)} + n_{j,2}^{(s)} > 0\right)\frac{n_{j,1}^{(s)}}{n_{j,1}^{(s)} + n_{j,2}^{(s)}} - \mathrm{sig}\left(\frac{1}{\tau}u^{(m'),*}(a^j)\right)\right| \\
&\leq \sqrt{\frac{1}{2m_1}\log\left(\frac{4|\mathcal{A}|}{\delta}\right)}.
\end{aligned}
$$

$(i)$ uses the monotonicity of sig and the property of projection.

For any $u \in [-1, 1]$, we have

$$
\begin{aligned}
\frac{\mathrm{dsig}\left(\frac{u}{\tau}\right)}{\mathrm{d}u} &= \frac{1}{\tau}\mathrm{sig}\left(\frac{u}{\tau}\right)\left(1 - \mathrm{sig}\left(-\frac{u}{\tau}\right)\right) \\
&\geq \frac{1}{\tau}\mathrm{sig}\left(-\frac{1}{\tau}\right)\left(1 - \mathrm{sig}\left(\frac{1}{\tau}\right)\right) \\
&= \frac{1}{\tau}\left(\mathrm{sig}\left(-\frac{1}{\tau}\right)\right)^2 = \frac{1}{\tau\left(e^{\frac{1}{\tau}} + 1\right)^2}.
\end{aligned}
$$

Since sig is monotonic, by Taylor expansion and the fact that $\widetilde{u}^{(m')}(a^j), u^{(m'),*}(a^j) \in [-1, 1]$, we get that with probability at least $1 - \frac{\delta}{2|\mathcal{A}|}$,

$$\left|\widetilde{u}^{(m')}(a^j) - u^{(m'),*}(a^j)\right| \leq \tau\left(e^{\frac{1}{\tau}} + 1\right)^2\sqrt{\frac{1}{2m_1}\log\left(\frac{4|\mathcal{A}|}{\delta}\right)}.$$

Lemma E.2 in the following shows that $u^{(m'),*}(a^j)$ is bounded between the minimum and maximum of $\left\{u^{(s)}(a^j)\right\}_{s=1}^{m'}$. Then, by further utilizing the assumption that the variation of the utility vectors is small, we can bound the distance between $\widetilde{\boldsymbol{u}}^{(m')}$ and $\boldsymbol{u}^{(m')}$.

**Lemma E.2.** *Let $x_1, \ldots, x_n \in [-1, 1]$ and $\mathrm{sig}_{\mathrm{avg}} := \frac{1}{n}\sum_{i=1}^{n}\mathrm{sig}(x_i)$, we have*

$$\min_{i \in [n]} x_i \leq \mathrm{sig}^{-1}\left(\mathrm{sig}_{\mathrm{avg}}\right) \leq \max_{i \in [n]} x_i.$$

The proof is postponed to Appendix E.2. Hence,

$$u^{(m'),*}(a^j) \in \left[\min\left\{u^{(s)}(a^j)\right\}_{s=1}^{m'}, \max\left\{u^{(s)}(a^j)\right\}_{s=1}^{m'}\right].$$

Finally,

$$
\begin{aligned}
\left|\widetilde{u}^{(m')}(a^j) - u^{(m')}(a^j)\right| &\leq \tau \left(e^{\frac{1}{\tau}} + 1\right)^2 \sqrt{\frac{1}{2m_1} \log\left(\frac{4|\mathcal{A}|}{\delta}\right)} + \left|u^{(m'),*}(a^j) - u^{(m')}(a^j)\right| \\
&\leq \tau \left(e^{\frac{1}{\tau}} + 1\right)^2 \sqrt{\frac{1}{2m_1} \log\left(\frac{4|\mathcal{A}|}{\delta}\right)} + \max_{s \in \{1,2,\dots,m'-1\}} \left|u^{(s)}(a^j) - u^{(m')}(a^j)\right| \\
&\leq \tau \left(e^{\frac{1}{\tau}} + 1\right)^2 \sqrt{\frac{1}{2m_1} \log\left(\frac{4|\mathcal{A}|}{\delta}\right)} + \sum_{s=1}^{m'-1} \left|u^{(s+1)}(a^j) - u^{(s)}(a^j)\right|.
\end{aligned}
$$

When $m'p^4 \geq 2\log\left(\frac{2}{\delta}\right)$, with probability at least $1 - \frac{\delta}{2}$,

$$
m_1 \geq \frac{m'}{2}p^2.
$$

By union bound, with a probability at least $1 - (\frac{\delta}{2} + \frac{\delta}{2|\mathcal{A}|})$, we have

$$
\left|\widetilde{u}^{(m')}(a^j) - u^{(m')}(a^j)\right| \leq \tau \left(e^{\frac{1}{\tau}} + 1\right)^2 \sqrt{\frac{1}{m'p^2} \log\left(\frac{4|\mathcal{A}|}{\delta}\right)} + \sum_{s=1}^{m'-1} \left|u^{(s+1)}(a^j) - u^{(s)}(a^j)\right|.
$$

Since $a^j$ can be any action other than $a^{|\mathcal{A}|}$, by applying union bound, the following holds with probability at least $1 - \delta$,

$$
\left\|\widetilde{\boldsymbol{u}}^{(m')} - \boldsymbol{u}^{(m')}\right\|_\infty \leq \frac{\tau \left(e^{\frac{1}{\tau}} + 1\right)^2}{p} \sqrt{\frac{\log\left(\frac{4|\mathcal{A}|}{\delta}\right)}{m'}} + \sum_{s=1}^{m'-1} \left\|\boldsymbol{u}^{(s+1)} - \boldsymbol{u}^{(s)}\right\|_\infty. \qquad \square
$$

**Remark E.3.** *Due to the monotonicity of the logistic function* sig*, the following two projections on* $\text{sig}\left(\frac{x}{\tau}\right)$ *are equivalent:*

$$
\text{Proj}_{[\text{sig}(-\frac{1}{\tau}), \text{sig}(\frac{1}{\tau})]}\left(\text{sig}\left(\frac{x}{\tau}\right)\right) := \min\left(\max\left(\text{sig}\left(\frac{x}{\tau}\right), \text{sig}\left(-\frac{1}{\tau}\right)\right), \text{sig}\left(\frac{1}{\tau}\right)\right),
$$

$$
\text{sig}\left(\frac{\text{Proj}_{[-1,1]}(x)}{\tau}\right) := \text{sig}\left(\frac{\min\left(\max\left(x, -1\right), 1\right)}{\tau}\right).
$$

## E.2 OMITTED PROOFS

**Lemma E.1.** *Let* $\#_{\mathcal{S}}(a) := \sum_{a' \in \mathcal{S}} \mathbb{1}(a' = a)$ *represent the number of elements in a multiset* $\mathcal{S}$ *that are equal to* $a \in \mathcal{A}$. *For any utility vector* $\boldsymbol{u}$*, temperature* $\tau > 0$*, a multiset of proposed actions* $\mathcal{S}$ *with cardinality* $|\mathcal{S}| = K$*, and any two actions* $a \neq b \in \mathcal{S}$*, we have*

$$
\frac{1}{\#_{\mathcal{S}}(a) \cdot \#_{\mathcal{S}}(b)} \mathbb{E}_\sigma \left[\sum_{k_1=1}^{K} \sum_{k_2=k_1+1}^{K} \mathbb{1}(\sigma(k_1) = a) \cdot \mathbb{1}(\sigma(k_2) = b) \,\middle|\, \mathcal{S}\right] = \text{sig}\left(\frac{u(a) - u(b)}{\tau}\right).
$$

*The expectation is taken over the distribution of the permutation* $\sigma$ *under the ranking model* (PL).

*Proof.* We will abuse the notion $\overset{\sigma}{>}$ from permutations to subsets of actions. When proposing a set of actions $\mathcal{S}$, let $a \overset{\mathcal{S}}{>} b$ denote the event that $a$ is ahead of $b$ in the permutation given by the environment. In PL model, the probability that action $a$ ranks before action $b$ is that

$$
\mathbb{P}\left(a \overset{\{a,b\}}{<} b \,\middle|\, \tau, \boldsymbol{u}\right) = \frac{\exp\left(\frac{1}{\tau} u(a)\right)}{\exp\left(\frac{1}{\tau} u(a)\right) + \exp\left(\frac{1}{\tau} u(b)\right)}.
$$

By definition, let the multiset of the $K$ proposed actions be $\mathcal{S}$. Then, the probability of the K-wise permutation is

$$\mathbb{P}\left(\sigma \mid \mathcal{S}, \tau, \boldsymbol{u}\right) = \prod_{k_1=1}^{K} \frac{\exp\left(\frac{1}{\tau} u\left(\sigma\left(k_1\right)\right)\right)}{\sum_{k_2=k_1}^{K} \exp\left(\frac{1}{\tau} u\left(\sigma\left(k_2\right)\right)\right)}. \tag{E.1}$$

Recall that $\Sigma(\mathcal{S})$ denotes the set that contains all the permutations of the elements in $\mathcal{S}$. Hence, we have

$$\mathbb{E}\left[\sum_{k_1=1}^{K} \sum_{k_2=k_1+1}^{K} \mathbb{1}\left(\sigma(k_1) = a\right) \cdot \mathbb{1}\left(\sigma(k_2) = b\right) \,\middle|\, \mathcal{S}\right]$$

$$= \sum_{\sigma \in \Sigma(\mathcal{S})} \mathbb{P}\left(\sigma \mid \mathcal{S}, \tau, \boldsymbol{u}\right) \sum_{k_1=1}^{K} \sum_{k_2=k_1+1}^{K} \mathbb{1}\left(\sigma(k_1) = a\right) \cdot \mathbb{1}\left(\sigma(k_2) = b\right)$$

$$= \sum_{\substack{\sigma \in \Sigma(\mathcal{S}): \\ \sigma(1)=a}} \mathbb{P}\left(\sigma \mid \mathcal{S}, \tau, \boldsymbol{u}\right) \sum_{k_1=1}^{K} \sum_{k_2=k_1+1}^{K} \mathbb{1}\left(\sigma(k_1) = a\right) \cdot \mathbb{1}\left(\sigma(k_2) = b\right) \tag{E.2}$$

$$+ \sum_{\substack{\sigma \in \Sigma(\mathcal{S}): \\ \sigma(1)=b}} \mathbb{P}\left(\sigma \mid \mathcal{S}, \tau, \boldsymbol{u}\right) \sum_{k_1=1}^{K} \sum_{k_2=k_1+1}^{K} \mathbb{1}\left(\sigma(k_1) = a\right) \cdot \mathbb{1}\left(\sigma(k_2) = b\right) \tag{E.3}$$

$$+ \sum_{\substack{\sigma \in \Sigma(\mathcal{S}): \\ \sigma(1) \notin \{a,b\}}} \mathbb{P}\left(\sigma \mid \mathcal{S}, \tau, \boldsymbol{u}\right) \sum_{k_1=1}^{K} \sum_{k_2=k_1+1}^{K} \mathbb{1}\left(\sigma(k_1) = a\right) \cdot \mathbb{1}\left(\sigma(k_2) = b\right). \tag{E.4}$$

We deal with (E.2) first:

$$\sum_{\substack{\sigma \in \Sigma(\mathcal{S}): \\ \sigma(1)=a}} \mathbb{P}\left(\sigma \mid \mathcal{S}, \tau, \boldsymbol{u}\right) \sum_{k_1=1}^{K} \sum_{k_2=k_1+1}^{K} \mathbb{1}\left(\sigma(k_1) = a\right) \cdot \mathbb{1}\left(\sigma(k_2) = b\right)$$

$$= \sum_{\substack{\sigma \in \Sigma(\mathcal{S}): \\ \sigma(1)=a}} \mathbb{P}\left(\sigma \mid \mathcal{S}, \tau, \boldsymbol{u}\right) \left(\#_{\mathcal{S}}(b) + \sum_{k_1=2}^{K} \sum_{k_2=k_1+1}^{K} \mathbb{1}\left(\sigma(k_1) = a\right) \cdot \mathbb{1}\left(\sigma(k_2) = b\right)\right)$$

$$= \#_{\mathcal{S}}(b)\mathbb{P}\left(\sigma(1) = a \mid \mathcal{S}, \tau, \boldsymbol{u}\right) + \sum_{\substack{\sigma \in \Sigma(\mathcal{S}): \\ \sigma(1)=a}} \mathbb{P}\left(\sigma \mid \mathcal{S}, \tau, \boldsymbol{u}\right) \sum_{k_1=2}^{K} \sum_{k_2=k_1+1}^{K} \mathbb{1}\left(\sigma(k_1) = a\right) \cdot \mathbb{1}\left(\sigma(k_2) = b\right)$$

$$= \#_{\mathcal{S}}(b)\mathbb{P}\left(\sigma(1) = a \mid \mathcal{S}, \tau, \boldsymbol{u}\right)$$

$$+ \mathbb{P}\left(\sigma(1) = a \mid \mathcal{S}, \tau, \boldsymbol{u}\right) \sum_{\sigma \in \Sigma(\mathcal{S}\setminus\{a\})} \mathbb{P}\left(\sigma \mid \mathcal{S} \setminus \{a\}, \tau, \boldsymbol{u}\right) \sum_{k_1=2}^{K} \sum_{k_2=k_1+1}^{K} \mathbb{1}\left(\sigma(k_1) = a\right) \cdot \mathbb{1}\left(\sigma(k_2) = b\right)$$

$$= \#_{\mathcal{S}}(b)\mathbb{P}\left(\sigma(1) = a \mid \mathcal{S}, \tau, \boldsymbol{u}\right) + \mathbb{P}\left(\sigma(1) = a \mid \mathcal{S}, \tau, \boldsymbol{u}\right) \mathbb{E}\left[\sum_{k_1=1}^{K-1} \sum_{k_2=k_1+1}^{K-1} \mathbb{1}\left(\sigma(k_1) = a\right) \cdot \mathbb{1}\left(\sigma(k_2) = b\right) \mid \mathcal{S} \setminus \{a\}\right].$$

Similarly, for (E.3), we have

$$\sum_{\substack{\sigma \in \Sigma(\mathcal{S}): \\ \sigma(1)=b}} \mathbb{P}\left(\sigma \mid \mathcal{S}, \tau, \boldsymbol{u}\right) \sum_{k_1=1}^{K} \sum_{k_2=k_1+1}^{K} \mathbb{1}\left(\sigma(k_1) = a\right) \cdot \mathbb{1}\left(\sigma(k_2) = b\right)$$

$$= \mathbb{P}\left(\sigma(1) = b \mid \mathcal{S}, \tau, \boldsymbol{u}\right) \mathbb{E}\left[\sum_{k_1=1}^{K-1} \sum_{k_2=k_1+1}^{K-1} \mathbb{1}\left(\sigma(k_1) = a\right) \cdot \mathbb{1}\left(\sigma(k_2) = b\right) \mid \mathcal{S} \setminus \{b\}\right].$$

Let Unique $(\mathcal{S})$ be the set of non-repeated elements in $\mathcal{S}$. Then, (E.4) can be written as,

$$\sum_{\substack{\sigma \in \Sigma(\mathcal{S}): \\ \sigma(1) \notin \{a,b\}}} \mathbb{P}\left(\sigma \mid \mathcal{S}, \tau, \boldsymbol{u}\right) \sum_{k_1=1}^{K} \sum_{k_2=k_1+1}^{K} \mathbb{1}\left(\sigma(k_1) = a\right) \cdot \mathbb{1}\left(\sigma(k_2) = b\right)$$

$$= \sum_{\substack{c \in \text{Unique}(\mathcal{S}): \\ c \notin \{a,b\}}} \mathbb{P}\left(\sigma(1) = c \mid \mathcal{S}, \tau, \boldsymbol{u}\right) \mathbb{E}\left[\sum_{k_1=1}^{K-1} \sum_{k_2=k_1+1}^{K-1} \mathbb{1}\left(\sigma(k_1) = a\right) \cdot \mathbb{1}\left(\sigma(k_2) = b\right) \mid \mathcal{S} \setminus \{c\}\right].$$

Next, we will use induction to show that for any actions $a \neq b \in \mathcal{S}$, the following holds:

$$\mathbb{E}\left[\sum_{k_1=1}^{K} \sum_{k_2=k_1+1}^{K} \mathbb{1}\left(\sigma(k_1) = a\right) \cdot \mathbb{1}\left(\sigma(k_2) = b\right) \mid \mathcal{S}\right] = \#_{\mathcal{S}}(a) \#_{\mathcal{S}}(b) \text{sig}\left(\frac{u(a) - u(b)}{\tau}\right). \quad \text{(E.5)}$$

**Base case.** When $\#_{\mathcal{S}}(a) = 0$ or $\#_{\mathcal{S}}(b) = 0$, (E.5) trivially holds. When $|\mathcal{S}| = 2$ and $\#_{\mathcal{S}}(a) = \#_{\mathcal{S}}(b) = 1$,

$$\mathbb{E}\left[\sum_{k_1=1}^{K} \sum_{k_2=k_1+1}^{K} \mathbb{1}\left(\sigma(k_1) = a\right) \cdot \mathbb{1}\left(\sigma(k_2) = b\right) \mid \mathcal{S}\right] = \mathbb{P}\left(\sigma = ((a,b)) \mid \{a,b\}, \tau, \boldsymbol{u}\right)$$

$$= \text{sig}\left(\frac{u(a) - u(b)}{\tau}\right)$$

$$= \#_{\{a,b\}}(a) \#_{\{a,b\}}(b) \text{sig}\left(\frac{u(a) - u(b)}{\tau}\right).$$

**Lemma E.4.** *For any utility vector $\boldsymbol{u}$, temperature $\tau > 0$, and a multiset of actions $\mathcal{S}$, the marginal probability of any action $a \in \mathcal{A}$ ranking at the first place of the permutation can be written as*

$$\mathbb{P}\left(\sigma(1) = a \mid \mathcal{S}, \tau, \boldsymbol{u}\right) = \#_{\mathcal{S}}(a) \frac{\exp\left(\frac{1}{\tau} u(a)\right)}{\sum_{a' \in \mathcal{S}} \exp\left(\frac{1}{\tau} u(a')\right)}.$$

The proof is presented later in this section.

**Induction step.** When (E.5) holds for any $\mathcal{S}$ with $|\mathcal{S}| = K - 1$. Then, we will show that it still holds for any $\mathcal{S}$ with $|\mathcal{S}| = K$. By Lemma E.4, (E.2) is equal to

$$\#_{\mathcal{S}}(b) \mathbb{P}\left(\sigma(1) = a \mid \mathcal{S}, \tau, \boldsymbol{u}\right)$$

$$+ \mathbb{P}\left(\sigma(1) = a \mid \mathcal{S}, \tau, \boldsymbol{u}\right) \sum_{\sigma \in \Sigma(\mathcal{S} \setminus \{a\})} \mathbb{P}\left(\sigma \mid \mathcal{S} \setminus \{a\}, \tau, \boldsymbol{u}\right) \sum_{k_1=2}^{K} \sum_{k_2=k_1+1}^{K} \mathbb{1}\left(\sigma(k_1) = a\right) \cdot \mathbb{1}\left(\sigma(k_2) = b\right)$$

$$= \#_{\mathcal{S}}(a) \#_{\mathcal{S}}(b) \frac{\exp\left(\frac{1}{\tau} u(a)\right)}{\sum_{a' \in \mathcal{S}} \exp\left(\frac{1}{\tau} u(a')\right)}$$

$$+ \#_{\mathcal{S}}(a) \frac{\exp\left(\frac{1}{\tau} u(a)\right)}{\sum_{a' \in \mathcal{S}} \exp\left(\frac{1}{\tau} u(a')\right)} \left(\#_{\mathcal{S} \setminus \{a\}}(a) \cdot \#_{\mathcal{S} \setminus \{a\}}(b) \text{sig}\left(\frac{u(a) - u(b)}{\tau}\right)\right)$$

$$= \#_{\mathcal{S}}(a) \#_{\mathcal{S}}(b) \frac{\exp\left(\frac{1}{\tau} u(a)\right)}{\sum_{a' \in \mathcal{S}} \exp\left(\frac{1}{\tau} u(a')\right)} + \#_{\mathcal{S}}(a) \frac{\exp\left(\frac{1}{\tau} u(a)\right)}{\sum_{a' \in \mathcal{S}} \exp\left(\frac{1}{\tau} u(a')\right)} \left(\#_{\mathcal{S}}(a) - 1\right) \cdot \#_{\mathcal{S}}(b) \text{sig}\left(\frac{u(a) - u(b)}{\tau}\right).$$

Similarly, (E.3) is equal to

$$\#_{\mathcal{S}}(b) \frac{\exp\left(\frac{1}{\tau} u(b)\right)}{\sum_{a' \in \mathcal{S}} \exp\left(\frac{1}{\tau} u(a')\right)} \#_{\mathcal{S}}(a) \cdot \left(\#_{\mathcal{S}}(b) - 1\right) \text{sig}\left(\frac{u(a) - u(b)}{\tau}\right),$$

and (E.4) is equal to

$$\left(1 - \frac{\#_{\mathcal{S}}(a) \exp\left(\frac{1}{\tau} u(a)\right) + \#_{\mathcal{S}}(b) \exp\left(\frac{1}{\tau} u(b)\right)}{\sum_{a' \in \mathcal{S}} \exp\left(\frac{1}{\tau} u(a')\right)}\right) \#_{\mathcal{S}}(a) \cdot \#_{\mathcal{S}}(b) \text{sig}\left(\frac{u(a) - u(b)}{\tau}\right).$$

Lastly, by summing them up, we have

$$\mathbb{E}\left[\sum_{k_1=1}^{K}\sum_{k_2=k_1+1}^{K}\mathbb{1}\left(\sigma(k_1)=a\right)\cdot\mathbb{1}\left(\sigma(k_2)=b\right)\mid\mathcal{S}\right]$$

$$=\#_{\mathcal{S}}(a)\#_{\mathcal{S}}(b)\frac{\exp\left(\frac{1}{\tau}u\left(a\right)\right)}{\sum_{a'\in\mathcal{S}}\exp\left(\frac{1}{\tau}u\left(a'\right)\right)}-\#_{\mathcal{S}}(a)\frac{\exp\left(\frac{1}{\tau}u\left(a\right)\right)+\exp\left(\frac{1}{\tau}u\left(b\right)\right)}{\sum_{a'\in\mathcal{S}}\exp\left(\frac{1}{\tau}u\left(a'\right)\right)}\cdot\#_{\mathcal{S}}(b)\mathrm{sig}\left(\frac{u(a)-u(b)}{\tau}\right)$$

$$+\#_{\mathcal{S}}(a)\cdot\#_{\mathcal{S}}(b)\mathrm{sig}\left(\frac{u(a)-u(b)}{\tau}\right).$$

Note that $\mathrm{sig}\left(\frac{u(a)-u(b)}{\tau}\right)=\frac{\exp\left(\frac{u(a)-u(b)}{\tau}\right)}{\exp\left(\frac{u(a)-u(b)}{\tau}\right)+1}=\frac{\exp\left(\frac{u(a)}{\tau}\right)}{\exp\left(\frac{u(a)}{\tau}\right)+\exp\left(\frac{u(b)}{\tau}\right)}$. Therefore,

$$\mathbb{E}\left[\sum_{k_1=1}^{K}\sum_{k_2=k_1+1}^{K}\mathbb{1}\left(\sigma(k_1)=a\right)\cdot\mathbb{1}\left(\sigma(k_2)=b\right)\,\middle|\,\mathcal{S}\right]=\#_{\mathcal{S}}(a)\cdot\#_{\mathcal{S}}(b)\mathrm{sig}\left(\frac{u(a)-u(b)}{\tau}\right),$$

and we complete the induction. $\qquad\square$

**Lemma E.4.** *For any utility vector $\boldsymbol{u}$, temperature $\tau>0$, and a multiset of actions $\mathcal{S}$, the marginal probability of any action $a\in\mathcal{A}$ ranking at the first place of the permutation can be written as*

$$\mathbb{P}\left(\sigma(1)=a\mid\mathcal{S},\tau,\boldsymbol{u}\right)=\#_{\mathcal{S}}\left(a\right)\frac{\exp\left(\frac{1}{\tau}u\left(a\right)\right)}{\sum_{a'\in\mathcal{S}}\exp\left(\frac{1}{\tau}u\left(a'\right)\right)}.$$

*Proof.* Let $\Sigma\left(\mathcal{S}\right)$ be the set containing all permutations of $\mathcal{S}$. By definition, for any action $a\in\mathcal{A}$,

$$\mathbb{P}\left(\sigma(1)=a\mid\mathcal{S},\tau,\boldsymbol{u}\right)=\sum_{\substack{\sigma\in\Sigma(\mathcal{S}):\\\sigma(1)=a}}\mathbb{P}\left(\sigma\mid\mathcal{S},\tau,\boldsymbol{u}\right)=\sum_{\substack{\sigma\in\Sigma(\mathcal{S}):\\\sigma(1)=a}}\prod_{k_1=1}^{|\mathcal{S}|}\frac{\exp\left(\frac{1}{\tau}u\left(\sigma\left(k_1\right)\right)\right)}{\sum_{k_2=k_1}^{|\mathcal{S}|}\exp\left(\frac{1}{\tau}u\left(\sigma\left(k_2\right)\right)\right)}.$$

Since there are $\#_{\mathcal{S}}(a)$ action $a$ in $\mathcal{S}$, by rearranging the terms, we have

$$\mathbb{P}\left(\sigma(1)=a\mid\mathcal{S},\tau,\boldsymbol{u}\right)=\#_{\mathcal{S}}(a)\frac{\exp\left(\frac{1}{\tau}u\left(a\right)\right)}{\sum_{a'\in\mathcal{S}}\exp\left(\frac{1}{\tau}u\left(a'\right)\right)}\sum_{\sigma\in\Sigma(\mathcal{S}\backslash\{a\})}\prod_{k_1=1}^{|\mathcal{S}|-1}\frac{\exp\left(\frac{1}{\tau}u\left(\sigma\left(k_1\right)\right)\right)}{\sum_{k_2=k_1}^{|\mathcal{S}|-1}\exp\left(\frac{1}{\tau}u\left(\sigma\left(k_2\right)\right)\right)}$$

$$=\#_{\mathcal{S}}(a)\frac{\exp\left(\frac{1}{\tau}u\left(a\right)\right)}{\sum_{a'\in\mathcal{S}}\exp\left(\frac{1}{\tau}u\left(a'\right)\right)}\sum_{\sigma\in\Sigma(\mathcal{S}\backslash\{a\})}\mathbb{P}\left(\sigma\mid\mathcal{S}\backslash\{a\},\tau,\boldsymbol{u}\right)$$

$$=\#_{\mathcal{S}}(a)\frac{\exp\left(\frac{1}{\tau}u\left(a\right)\right)}{\sum_{a'\in\mathcal{S}}\exp\left(\frac{1}{\tau}u\left(a'\right)\right)}.\qquad\square$$

**Lemma E.2.** *Let $x_1,\ldots,x_n\in[-1,1]$ and $\mathrm{sig}_{\mathrm{avg}}:=\frac{1}{n}\sum_{i=1}^{n}\mathrm{sig}(x_i)$, we have*

$$\min_{i\in[n]}x_i\leq\mathrm{sig}^{-1}\left(\mathrm{sig}_{\mathrm{avg}}\right)\leq\max_{i\in[n]}x_i.$$

*Proof.* The logistic function $\mathrm{sig}(x)$ is increasing monotonically with respect to $x$, since $\frac{\mathrm{dsig}}{\mathrm{d}x}=\frac{\exp(x)}{(\exp(x)+1)^2}>0$. Then, without loss of generality, let $x_1\leq x_2\leq\cdots\leq x_n$. Thus, $\mathrm{sig}(x_1)\leq\mathrm{sig}_{\mathrm{avg}}\leq\mathrm{sig}(x_n)$.

Since $\mathrm{sig}(x)$ is monotonic and continuous, there exists only one $\zeta\in[x_1,x_n]$ such that $\mathrm{sig}(\zeta)=\mathrm{sig}_{\mathrm{avg}}$. $\qquad\square$

## F   ALGORITHMS AND DIAGRAMS

In this section, we present the algorithms' diagrams and pseudo-code of learning with **InstUtil Rank** and **AvgUtil Rank** individually.

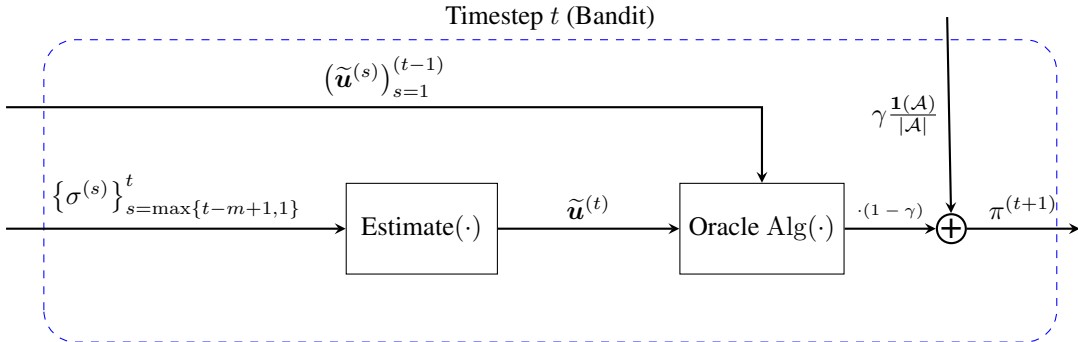

Figure 10: The diagram of Algorithm 2 with **InstUtil Rank** under full-information feedback (top) and bandit feedback (bottom). $\oplus$ represents the addition of $(1 - \gamma)$ times the output the Alg and $\gamma$ times a uniform distribution over $\mathcal{A}$.

### F.1 THE ALGORITHM AND DIAGRAM FOR **InstUtil Rank**

We present the diagram and the algorithm pseudo-code of learning with **InstUtil Rank**: Figure 10 and Algorithm 2.

---

**Algorithm 2** Online Learning with **InstUtil Rank** Feedback
___

1: **Input:** Action space $\mathcal{A}$, any full-information no-regret learning algorithm Alg with numeric utility feedback, selected action number $K$, estimation window size $m$, and exploration rate $\gamma$.
2: Initialize $\pi^{(1)}$ as uniform distribution $\frac{1}{|\mathcal{A}|}$ over $\mathcal{A}$
3: **for** timestep $t = 1, 2, \ldots, T$ **do**
4:     **if** Full-information setting **then**
5:         $K = |\mathcal{A}|$ in this case. Select all $|\mathcal{A}|$ actions.
6:     **else if** Bandit setting **then**
7:         Sample $K$ actions independently with replacement from $\pi^{(t)}$.
8:     **end if**
9:     Receive a ranking feedback $\sigma^{(t)} = \left( \sigma^{(t)}(1), \sigma^{(t)}(2), \ldots, \sigma^{(t)}(K) \right)$ from the environment.
10:     $\widetilde{\boldsymbol{u}}^{(t)} = \text{Estimate}\left( \left\{ \sigma^{(s)} \right\}_{s=\max\{t-m+1,1\}}^{t} \right)$ by calling Algorithm 1.
11:     **if** Full-information setting **then**
12:         $\pi^{(t+1)} \leftarrow \text{Alg}\left( \left( \widetilde{\boldsymbol{u}}^{(s)} \right)_{s=1}^{t} \right)$.
13:     **else if** Bandit setting **then**
14:         $\pi^{(t+1)} \leftarrow (1 - \gamma)\text{Alg}\left( \left( \widetilde{\boldsymbol{u}}^{(s)} \right)_{s=1}^{t} \right) + \gamma \frac{\mathbf{1}(\mathcal{A})}{|\mathcal{A}|}$.
15:     **end if**
16: **end for**

---

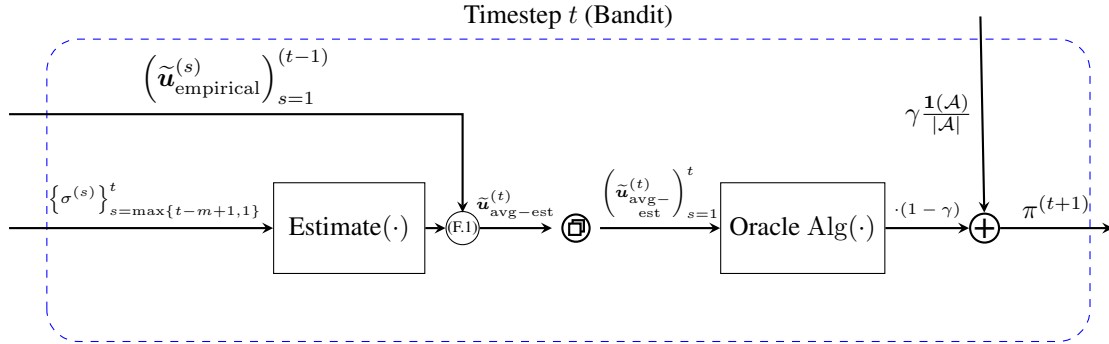

Figure 11: The diagram of Algorithm 3 with **AvgUtil Rank** under full-information feedback (top) and bandit feedback (bottom). ⊡ represents copying the estimated utility vector for $t$ times. ⊕ represents the addition of $(1 - \gamma)$ times the output the Alg and $\gamma$ times a uniform distribution over $\mathcal{A}$.

### F.2 THE ALGORITHM AND DIAGRAM FOR **AvgUtil Rank**

We present the diagram and the algorithm pseudo-code of learning with **AvgUtil Rank**: Figure 11 and Algorithm 3.

## G PROOF OF THEOREM 5.2 (FULL-INFORMATION)

In this section, we prove the regret upper bound under **InstUtil Rank** and full-information feedback.

**Theorem G.1** (Formal version of Theorem 5.2 (Full-Information)). *Consider Algorithm 2 and full-information feedback. For any $\delta \in (0, 1)$, $T > 0$, and any no-regret learning algorithm with numeric utility feedback,* Alg*, with probability at least $(1 - \delta)$, by choosing $m = \left(\frac{T}{P^{(T)}}\right)^{\frac{2}{3}} \left(\log\left(\frac{4|\mathcal{A}|T}{\delta}\right)\right)^{\frac{1}{3}}$, $R^{(T),\text{external}}$ satisfies*

$$
\begin{aligned}
R^{(T),\text{external}} \leq & R^{(T),\text{external}} \left( \text{Alg}, \left(\widetilde{\boldsymbol{u}}^{(t)}\right)_{t=1}^{T} \right) + 2\tau \left( e^{\frac{1}{\tau}} + 1 \right)^{2} \left( P^{(t)} \right)^{\frac{1}{3}} T^{\frac{2}{3}} \left( \log\left( \frac{4|\mathcal{A}|T}{\delta} \right) \right)^{\frac{1}{3}} \\
& + 2 \left( P^{(t)} \right)^{\frac{1}{3}} T^{\frac{2}{3}} \left( \log\left( \frac{4|\mathcal{A}|T}{\delta} \right) \right)^{\frac{1}{3}} + 4 \left( P^{(t)} \right)^{-\frac{2}{3}} T^{\frac{2}{3}} \left( \log\left( \frac{4|\mathcal{A}|T}{\delta} \right) \right)^{\frac{1}{3}}.
\end{aligned}
$$

$$(\text{G.1})$$

---

**Algorithm 3** Online Learning with **AvgUtil Rank** Feedback

---

1: **Input:** Action space $\mathcal{A}$, any full-information no-regret algorithm Alg under numeric feedback, selected action number $K$, estimation window size $m$, exploration rate $\gamma$, and block size $M$.
2: Initialize $\pi^{(1)}$ as uniform distribution $\frac{\mathbf{1}}{|\mathcal{A}|}$ over $\mathcal{A}$
3: **for** timestep $t = 1, 2, \ldots, T$ **do**
4:    **if** Full-information setting **then**
5:       $K = |\mathcal{A}|$ in this case. Select all $|\mathcal{A}|$ actions.
6:    **else if** Bandit setting **then**
7:       Sample $K$ actions independently with replacement from $\pi^{(t)}$.
8:    **end if**
9:    Receive a ranking feedback $\sigma^{(t)} = \left(\sigma^{(t)}(1), \sigma^{(t)}(2), \ldots, \sigma^{(t)}(K)\right)$ from the environment.
10:    **if** Full-information setting **then**
11:       $\widetilde{\boldsymbol{u}}_{\text{avg}}^{(t)} = \text{Estimate}\left(\left\{\sigma^{(s)}\right\}_{s=\max\{t-m+1,1\}}^{t}\right)$ by calling Algorithm 1.
12:       $\pi^{(t+1)} \leftarrow \text{Alg}\left(\left(\widetilde{\boldsymbol{u}}_{\text{avg}}^{(t)}\right)_{s=1}^{t}\right)$, *i.e.*, the strategy generated by Alg by setting all utility vectors from timestep 1 to $t$ as $\widetilde{\boldsymbol{u}}_{\text{avg}}^{(t)}$.
13:    **else if** Bandit setting **then**
14:       $\widetilde{\boldsymbol{u}}_{\text{empirical}}^{(t)} = \text{Estimate}\left(\left\{\sigma^{(s)}\right\}_{s=\max\{t-m+1,1\}}^{t}\right)$ by calling Algorithm 1.
15:       Let $n^{(t)}(a) := \sum_{s=1}^{t} \#_{o^{(s)}}(a)$ for any $a \in \mathcal{A}$ as the number of times action $a$ has been proposed up to timestep $t$. Then, the estimated average utility is

$$\widetilde{u}_{\text{avg}-\text{est}}^{(t)}(a) := \begin{cases} \frac{1}{\lfloor t/M \rfloor} \sum_{s=1}^{\lfloor t/M \rfloor} \frac{\widetilde{u}_{\text{empirical}}^{(s\cdot M)}(a) n^{(s\cdot M)}(a) - \widetilde{u}_{\text{empirical}}^{((s-1)M)}(a) n^{((s-1)M)}(a)}{n^{(s\cdot M)}(a) - n^{((s-1)M)}(a)} & t \geq M \\ 0 & t < M \end{cases}$$
$$\text{(F.1)}$$

      $\{$Let $\widetilde{u}_{\text{empirical}}^{(0)}(a) = n^{(0)}(a) = 0$ for any action $a \in \mathcal{A}\}$
16:       $\pi^{(t+1)} \leftarrow (1-\gamma)\text{Alg}\left(\left(\widetilde{\boldsymbol{u}}_{\text{avg}-\text{est}}^{(t)}\right)_{s=1}^{t}\right) + \gamma\frac{\mathbf{1}(\mathcal{A})}{|\mathcal{A}|}$.
17:    **end if**
18: **end for**

---

*Proof.* By Theorem 5.1, we have

$$\left| R^{(T),\text{external}} - R^{(T),\text{external}}\left(\text{Alg}, \left(\widetilde{\boldsymbol{u}}^{(t)}\right)_{t=1}^{T}\right)\right|$$

$$= \left| \max_{\widehat{\pi} \in \Delta^{\mathcal{A}}} \sum_{t=1}^{T} \left\langle \boldsymbol{u}^{(t)}, \widehat{\pi} - \pi^{(t)} \right\rangle - \max_{\widehat{\pi} \in \Delta^{\mathcal{A}}} \sum_{t=1}^{T} \left\langle \widetilde{\boldsymbol{u}}^{(t)}, \widehat{\pi} - \pi^{(t)} \right\rangle \right|$$

$$\leq \max_{\widehat{\pi} \in \Delta^{\mathcal{A}}} \left| \sum_{t=1}^{T} \left\langle \boldsymbol{u}^{(t)} - \widetilde{\boldsymbol{u}}^{(t)}, \widehat{\pi} - \pi^{(t)} \right\rangle \right|$$

$$\leq \sum_{t=1}^{T} \left\| \boldsymbol{u}^{(t)} - \widetilde{\boldsymbol{u}}^{(t)} \right\|_{\infty} \cdot \max_{\widehat{\pi} \in \Delta^{\mathcal{A}}} \left\| \widehat{\pi} - \pi^{(t)} \right\|_{1}.$$

When $t \geq m$, the estimation error between $\widetilde{\boldsymbol{u}}^{(t)}$ and $\boldsymbol{u}^{(t)}$ is given by Theorem 5.1 with $p = 1$ since we are considering full-information feedback. When $t < m$,

$$\sum_{t=1}^{m-1} \left\| \boldsymbol{u}^{(t)} - \widetilde{\boldsymbol{u}}^{(t)} \right\|_{\infty} \cdot \max_{\widehat{\pi} \in \Delta^{\mathcal{A}}} \left\| \widehat{\pi} - \pi^{(t)} \right\|_{1} \leq 4m.$$

For any given $\delta$, we require the utility estimation bound to hold with probability at least $1 - \frac{\delta}{T}$, then by union bound, with probability at least $1 - \delta$,

$$\left| R^{(T),\text{external}} - R^{(T),\text{external}} \left( \text{Alg}, \left( \widetilde{\boldsymbol{u}}^{(t)} \right)_{t=1}^T \right) \right|$$

$$\leq 2\tau \left( e^{\frac{1}{\tau}} + 1 \right)^2 \sqrt{\frac{\log \left( \frac{4|\mathcal{A}|T}{\delta} \right)}{m}} T + 2m \left( P^{(T)} + 2 \right).$$

By choosing $m = \left( \frac{T}{P^{(T)}} \right)^{\frac{2}{3}} \left( \log \left( \frac{4|\mathcal{A}|T}{\delta} \right) \right)^{\frac{1}{3}}$, we conclude the proof. $\qquad\square$

## H   PROOF OF THEOREM 5.2 (BANDIT)

In this section, we prove the regret upper bound under **InstUtil Rank** and bandit feedback.

**Theorem H.1** (Formal version of Theorem 5.2 (Bandit)). *Consider Algorithm 2 and bandit feedback. For any $\delta \in (0, 1)$, $T > 0$, and any no-regret learning algorithm with numeric utility feedback, Alg, with probability at least $(1-\delta)$, by choosing $\gamma = \left( \frac{P^{(T)}}{T} \right)^{\frac{1}{5}}$, $m = \frac{32|\mathcal{A}|^4}{K^4} \left( \frac{T}{P^{(T)}} \right)^{\frac{4}{5}} \log \left( \frac{8|\mathcal{A}|T}{\delta} \right)$, $R^{(T)}$ satisfies*

$$R^{(T)} \leq R^{(T),\text{external}} \left( \text{Alg}, \left( \widetilde{\boldsymbol{u}}^{(t)} \right)_{t=1}^T \right) + 2\sqrt{2T \log \left( \frac{2}{\delta} \right)} + \left( \frac{\tau K \left( e^{\frac{1}{\tau}} + 1 \right)^2}{|\mathcal{A}|} + 1 \right) \left( P^{(T)} \right)^{\frac{1}{5}} T^{\frac{4}{5}}$$

$$\tag{H.1}$$

$$+ \frac{64|\mathcal{A}|^4}{K^4} \left( P^{(T)} \right)^{\frac{1}{5}} T^{\frac{4}{5}} \log \left( \frac{8|\mathcal{A}|T}{\delta} \right) + \frac{128|\mathcal{A}|^4}{K^4} \left( \frac{T}{P^{(T)}} \right)^{\frac{4}{5}} \log \left( \frac{8|\mathcal{A}|T}{\delta} \right).$$

*Proof.* We define:

$$\widetilde{R}^{(T)} := \max_{\widehat{\pi} \in \Delta^{\mathcal{A}}} \sum_{t=1}^T \left\langle \widetilde{\boldsymbol{u}}^{(t)}, \widehat{\pi} - \pi^{(t)} \right\rangle.$$

Then,

$$R^{(T)} \leq \underbrace{R^{(T)} - R^{(T),\text{external}}}_{\heartsuit} + \underbrace{\left| R^{(T),\text{external}} - \widetilde{R}^{(T)} \right|}_{\spadesuit} + \underbrace{\left| \widetilde{R}^{(T)} - R^{(T),\text{external}} \left( \text{Alg}, \left( \widetilde{\boldsymbol{u}}^{(t)} \right)_{t=1}^T \right) \right|}_{\blacklozenge}$$

$$+ \underbrace{R^{(T),\text{external}} \left( \text{Alg}, \left( \widetilde{\boldsymbol{u}}^{(t)} \right)_{t=1}^T \right)}_{\clubsuit}.$$

Note that $\spadesuit$ can be bounded by bounding $\left\| \widetilde{\boldsymbol{u}}^{(t)} - \boldsymbol{u}^{(t)} \right\|_\infty$ as in Appendix G. $\clubsuit$ is sublinear by the definition of Alg. Next, we will introduce lemmas that individually bound $\heartsuit$, $\blacklozenge$. The proofs are postponed to Appendices H.1 and H.2.

**Lemma H.2** ($\heartsuit$). *For any $T > 0$ and $\delta \in (0, 1)$, with probability at least $1 - \delta$:*

$$R^{(T)} - R^{(T),\text{external}} \leq 2\sqrt{2T \log \left( \frac{1}{\delta} \right)}.$$

**Lemma H.3** ($\blacklozenge$). *The difference between $\widetilde{R}^{(T)}$ and $R^{(T),\text{external}} \left( \text{Alg}, \left( \widetilde{\boldsymbol{u}}^{(t)} \right)_{t=1}^T \right)$ satisfies:*

$$\left| \widetilde{R}^{(T)} - R^{(T),\text{external}} \left( \text{Alg}, \left( \widetilde{\boldsymbol{u}}^{(t)} \right)_{t=1}^T \right) \right| \leq 2\gamma T.$$

With Lemma H.2 and Lemma H.3, under the conditions in Theorem 5.2, by letting Theorem 5.1 hold with probability $1 - \frac{\delta}{2T}$ at each timestep and using the union bound, with probability at least $1 - \delta$, the regret satisfies

$$R^{(T)} \leq R^{(T),\text{external}} \left( \text{Alg}, \left( \widetilde{\boldsymbol{u}}^{(t)} \right)_{t=1}^{T} \right) + 2\sqrt{2T \log \left( \frac{2}{\delta} \right)}$$

$$+ 2\frac{\tau \left( e^{\frac{1}{\tau}} + 1 \right)^2}{p} \sqrt{\frac{\log \left( \frac{8|\mathcal{A}|T}{\delta} \right)}{m}} T + 2m \left( P^{(T)} + 2 \right) + \gamma T.$$

In this case, each action $a \in \mathcal{A}$ is chosen with probability at least $p$ that satisfies

$$p \geq 1 - \left( 1 - \frac{\gamma}{|\mathcal{A}|} \right)^K \geq 1 - \exp \left( -K\frac{\gamma}{|\mathcal{A}|} \right)$$

$$\geq 1 - \left( 1 - K\frac{\gamma}{|\mathcal{A}|} + \frac{1}{2} \left( K\frac{\gamma}{|\mathcal{A}|} \right)^2 \right) = K\frac{\gamma}{|\mathcal{A}|} - \frac{1}{2} \left( K\frac{\gamma}{|\mathcal{A}|} \right)^2.$$

Since $K\frac{\gamma}{|\mathcal{A}|} \leq 1$, we have

$$\frac{1}{2}K\frac{\gamma}{|\mathcal{A}|} \geq \frac{1}{2} \left( K\frac{\gamma}{|\mathcal{A}|} \right)^2 \quad \Rightarrow \quad p \geq \frac{K\gamma}{2|\mathcal{A}|}.$$

By letting $\gamma = \left( \frac{P^{(T)}}{T} \right)^{\frac{1}{5}}, m = \frac{32|\mathcal{A}|^4}{K^4} \left( \frac{T}{P^{(T)}} \right)^{\frac{4}{5}} \log \left( \frac{8|\mathcal{A}|T}{\delta} \right)$, we have

$$R^{(T)} \leq R^{(T),\text{external}} \left( \text{Alg}, \left( \widetilde{\boldsymbol{u}}^{(t)} \right)_{t=1}^{T} \right) + \mathcal{O} \left( T^{\frac{4}{5}} \left( P^{(T)} \right)^{\frac{1}{5}} \log \left( \frac{T}{\delta} \right) \right).$$

The condition $m \geq \frac{2\log\left( \frac{4T}{\delta} \right)}{p^4}$ is also satisfied since

$$mp^4 \geq m\frac{K^4\gamma^4}{16|\mathcal{A}|^4} = 2\log \left( \frac{8|\mathcal{A}|T}{\delta} \right) \geq 2\log \left( \frac{4T}{\delta} \right). \qquad \square$$

## H.1 BOUNDING ♥

We will show that $\left| R^{(T)} - R^{(T),\text{external}} \right|$ is sublinear by using a standard concentration bound.

**Lemma H.2 (♥).** *For any $T > 0$ and $\delta \in (0, 1)$, with probability at least $1 - \delta$:*

$$R^{(T)} - R^{(T),\text{external}} \leq 2\sqrt{2T \log \left( \frac{1}{\delta} \right)}.$$

*Proof.* Let

$$d^{(t)} := \frac{1}{K} \sum_{a \in o^{(t)}} u^{(t)}(a) - \left\langle \boldsymbol{u}^{(t)}, \pi^{(t)} \right\rangle.$$

By our algorithm design, each element of $o^{(t)}$ is sampled *i.i.d.* from $\pi^{(t)}$ with replacement and the update rule of $\pi^{(t)}$ is deterministic, $\mathbb{E} \left[ d^{(t)} \mid \{\sigma^{(s)}\}_{s=1}^{t-1}, \{\boldsymbol{u}^{(s)}\}_{s=1}^{t-1} \right] = 0$, so that $\{d^{(t)}\}$ is a martingale difference sequence.

Due to the bounds of $\left| \frac{1}{K} \sum_{a \in o^{(t)}} u^{(t)}(a) \right| \leq 1, \left| \left\langle \boldsymbol{u}^{(t)}, \pi^{(t)} \right\rangle \right| \leq 1$, we have

$$\left| d^{(t)} \right| = \left| \frac{1}{K} \sum_{a \in o^{(t)}} u^{(t)}(a) - \left\langle \boldsymbol{u}^{(t)}, \pi^{(t)} \right\rangle \right| \leq \left| \frac{1}{K} \sum_{a \in o^{(t)}} u^{(t)}(a) \right| + \left| \left\langle \boldsymbol{u}^{(t)}, \pi^{(t)} \right\rangle \right| \leq 2.$$

Furthermore, we have

$$\sum_{t=1}^{T} d^{(t)} = \sum_{t=1}^{T} \left( \frac{1}{K} \sum_{a \in o^{(t)}} u^{(t)}(a) \right) - \sum_{t=1}^{T} \left( \left\langle \boldsymbol{u}^{(t)}, \pi^{(t)} \right\rangle \right)$$

$$= \max_{\widehat{\pi} \in \Delta^{\mathcal{A}}} \sum_{t=1}^{T} \left( \left\langle \boldsymbol{u}^{(t)}, \widehat{\pi} \right\rangle - \left\langle \boldsymbol{u}^{(t)}, \pi^{(t)} \right\rangle \right) - \max_{\widehat{\pi} \in \Delta^{\mathcal{A}}} \sum_{t=1}^{T} \left( \left\langle \boldsymbol{u}^{(t)}, \widehat{\pi} \right\rangle - \frac{1}{K} \sum_{j=1}^{K} u^{(t)} \left( \sigma^{(j)} \right) \right)$$

$$= R^{(T),\text{external}} - R^{(T)}.$$

Next, we will introduce Azuma-Hoeffding inequality to finish the concentration bound.

**Theorem H.4** (Azuma-Hoeffding inequality). *For any martingale difference sequence $Y_1, \ldots, Y_n$ such that $\forall j \in [n], a_j \leq Y_j \leq b_j$, the following holds for any $w \geq 0$.*

$$\mathbb{P}\left( \sum_{j=1}^{n} Y_j \geq w \right) \leq \exp\left( -\frac{2w^2}{\sum_{j=1}^{n} (b_j - a_j)^2} \right).$$

Then by Theorem H.4, with probability at least $1 - \delta$

$$R^{(T)} \leq R^{(T),\text{external}} + 2\sqrt{2T \log\left( \frac{1}{\delta} \right)}. \qquad \square$$

## H.2 Bounding ◆

◆ can be bounded by $\mathcal{O}\left( \gamma T \right)$ by definition of $\pi^{(t)}$ in Algorithm 2.

**Lemma H.3** (◆). *The difference between $\widetilde{R}^{(T)}$ and $R^{(T),\text{external}} \left( \text{Alg}, \left( \widetilde{\boldsymbol{u}}^{(t)} \right)_{t=1}^{T} \right)$ satisfies:*

$$\left| \widetilde{R}^{(T)} - R^{(T),\text{external}} \left( \text{Alg}, \left( \widetilde{\boldsymbol{u}}^{(t)} \right)_{t=1}^{T} \right) \right| \leq 2\gamma T.$$

*Proof.* Let $\bar{\pi}^{(t+1)} = \text{Alg}\left( \left( \widetilde{\boldsymbol{u}}^{(s)} \right)_{s=1}^{t} \right)$. Then,

$$\left| \widetilde{R}^{(T)} - R^{(T),\text{external}} \left( \text{Alg}, \left( \widetilde{\boldsymbol{u}}^{(t)} \right)_{t=1}^{T} \right) \right| = \left| \max_{\widehat{\pi} \in \Delta^{\mathcal{A}}} \sum_{t=1}^{T} \left\langle \widetilde{\boldsymbol{u}}^{(s)}, \widehat{\pi} - \pi^{(t)} \right\rangle - \max_{\widehat{\pi} \in \Delta^{\mathcal{A}}} \sum_{t=1}^{T} \left\langle \widetilde{\boldsymbol{u}}^{(t)}, \widehat{\pi} - \bar{\pi}^{(t)} \right\rangle \right|$$

$$= \left| \sum_{t=1}^{T} \left\langle \widetilde{\boldsymbol{u}}^{(t)}, \bar{\pi}^{(t)} - \pi^{(t)} \right\rangle \right|$$

$$= \left| \sum_{t=1}^{T} \left\langle \widetilde{\boldsymbol{u}}^{(t)}, \bar{\pi}^{(t)} - \left( (1-\gamma)\bar{\pi}^{(t)} + \gamma \frac{\mathbf{1}(\mathcal{A})}{|\mathcal{A}|} \right) \right\rangle \right|$$

$$\leq \gamma \sum_{t=1}^{T} \left\| \widetilde{\boldsymbol{u}}^{(t)} \right\|_{\infty} \cdot \left( \left\| \bar{\pi}^{(t)} \right\|_1 + \left\| \frac{\mathbf{1}(\mathcal{A})}{|\mathcal{A}|} \right\|_1 \right) \leq 2\gamma T. \qquad \square$$

# I Proof of Theorem 6.2

**Theorem I.1** (Formal version of Theorem 6.2). *Consider **AvgUtil Rank** with full-information feedback and Algorithm 3. For any $\delta \in (0, 1)$, $T > 0$, and any no-regret learning algorithm with numeric utility feedback $\text{Alg}$ that satisfies Assumption 6.1, with probability at least $(1 - \delta)$, by*

*choosing* $m = 2T^{\frac{2}{3}} \log\left(\frac{4|\mathcal{A}|T}{\delta}\right)$, $R^{(T),\text{external}}$ *satisfies*

$$R^{(T),\text{external}} \leq R^{(T),\text{external}}\left(\text{Alg}, \left(\boldsymbol{u}^{(t)}\right)_{t=1}^{T}\right) + |\mathcal{A}|\tau\left(e^{\frac{1}{\tau}}+1\right)^2 LT^{\frac{5}{3}} + 4T^{\frac{2}{3}}\log\left(\frac{4|\mathcal{A}|T}{\delta}\right)$$

(I.1)

$$+ 4|\mathcal{A}|LT^{\frac{4}{3}}\left(\log T + 1\right)\left(\log\left(\frac{4|\mathcal{A}|T}{\delta}\right)\right)^2 + 4|\mathcal{A}|LT^{\frac{5}{3}}\log\left(\frac{4|\mathcal{A}|T}{\delta}\right).$$

*Proof.* Let $\bar{\pi}^{(t+1)} = \text{Alg}\left(\left(\boldsymbol{u}^{(s)}\right)_{s=1}^{t}\right)$, *i.e.*, the strategy generated by Alg when the ground-truth utility vectors are given. Then,

$$\left| R^{(T),\text{external}} - R^{(T),\text{external}}\left(\text{Alg}, \left(\boldsymbol{u}^{(t)}\right)_{t=1}^{T}\right) \right|$$

$$= \left| \max_{\widehat{\pi} \in \Delta^{\mathcal{A}}} \sum_{t=1}^{T} \left\langle \boldsymbol{u}^{(t)}, \widehat{\pi} - \pi^{(t)} \right\rangle - \max_{\widehat{\pi} \in \Delta^{\mathcal{A}}} \sum_{t=1}^{T} \left\langle \boldsymbol{u}^{(t)}, \widehat{\pi} - \bar{\pi}^{(t)} \right\rangle \right|$$

$$= \left| \sum_{t=1}^{T} \left\langle \boldsymbol{u}^{(t)}, \bar{\pi}^{(t)} - \pi^{(t)} \right\rangle \right|$$

$$\leq \sum_{t=1}^{m-1} \left\| \boldsymbol{u}^{(t)} \right\|_{\infty} \cdot \left\| \bar{\pi}^{(t)} - \pi^{(t)} \right\|_{1} + \sum_{t=m}^{T} \left\| \boldsymbol{u}^{(t)} \right\| \cdot \left\| \bar{\pi}^{(t)} - \pi^{(t)} \right\|$$

$$\leq 2m + \sum_{t=m}^{T} \left\| \boldsymbol{u}^{(t)} \right\| \cdot \left\| \bar{\pi}^{(t)} - \pi^{(t)} \right\|.$$

By Assumption 6.1 and Theorem 5.1, for any $t \geq m$, with probability at least $1 - \delta$ we have

$$\left\| \bar{\pi}^{(t)} - \pi^{(t)} \right\| \leq Lt \left\| \widetilde{\boldsymbol{u}}_{\text{avg}}^{(t)} - \boldsymbol{u}_{\text{avg}}^{(t)} \right\| \leq Lt\sqrt{|\mathcal{A}|}\left( \tau\left(e^{\frac{1}{\tau}}+1\right)^2 \sqrt{\frac{\log\left(\frac{4|\mathcal{A}|T}{\delta}\right)}{m}} + \sum_{s=t-m+1}^{t-1} \frac{2}{s+1} \right).$$

Then,

$$\left| R^{(T),\text{external}} - R^{(T),\text{external}}\left(\text{Alg}, \left(\boldsymbol{u}^{(t)}\right)_{t=1}^{T}\right) \right|$$

$$\leq 2m + L|\mathcal{A}|\tau\left(e^{\frac{1}{\tau}}+1\right)^2 \sqrt{\frac{\log\left(\frac{4|\mathcal{A}|T}{\delta}\right)}{m}} T^2 + 2L|\mathcal{A}| \sum_{t=m}^{T} t \sum_{s=t-m+1}^{t-1} \frac{1}{s+1}$$

$$\leq 2m + L|\mathcal{A}|\tau\left(e^{\frac{1}{\tau}}+1\right)^2 \sqrt{\frac{\log\left(\frac{4|\mathcal{A}|T}{\delta}\right)}{m}} T^2 + L|\mathcal{A}| \sum_{t=1}^{T} \frac{m(2t+m-1)}{t}$$

$$\leq 2m + L|\mathcal{A}|\tau\left(e^{\frac{1}{\tau}}+1\right)^2 \sqrt{\frac{\log\left(\frac{4|\mathcal{A}|T}{\delta}\right)}{m}} T^2 + Lm^2|\mathcal{A}| \sum_{t=1}^{T} \frac{1}{t} + 2|\mathcal{A}|mLT$$

$$\leq L|\mathcal{A}|\tau\left(e^{\frac{1}{\tau}}+1\right)^2 \sqrt{\frac{\log\left(\frac{4|\mathcal{A}|T}{\delta}\right)}{m}} T^2 + 2m + Lm^2|\mathcal{A}|\left(\log T + 1\right) + 2|\mathcal{A}|mLT.$$

By choosing $m = 2T^{\frac{2}{3}}\log\left(\frac{4|\mathcal{A}|T}{\delta}\right)$, we have

$$R^{(T),\text{external}} \leq R^{(T),\text{external}}\left(\text{Alg}, \left(\boldsymbol{u}^{(t)}\right)_{t=1}^{T}\right) + \mathcal{O}\left(LT^{\frac{5}{3}}\log\left(\frac{4|\mathcal{A}|T}{\delta}\right)\right).$$

Moreover, now $m \geq \frac{2\log\left(\frac{2T}{\delta}\right)}{p^4}$, where $p = 1$ since we are considering full-information feedback. $\qed$

## J    PROOF OF THEOREM 6.3

**Theorem J.1** (Formal version of Theorem 6.3). *Consider* **AvgUtil Rank** *with bandit feedback and Algorithm 3. For any* $\delta \in (0,1)$, $T > 0$, *and any no-regret learning algorithm with numeric utility feedback* Alg *that satisfies Assumption 6.1, with probability at least* $(1 - \delta)$, *by choosing* $m = 2T^{\frac{2}{3}}|\mathcal{A}|^4 \log\left(\frac{12|\mathcal{A}|T}{\delta}\right)$, $\gamma = \min\left\{L^{\frac{1}{3}}T^{\frac{5}{18}}\left(P^{(T)}\right)^{\frac{1}{6}}, 1\right\}$, *and* $M = \max\left\{4T^{\frac{5}{6}}\left(P^{(T)}\right)^{-\frac{1}{2}}|\mathcal{A}|^4 \log\left(\frac{12|\mathcal{A}|^2T}{\delta}\right), 2m\right\}$, $R^{(T)}$ *satisfies*

$$R^{(T)} \leq R^{(T),\text{external}}\left(\text{Alg}, \left(\boldsymbol{u}^{(t)}\right)_{t=1}^{T}\right) + L|\mathcal{A}|TW^{(T)} + 2\gamma\sqrt{|\mathcal{A}|T} + 2\sqrt{2T\log\left(\frac{3}{\delta}\right)}, \quad \text{(J.1)}$$

*where*

$$C_\delta := \frac{|\mathcal{A}|\log\left(\frac{3|\mathcal{A}|^2T}{\delta}\right)}{\gamma}$$

$$W^{(T)} := 4C_\delta(\log T + 1)\left(\frac{T^2}{M}\frac{\tau|\mathcal{A}|\left(e^{\frac{1}{\tau}}+1\right)^2}{\gamma}\sqrt{\frac{\log\left(\frac{12|\mathcal{A}|T}{\delta}\right)}{m}} + 16KC_\delta\frac{m}{M}T\right)$$

$$+ M(\log T + 1)P^{(T)} + 2M(\log T + 2).$$

*Proof.* In the first part of the proof, we will bound $\left\|\widetilde{\boldsymbol{u}}^{(t)}_{\text{empirical}} - \boldsymbol{u}^{(t)}_{\text{empirical}}\right\|_\infty$. According to Theorem 5.1 and union bound, since each action is proposed with probability at least $\frac{\gamma}{|\mathcal{A}|}$, with probability at least $1 - \frac{\delta}{3}$, for any $t \geq m$, we have

$$\left\|\widetilde{\boldsymbol{u}}^{(t)}_{\text{empirical}} - \boldsymbol{u}^{(t)}_{\text{empirical}}\right\|_\infty \leq \frac{\tau|\mathcal{A}|\left(e^{\frac{1}{\tau}}+1\right)^2}{\gamma}\sqrt{\frac{\log\left(\frac{12|\mathcal{A}|T}{\delta}\right)}{m}} + \sum_{s=t-m+1}^{t-1}\left\|\boldsymbol{u}^{(s+1)}_{\text{empirical}} - \boldsymbol{u}^{(s)}_{\text{empirical}}\right\|_\infty.$$

Let $\#_{o^{(t)}}(a)$ be the number of action $a \in \mathcal{A}$ being proposed in $o^{(t)}$. Then, for any $t \in [T-1]$ and $a \in \mathcal{A}$, we have

$$\left|u^{(t+1)}_{\text{empirical}}(a) - u^{(t)}_{\text{empirical}}(a)\right| = \left|\frac{u^{(t)}_{\text{empirical}}(a)\sum_{s=1}^{t}\#_{o^{(s)}}(a) + u^{(t+1)}(a)\#_{o^{(t+1)}}(a)}{\sum_{s=1}^{t}\#_{o^{(s)}}(a) + \#_{o^{(t+1)}}(a)} - u^{(t)}_{\text{empirical}}(a)\right|$$

$$\leq \left|\frac{u^{(t)}_{\text{empirical}}(a)\#_{o^{(t+1)}}(a)}{\sum_{s=1}^{t}\#_{o^{(s)}}(a) + \#_{o^{(t+1)}}(a)}\right| + \left|\frac{u^{(t+1)}(a)\#_{o^{(t+1)}}(a)}{\sum_{s=1}^{t}\#_{o^{(s)}}(a) + \#_{o^{(t+1)}}(a)}\right|$$

$$\leq K\left|\frac{u^{(t)}_{\text{empirical}}(a)}{\sum_{s=1}^{t}\#_{o^{(s)}}(a) + \#_{o^{(t+1)}}(a)}\right| + K\left|\frac{u^{(t+1)}(a)}{\sum_{s=1}^{t}\#_{o^{(s)}}(a) + \#_{o^{(t+1)}}(a)}\right|.$$

Next, we will show that since each action will be proposed with probability at least $\frac{\gamma}{|\mathcal{A}|}$, with high probability, there is a lowerbound for $\sum_{s=1}^{t}\#_{o^{(s)}}(a)$ for any timestep $t$.

**Lemma J.2.** *Consider the case when actions are proposed with probability at least $p > 0$ at each timestep. Then, for any $\delta > 0$, any action $a \in \mathcal{A}$, and $T > 0$, with probability at least $1 - \delta$, the following holds for any $t \geq \frac{\log\left(\frac{|\mathcal{A}|T}{\delta}\right)}{p}$:*

$$\exists t' \in [T], \quad \text{such that} \quad t - \frac{\log\left(\frac{|\mathcal{A}|T}{\delta}\right)}{p} \leq t' \leq t \text{ and } a \in o^{(t')}. \quad \text{(J.2)}$$

*Proof.* For any $t \geq \frac{\log\left(\frac{|\mathcal{A}|T}{\delta}\right)}{p}$ and action $a \in \mathcal{A}$, the probability of (J.2) does not hold is at most

$$(1 - p)^{\frac{\log\left(\frac{|\mathcal{A}|T}{\delta}\right)}{p}} \leq \exp\left(-\log\left(\frac{|\mathcal{A}|T}{\delta}\right)\right) = \frac{\delta}{|\mathcal{A}|T}.$$

Therefore, by union bound, with probability $1 - \delta$, (J.2) holds for any $t \in [T]$ and any action $a \in \mathcal{A}$. $\square$

For notational simplicity, let $C_\delta := \frac{|\mathcal{A}| \log\left(\frac{3|\mathcal{A}|^2 T}{\delta}\right)}{\gamma}$. According to Lemma J.2, with probability at least $1 - \frac{\delta}{3|\mathcal{A}|}$, for any timestep $t \geq C_\delta$, we have

$$\sum_{s=1}^{t} \#_{o^{(s)}}(a) \geq \left\lfloor \frac{t}{C_\delta} \right\rfloor \geq \frac{t}{2C_\delta}. \tag{J.3}$$

By union bound, (J.3) holds for any action $a$ with probability $1 - \frac{\delta}{3}$.

Therefore, for any $t \in [T - 1]$ and $a \in \mathcal{A}$, we have

$$\left| u_{\text{empirical}}^{(t+1)}(a) - u_{\text{empirical}}^{(t)}(a) \right| \leq 4K \frac{C_\delta}{t + 1}.$$

It holds for $t < C_\delta - 1$ because $4K\frac{C_\delta}{t+1} \geq 4K \geq 2$ and all utilities are bounded in $[-1, 1]$. Finally, by Theorem 5.1 and union bound, with probability at least $1 - \frac{2\delta}{3}$, we have

$$\left\| \widetilde{\boldsymbol{u}}_{\text{empirical}}^{(t)} - \boldsymbol{u}_{\text{empirical}}^{(t)} \right\|_\infty \leq \frac{\tau|\mathcal{A}|\left(e^{\frac{1}{\tau}} + 1\right)^2}{\gamma} \sqrt{\frac{\log\left(\frac{12|\mathcal{A}|T}{\delta}\right)}{m}} + 4KC_\delta \sum_{s=t-m+1}^{t-1} \frac{1}{s + 1}.$$

Let $n^{(t)}(a) := \sum_{s=1}^{t} \#_{o^{(s)}}(a)$ for any $a \in \mathcal{A}$ as the number of times action $a$ is proposed up to timestep $t$. For any $a \in \mathcal{A}$ and $t \geq M$, we define

$$u_{\text{avg}-\text{est}}^{(t)}(a) := \frac{1}{\lfloor t/M \rfloor} \sum_{s=1}^{\lfloor t/M \rfloor} \frac{u_{\text{empirical}}^{(s \cdot M)}(a) n^{(s \cdot M)}(a) - u_{\text{empirical}}^{((s-1)M)}(a) n^{((s-1)M)}(a)}{n^{(s \cdot M)}(a) - n^{((s-1)M)}(a)}$$

$$\widetilde{u}_{\text{avg}-\text{est}}^{(t)}(a) := \frac{1}{\lfloor t/M \rfloor} \sum_{s=1}^{\lfloor t/M \rfloor} \frac{\widetilde{u}_{\text{empirical}}^{(s \cdot M)}(a) n^{(s \cdot M)}(a) - \widetilde{u}_{\text{empirical}}^{((s-1)M)}(a) n^{((s-1)M)}(a)}{n^{(s \cdot M)}(a) - n^{((s-1)M)}(a)}.$$

Note that we define $u_{\text{empirical}}^{(0)}(a) = \widetilde{u}_{\text{empirical}}^{(0)}(a) = n_{\text{empirical}}^{(0)}(a) = 0$. For $t < M$, we define $u_{\text{avg}-\text{est}}^{(t)}(a) = \widetilde{u}_{\text{avg}-\text{est}}^{(t)}(a) = 0$ for any action $a \in \mathcal{A}$. In the rest of the proof, we will bound $\left\| \widetilde{\boldsymbol{u}}_{\text{avg}-\text{est}}^{(t)} - \boldsymbol{u}_{\text{avg}-\text{est}}^{(t)} \right\|_\infty$ and $\left\| \boldsymbol{u}_{\text{avg}-\text{est}}^{(t)} - \boldsymbol{u}_{\text{avg}}^{(t)} \right\|_\infty$ individually.

## J.1 $\left\| \widetilde{\mathbf{u}}_{\text{avg}-\text{est}}^{(t)} - \mathbf{u}_{\text{avg}-\text{est}}^{(t)} \right\|_\infty$ UPPER BOUND

For any $a \in \mathcal{A}$, we have

$$\left| \widetilde{u}_{\text{avg}-\text{est}}^{(t)}(a) - u_{\text{avg}-\text{est}}^{(t)}(a) \right| \leq \frac{1}{\lfloor t/M \rfloor} \sum_{s=1}^{\lfloor t/M \rfloor} \frac{n^{(s \cdot M)}(a)}{n^{(s \cdot M)}(a) - n^{((s-1)M)}(a)} \left| \widetilde{u}_{\text{empirical}}^{(s \cdot M)}(a) - u_{\text{empirical}}^{(s \cdot M)}(a) \right|$$

$$+ \frac{1}{\lfloor t/M \rfloor} \sum_{s=1}^{\lfloor t/M \rfloor} \frac{n^{((s-1)M)}(a)}{n^{(s \cdot M)}(a) - n^{((s-1)M)}(a)} \left| \widetilde{u}_{\text{empirical}}^{((s-1)M)}(a) - u_{\text{empirical}}^{((s-1)M)}(a) \right|.$$

According to Lemma J.2, $n^{(s \cdot M)}(a) - n^{((s-1)M)}(a) \geq \frac{M}{2C_\delta}$ when $M \geq C_\delta$. Therefore, when $M \geq C_\delta$, since $n^{(s \cdot M)}(a) \leq s \cdot M$, we have

$$\left| \widetilde{u}_{\text{avg-est}}^{(t)}(a) - u_{\text{avg-est}}^{(t)}(a) \right|$$

$$\leq \frac{2C_\delta}{\lfloor t/M \rfloor} \sum_{s=1}^{\lfloor t/M \rfloor} s \left( \left| \widetilde{u}_{\text{empirical}}^{(s \cdot M)}(a) - u_{\text{empirical}}^{(s \cdot M)}(a) \right| + \left| \widetilde{u}_{\text{empirical}}^{((s-1)M)}(a) - u_{\text{empirical}}^{((s-1)M)}(a) \right| \right).$$

## J.2 $\left\| \mathbf{u}_{\text{avg-est}}^{(t)} - \mathbf{u}_{\text{avg}}^{(t)} \right\|_\infty$ UPPER BOUND

For any $a \in \mathcal{A}$,

$$\left\| u_{\text{avg-est}}^{(t)}(a) - u_{\text{avg}}^{(t)}(a) \right\|_\infty$$

$$= \left| \frac{1}{\lfloor t/M \rfloor} \sum_{s=1}^{\lfloor t/M \rfloor} \frac{u_{\text{empirical}}^{(s \cdot M)}(a) n^{(s \cdot M)}(a) - u_{\text{empirical}}^{((s-1)M)}(a) n^{((s-1)M)}(a)}{n^{(s \cdot M)}(a) - n^{((s-1)M)}(a)} - u_{\text{avg}}^{(t)}(a) \right|$$

$$\leq \underbrace{\left| \frac{1}{\lfloor t/M \rfloor} \sum_{s=1}^{\lfloor t/M \rfloor} \frac{u_{\text{empirical}}^{(s \cdot M)}(a) n^{(s \cdot M)}(a) - u_{\text{empirical}}^{((s-1)M)}(a) n^{((s-1)M)}(a)}{n^{(s \cdot M)}(a) - n^{((s-1)M)}(a)} - u_{\text{avg}}^{(M \lfloor t/M \rfloor)}(a) \right|}_{\spadesuit}$$

$$+ \underbrace{\left| u_{\text{avg}}^{(t)}(a) - u_{\text{avg}}^{(M \lfloor t/M \rfloor)}(a) \right|}_{\clubsuit}.$$

Note that $\clubsuit$ can be bounded by

$$\clubsuit = \left| \frac{(M \lfloor t/M \rfloor) u_{\text{avg}}^{(M \lfloor t/M \rfloor)}(a) + \sum_{s=M \lfloor t/M \rfloor + 1}^{t} u^{(s)}(a)}{t} - u_{\text{avg}}^{(M \lfloor t/M \rfloor)}(a) \right|$$

$$\leq \frac{M}{t} \left| u_{\text{avg}}^{(M \lfloor t/M \rfloor)}(a) \right| + \frac{1}{t} \left| \sum_{s=M \lfloor t/M \rfloor + 1}^{t} u^{(s)}(a) \right| \leq \frac{2M}{t}.$$

For $\spadesuit$, we have

$$\spadesuit = \left| \frac{1}{\lfloor t/M \rfloor} \sum_{s=1}^{\lfloor t/M \rfloor} \left( \frac{u_{\text{empirical}}^{(s \cdot M)}(a) n^{(s \cdot M)}(a) - u_{\text{empirical}}^{((s-1)M)}(a) n^{((s-1)M)}(a)}{n^{(s \cdot M)}(a) - n^{((s-1)M)}(a)} - \frac{1}{M} \sum_{s'=(s-1)M+1}^{s \cdot M} u^{(s')}(a) \right) \right|$$

$$\leq \frac{1}{\lfloor t/M \rfloor} \sum_{s=1}^{\lfloor t/M \rfloor} \left| \frac{u_{\text{empirical}}^{(s \cdot M)}(a) n^{(s \cdot M)}(a) - u_{\text{empirical}}^{((s-1)M)}(a) n^{((s-1)M)}(a)}{n^{(s \cdot M)}(a) - n^{((s-1)M)}(a)} - \frac{1}{M} \sum_{s'=(s-1)M+1}^{s \cdot M} u^{(s')}(a) \right|.$$

When $n^{(s \cdot M)}(a) - n^{((s-1)M)}(a) > 0$, both $\frac{u_{\text{empirical}}^{(s \cdot M)}(a) n^{(s \cdot M)}(a) - u_{\text{empirical}}^{((s-1)M)}(a) n^{((s-1)M)}(a)}{n^{(s \cdot M)}(a) - n^{((s-1)M)}(a)}$ and $\frac{1}{M} \sum_{s'=(s-1)M+1}^{s \cdot M} u^{(s')}(a)$ are in the convex hull of $\left\{ u^{(s')}(a) \right\}_{s'=(s-1)M+1}^{s \cdot M}$. Therefore,

$$\left| \frac{u_{\text{empirical}}^{(s \cdot M)}(a) n^{(s \cdot M)}(a) - u_{\text{empirical}}^{((s-1)M)}(a) n^{((s-1)M)}(a)}{n^{(s \cdot M)}(a) - n^{((s-1)M)}(a)} - \frac{1}{M} \sum_{s'=(s-1)M+1}^{s \cdot M} u^{(s')}(a) \right|$$

$$\leq \max_{(s-1)M+1 \leq s', s'' \leq s \cdot M} \left| u^{(s')}(a) - u^{(s'')}(a) \right|$$

$$\leq \sum_{s'=(s-1)M+1}^{s \cdot M - 1} \left| u^{(s'+1)}(a) - u^{(s')}(a) \right|.$$

Therefore, for any $t \geq M$ and $a \in \mathcal{A}$, we have

$$\left| u_{\text{avg}-\text{est}}^{(t)}(a) - u_{\text{avg}}^{(t)}(a) \right|_{\infty} \leq \frac{1}{\lfloor t/M \rfloor} \sum_{s=1}^{\lfloor t/M \rfloor} \sum_{s'=(s-1)M+1}^{s \cdot M - 1} \left| u^{(s'+1)}(a) - u^{(s')}(a) \right| + \frac{2M}{t}.$$

By combining all the pieces together, we have

$$\left| \widetilde{u}_{\text{avg}-\text{est}}^{(t)}(a) - u_{\text{avg}}^{(t)}(a) \right|$$

$$\leq \frac{2C_{\delta}}{\lfloor t/M \rfloor} \sum_{s=1}^{\lfloor t/M \rfloor} s \left( \left| \widetilde{u}_{\text{empirical}}^{(s \cdot M)}(a) - u_{\text{empirical}}^{(s \cdot M)}(a) \right| + \left| \widetilde{u}_{\text{empirical}}^{((s-1)M)}(a) - u_{\text{empirical}}^{((s-1)M)}(a) \right| \right)$$

$$+ \frac{1}{\lfloor t/M \rfloor} \sum_{s=1}^{\lfloor t/M \rfloor} \sum_{s'=(s-1)M+1}^{s \cdot M - 1} \left| u^{(s'+1)}(a) - u^{(s')}(a) \right| + \frac{2M}{t}.$$

Then, since

$$\sum_{t=M}^{T} \frac{1}{\lfloor t/M \rfloor} = \sum_{s=1}^{\lfloor T/M \rfloor} \sum_{t=s \cdot M}^{\min\{(s+1) \cdot M - 1, T\}} \frac{1}{s} \leq \sum_{s=1}^{\lfloor T/M \rfloor} \frac{M}{s},$$

we have

$$\sum_{t=1}^{T} \left| \widetilde{u}_{\text{avg}-\text{est}}^{(t)}(a) - u_{\text{avg}}^{(t)}(a) \right|$$

$$= \sum_{t=M}^{T} \left| \widetilde{u}_{\text{avg}-\text{est}}^{(t)}(a) - u_{\text{avg}}^{(t)}(a) \right| + \sum_{t=1}^{M-1} \left| \widetilde{u}_{\text{avg}-\text{est}}^{(t)}(a) - u_{\text{avg}}^{(t)}(a) \right|$$

$$\leq 2C_{\delta} \sum_{s=1}^{\lfloor T/M \rfloor} s \left( \left| \widetilde{u}_{\text{empirical}}^{(s \cdot M)}(a) - u_{\text{empirical}}^{(s \cdot M)}(a) \right| + \left| \widetilde{u}_{\text{empirical}}^{((s-1)M)}(a) - u_{\text{empirical}}^{((s-1)M)}(a) \right| \right) \sum_{s'=1}^{\lfloor T/M \rfloor} \frac{M}{s'}$$

$$+ \sum_{s=1}^{\lfloor T/M \rfloor} \left( \sum_{s'=(s-1)M+1}^{s \cdot M - 1} \left| u^{(s'+1)}(a) - u^{(s')}(a) \right| \right) \cdot \left( \sum_{s'=1}^{\lfloor T/M \rfloor} \frac{M}{s'} \right) + \sum_{t=1}^{T} \frac{2M}{t} + 2M$$

$$\leq 2C_{\delta}M \sum_{s=1}^{\lfloor T/M \rfloor} s \left( \left| \widetilde{u}_{\text{empirical}}^{(s \cdot M)}(a) - u_{\text{empirical}}^{(s \cdot M)}(a) \right| + \left| \widetilde{u}_{\text{empirical}}^{((s-1)M)}(a) - u_{\text{empirical}}^{((s-1)M)}(a) \right| \right) \left( \log \left( \lfloor T/M \rfloor \right) + 1 \right)$$

$$+ M \sum_{s=1}^{\lfloor T/M \rfloor} \left( \sum_{s'=(s-1)M+1}^{s \cdot M - 1} \left| u^{(s'+1)}(a) - u^{(s')}(a) \right| \right) \cdot \left( \log \left( \lfloor T/M \rfloor \right) + 1 \right) + 2M \left( \log T + 1 \right) + 2M.$$

When $s = 1$, $s \left| \widetilde{u}_{\text{empirical}}^{((s-1)M)}(a) - u_{\text{empirical}}^{((s-1)M)}(a) \right| = 0$ by definition. When $s > 1$, since $M \geq 2m$, we have

$$\frac{s}{(s-1)M - m + 2} \leq \frac{s}{(s-1)M/2 + 2} \leq \frac{s}{(s-1)M/2} \leq \frac{4s}{s \cdot M} = \frac{4}{M}.$$

Hence,

$$s \left| \widetilde{u}_{\text{empirical}}^{((s-1)M)}(a) - u_{\text{empirical}}^{((s-1)M)}(a) \right| \leq s \frac{\tau |\mathcal{A}| \left( e^{\frac{1}{\tau}} + 1 \right)^2}{\gamma} \sqrt{\frac{\log \left( \frac{12|\mathcal{A}|T}{\delta} \right)}{m}} + 4KC_{\delta} \sum_{s'=(s-1)M-m+1}^{(s-1)M-1} \frac{s}{s'+1}$$

$$\leq \frac{T}{M} \frac{\tau |\mathcal{A}| \left( e^{\frac{1}{\tau}} + 1 \right)^2}{\gamma} \sqrt{\frac{\log \left( \frac{12|\mathcal{A}|T}{\delta} \right)}{m}} + 16KC_{\delta} \frac{m}{M}.$$

Therefore,

$$\sum_{t=1}^{T}\left|\widetilde{u}^{(t)}_{\text{avg}-\text{est}}(a) - u^{(t)}_{\text{avg}}(a)\right|$$

$$\leq 4C_\delta M \cdot \lfloor T/M \rfloor (\log T + 1)\left(\frac{T}{M}\frac{\tau|\mathcal{A}|\left(e^{\frac{1}{\tau}}+1\right)^2}{\gamma}\sqrt{\frac{\log\left(\frac{12|\mathcal{A}|T}{\delta}\right)}{m}} + 16KC_\delta\frac{m}{M}\right)$$

$$+ M(\log T + 1)P^{(T)} + 2M(\log T + 1)$$

$$\leq 4C_\delta(\log T + 1)\left(\frac{T^2}{M}\frac{\tau|\mathcal{A}|\left(e^{\frac{1}{\tau}}+1\right)^2}{\gamma}\sqrt{\frac{\log\left(\frac{12|\mathcal{A}|T}{\delta}\right)}{m}} + 16KC_\delta\frac{m}{M}T\right)$$

$$+ M(\log T + 1)P^{(T)} + 2M(\log T + 2).$$

Lastly, similar to the proof in Appendix I, let $\bar{\pi}^{(t+1)} = \text{Alg}\left(\left(\boldsymbol{u}^{(s)}\right)_{s=1}^{t}\right)$. Then, we have

$$\left|R^{(T),\text{external}} - R^{(T),\text{external}}\left(\text{Alg}, \left(\boldsymbol{u}^{(t)}\right)_{t=1}^{T}\right)\right|$$

$$= \left|\sum_{t=1}^{T}\left\langle \boldsymbol{u}^{(t)}, \pi^{(t)} - \bar{\pi}^{(t)}\right\rangle\right|$$

$$\leq \sum_{t=1}^{T}\left\|\boldsymbol{u}^{(t)}\right\| \cdot \left\|\pi^{(t)} - \bar{\pi}^{(t)}\right\|$$

$$\leq \sqrt{|\mathcal{A}|}\sum_{t=1}^{T}\left\|(1-\gamma)\text{Alg}\left(\left(\widetilde{\boldsymbol{u}}^{(t)}_{\text{avg}-\text{est}}\right)_{s=1}^{t}\right) + \gamma\frac{\mathbf{1}(\mathcal{A})}{|\mathcal{A}|} - \bar{\pi}^{(t)}\right\|$$

$$\leq (1-\gamma)\sqrt{|\mathcal{A}|}\sum_{t=1}^{T}\left\|\text{Alg}\left(\left(\widetilde{\boldsymbol{u}}^{(t)}_{\text{avg}-\text{est}}\right)_{s=1}^{t}\right) - \bar{\pi}^{(t)}\right\| + \gamma\sqrt{|\mathcal{A}|}\sum_{t=1}^{T}\left\|\frac{\mathbf{1}(\mathcal{A})}{|\mathcal{A}|} - \bar{\pi}^{(t)}\right\|$$

$$\leq \sqrt{|\mathcal{A}|}\sum_{t=1}^{T}\left\|\text{Alg}\left(\left(\widetilde{\boldsymbol{u}}^{(t)}_{\text{avg}-\text{est}}\right)_{s=1}^{t}\right) - \bar{\pi}^{(t)}\right\| + 2\gamma\sqrt{|\mathcal{A}|}T.$$

Further, by Assumption 6.1, we have

$$\sqrt{|\mathcal{A}|}\sum_{t=1}^{T}\left\|\text{Alg}\left(\left(\widetilde{\boldsymbol{u}}^{(t)}_{\text{avg}-\text{est}}\right)_{s=1}^{t}\right) - \bar{\pi}^{(t)}\right\| \leq L\sqrt{|\mathcal{A}|}\sum_{t=1}^{T} t\left\|\widetilde{\boldsymbol{u}}^{(t)}_{\text{avg}-\text{est}} - \boldsymbol{u}^{(t)}_{\text{avg}}\right\|$$

$$\leq L|\mathcal{A}|T\sum_{t=1}^{T}\left\|\widetilde{\boldsymbol{u}}^{(t)}_{\text{avg}-\text{est}} - \boldsymbol{u}^{(t)}_{\text{avg}}\right\|_{\infty}.$$

By combining all the pieces together, we have

$$
\begin{aligned}
R^{(T),\text{external}} \leq & R^{(T),\text{external}}\left(\text{Alg}, \left(\boldsymbol{u}^{(t)}\right)_{t=1}^{T}\right) \\
& + L|\mathcal{A}|T\left(4C_\delta\left(\log T + 1\right)\left(\frac{T^2}{M}\frac{\tau|\mathcal{A}|\left(e^{\frac{1}{\tau}}+1\right)^2}{\gamma}\sqrt{\frac{\log\left(\frac{12|\mathcal{A}|T}{\delta}\right)}{m}} + 16KC_\delta\frac{m}{M}T\right)\right) \\
& + L|\mathcal{A}|T\left(M\left(\log T + 1\right)P^{(T)} + 2M\left(\log T + 2\right)\right) + 2\gamma\sqrt{|\mathcal{A}|}T \\
\leq & R^{(T),\text{external}}\left(\text{Alg}, \left(\boldsymbol{u}^{(t)}\right)_{t=1}^{T}\right) \\
& + \widetilde{\mathcal{O}}\left(\frac{\left(\log\left(\frac{1}{\delta}\right)\right)^{\frac{3}{2}}}{\gamma^2}\frac{LT^3}{M\sqrt{m}} + \frac{m\left(\log\left(\frac{1}{\delta}\right)\right)^2}{\gamma^2 M}LT^2 + LMP^{(T)}T + 2\gamma T\right),
\end{aligned}
$$

where $\widetilde{\mathcal{O}}$ hides all the $\log T$ terms. Lastly, by Lemma H.2, with probability at least $1 - \frac{\delta}{3}$, we have

$$
R^{(T)} \leq R^{(T),\text{external}} + 2\sqrt{2T\log\left(\frac{3}{\delta}\right)}.
$$

Moreover, by taking $\gamma = \min\left\{L^{\frac{1}{3}}T^{\frac{5}{18}}\left(P^{(T)}\right)^{\frac{1}{6}}, 1\right\}$, $m = 2T^{\frac{2}{3}}|\mathcal{A}|^4\log\left(\frac{12|\mathcal{A}|T}{\delta}\right)$, and $M = \max\left\{4T^{\frac{5}{6}}\left(P^{(T)}\right)^{-\frac{1}{2}}|\mathcal{A}|^4\log\left(\frac{12|\mathcal{A}|^2T}{\delta}\right), 2m\right\}$, we can easily verify that $m \geq \frac{2\log\left(\frac{6}{\delta}\right)}{\gamma^4}|\mathcal{A}|^4$ and $M \geq C_\delta$. Finally, by a union bound argument, we complete the proof. $\qquad\square$

## K  PROOF OF SECTION 7

**Lemma K.1.** *For any $T > 0$ and sequence of strategy profiles $\left(\boldsymbol{\pi}^{(1)}, \boldsymbol{\pi}^{(2)}, \ldots, \boldsymbol{\pi}^{(T)}\right)$, the variation of utility vectors of any player $i \in [N]$ satisfies that*

$$
\sum_{t=2}^{T}\left\|\boldsymbol{u}_i^{(t)} - \boldsymbol{u}_i^{(t-1)}\right\| \leq \sqrt{A}\prod_{j'=1}^{N}|\mathcal{A}_{j'}|\sum_{t=2}^{T}\sum_{j=1}^{N}\left\|\pi_j^{(t)} - \pi_j^{(t-1)}\right\|, \tag{K.1}
$$

*where $A = \max_j |\mathcal{A}_j|$.*

*Proof.* For any timestep $t$, player $i \in [N]$, and joint action $\boldsymbol{a}_{-i} \in \bigtimes_{j\neq i}\mathcal{A}_j$, let $\pi_{-i}^{(t)}(\boldsymbol{a}_{-i}) := \prod_{j\neq i}\pi_j^{(t)}(a_j)$.

Then, for any timestep $t$, player $i \in [N]$, and action $a_i \in \mathcal{A}_i$, we have

$$
\begin{aligned}
\left|u_i^{(t)}(a_i) - u_i^{(t-1)}(a_i)\right| & \leq \left|\sum_{\boldsymbol{a}'\in\bigtimes_{j=1}^{N}\mathcal{A}_j}\mathcal{U}_i(\boldsymbol{a}')\mathbb{1}\left(a_i' = a_i\right)\left(\pi_{-i}^{(t)}(\boldsymbol{a}'_{-i}) - \pi_{-i}^{(t-1)}(\boldsymbol{a}'_{-i})\right)\right| \\
& = \left|\left\langle\left(\mathcal{U}_i(a_i, \boldsymbol{a}'_{-i})\right)_{\boldsymbol{a}'_{-i}\in\bigtimes_{j\neq i}\mathcal{A}_j}, \pi_{-i}^{(t)} - \pi_{-i}^{(t-1)}\right\rangle\right| \\
& \leq \left\|\left(\mathcal{U}_i(a_i, \boldsymbol{a}'_{-i})\right)_{\boldsymbol{a}'_{-i}\in\bigtimes_{j\neq i}\mathcal{A}_j}\right\|_\infty \cdot \left\|\pi_{-i}^{(t)} - \pi_{-i}^{(t-1)}\right\|_1 \\
& \leq \left\|\pi_{-i}^{(t)} - \pi_{-i}^{(t-1)}\right\|_1.
\end{aligned}
$$

Further, for any $a, b, a', b' \in [0, 1]$, we have $|ab - a'b'| = |ab - ab' + ab' - a'b'| \leq a|b - b'| + |a - a'|b' \leq |a - a'| + |b - b'|$. Therefore, by recursively using it, for any $\boldsymbol{a}_{-i} \in \bigtimes_{j\neq i}\mathcal{A}_j$, we have

$$
\left|\pi_{-i}^{(t)}(\boldsymbol{a}_{-i}) - \pi_{-i}^{(t-1)}(\boldsymbol{a}_{-i})\right| = \left|\prod_{j\neq i}\pi_j^{(t)}(a_j) - \prod_{j\neq i}\pi_j^{(t-1)}(a_j)\right| \leq \sum_{j\neq i}\left|\pi_j^{(t)}(a_j) - \pi_j^{(t-1)}(a_j)\right|.
$$

Finally,

$$\left\| u_i^{(t)} - u_i^{(t-1)} \right\| \le \sqrt{|\mathcal{A}_i|} \left\| \pi_{-i}^{(t)} - \pi_{-i}^{(t-1)} \right\|_1 \le \sqrt{|\mathcal{A}_i|} \prod_{j=1}^{N} |\mathcal{A}_j| \sum_{j' \ne i} \left\| \pi_{j'}^{(t)} - \pi_{j'}^{(t-1)} \right\|_1. \qquad \square$$

### K.1 PROOF OF THEOREM 7.2 AND THEOREM 7.3

Before proving Theorem 7.2 and Theorem 7.3, we will show that when Assumption 7.1 is satisfied, the strategy variation is bounded.

**Lemma K.2.** *Suppose Assumption 7.1 is satisfied. For both full-information and bandit settings, Algorithm 2 satisfies the following,*

$$\sum_{t=1}^{T-1} \left\| \pi^{(t)} - \pi^{(t+1)} \right\| \le \eta T.$$

*Suppose Assumption 6.1 is also satisfied, then the following holds for Algorithm 3 in the bandit setting,*

$$\sum_{t=1}^{T-1} \left\| \pi^{(t)} - \pi^{(t+1)} \right\| \le \eta T + 2\sqrt{|\mathcal{A}|} L T.$$

*Proof.* For Algorithm 2 and the full-information setting, the proof simply follows from the fact that $\widetilde{\boldsymbol{u}}^{(t)} \in [-1, 1]^{\mathcal{A}}$ and Assumption 7.1.

For Algorithm 2 and the bandit setting, we have

$$\left\| \pi^{(t+1)} - \pi^{(t)} \right\| = (1 - \gamma) \left\| \mathrm{Alg}\left( \left( \widetilde{\boldsymbol{u}}^{(s)} \right)_{s=1}^{t+1} \right) - \mathrm{Alg}\left( \left( \widetilde{\boldsymbol{u}}^{(s)} \right)_{s=1}^{t} \right) \right\| \le \eta.$$

Thus, we can conclude the proof.

For Algorithm 3 and the bandit setting, for any $t \not\equiv 0 \pmod{M}$, we have $\widetilde{\boldsymbol{u}}_{\mathrm{avg-est}}^{(t)} = \widetilde{\boldsymbol{u}}_{\mathrm{avg-est}}^{(t-1)}$. Therefore, by Assumption 7.1, we have

$$\left\| \pi^{(t-1)} - \pi^{(t)} \right\| \le \left\| \mathrm{Alg}\left( \left( \widetilde{\boldsymbol{u}}_{\mathrm{avg-est}}^{(t-1)} \right)_{s=1}^{t-1} \right) - \mathrm{Alg}\left( \left( \widetilde{\boldsymbol{u}}_{\mathrm{avg-est}}^{(t)} \right)_{s=1}^{t} \right) \right\|$$

$$= \left\| \mathrm{Alg}\left( \left( \widetilde{\boldsymbol{u}}_{\mathrm{avg-est}}^{(t-1)} \right)_{s=1}^{t-1} \right) - \mathrm{Alg}\left( \left( \widetilde{\boldsymbol{u}}_{\mathrm{avg-est}}^{(t-1)} \right)_{s=1}^{t} \right) \right\| \le \eta.$$

For any $t \equiv 0 \pmod{M}$, let $\boldsymbol{u} = \frac{\widetilde{u}_{\mathrm{empirical}}^{(t)}(a) n^{(t)}(a) - \widetilde{u}_{\mathrm{empirical}}^{(t-M)}(a) n^{(t-M)}(a)}{n^{(t)}(a) - n^{(t-M)}(a)}$, then we have

$$\left\| \widetilde{\boldsymbol{u}}_{\mathrm{avg-est}}^{(t)} - \widetilde{\boldsymbol{u}}_{\mathrm{avg-est}}^{(t-1)} \right\| = \left\| \left( \frac{(t/M - 1) \widetilde{\boldsymbol{u}}_{\mathrm{avg-est}}^{(t-1)} + \boldsymbol{u}}{t/M} \right) - \widetilde{\boldsymbol{u}}_{\mathrm{avg-est}}^{(t-1)} \right\|$$

$$\le \frac{M}{t} \left( \left\| \widetilde{\boldsymbol{u}}_{\mathrm{avg-est}}^{(t-1)} \right\| + \|\boldsymbol{u}\| \right) \le \frac{2M}{t} \sqrt{|\mathcal{A}|}.$$

Therefore,

$$\left\| \pi^{(t-1)} - \pi^{(t)} \right\| \le \left\| \mathrm{Alg}\left( \left( \widetilde{\boldsymbol{u}}_{\mathrm{avg-est}}^{(t-1)} \right)_{s=1}^{t-1} \right) - \mathrm{Alg}\left( \left( \widetilde{\boldsymbol{u}}_{\mathrm{avg-est}}^{(t)} \right)_{s=1}^{t} \right) \right\|$$

$$\le \left\| \mathrm{Alg}\left( \left( \widetilde{\boldsymbol{u}}_{\mathrm{avg-est}}^{(t-1)} \right)_{s=1}^{t-1} \right) - \mathrm{Alg}\left( \left( \widetilde{\boldsymbol{u}}_{\mathrm{avg-est}}^{(t-1)} \right)_{s=1}^{t} \right) \right\| + \left\| \mathrm{Alg}\left( \left( \widetilde{\boldsymbol{u}}_{\mathrm{avg-est}}^{(t-1)} \right)_{s=1}^{t} \right) - \mathrm{Alg}\left( \left( \widetilde{\boldsymbol{u}}_{\mathrm{avg-est}}^{(t)} \right)_{s=1}^{t} \right) \right\|$$

$$\overset{(i)}{\le} \eta + L t \left( \frac{2M}{t} \sqrt{|\mathcal{A}|} \right)$$

$$= \eta + 2 L M \sqrt{|\mathcal{A}|},$$

where $(i)$ uses Assumption 6.1 and Assumption 7.1. Then, the accumulated variation of $\pi^{(t)}$ over time is bounded by

$$\sum_{t=1}^{T-1} \left\| \pi^{(t+1)} - \pi^{(t)} \right\| \leq \eta T + 2\sqrt{|\mathcal{A}|} L M \frac{T}{M} = \eta T + 2\sqrt{|\mathcal{A}|} L T,$$

since there are at most $\frac{T}{M}$ timesteps of $t \in [T]$ satisfying $t \equiv 0 \pmod{M}$. $\square$

With Lemma K.2, we can prove that $R_i^{(T),\text{external}}$ is sublinear for any player $i \in [N]$ by Theorem 5.2, Theorem 6.2, and Theorem 6.3.[2] Then, by the folklore result that no-external-regret learning leads to approximate CCE (Hart & Mas-Colell, 2000; Blum & Mansour, 2007), Theorem 7.2 and Theorem 7.3 are proved. $\square$

**Remark K.3.** *With the hardness in Theorem 4.3, under* **AvgUtil Rank** *feedback, both our no-regret result for the online setting and the equilibrium computation result for the game setting hold for a constant $\tau > 0$ (that cannot be arbitrarily small). However, we note that the equilibrium computation result may still be possible when $\tau \to 0^+$: with such a* deterministic *ranking model, the* best-response *action against the history play of the opponents is now available, precisely leading to the celebrated algorithm of* fictitious-play *(FP) (Robinson, 1951; Brown, 1951). FP is known to converge to an equilibrium in certain games (Robinson, 1951; Monderer & Shapley, 1996; Sela, 1999; Berger, 2005) (with (slow) convergence rates (Robinson, 1951; Daskalakis & Pan, 2014; Abernethy et al., 2021)), despite that it fails to be no-regret in the online setting (Fudenberg & Levine, 1995; 1998).*

## L  PROPERTIES OF FOLLOW-THE-REGULARIZED-LEADER (FTRL)

Firstly, we will define the strongly convex function.

**Definition L.1.** *For any integer $n$ and a convex set $\mathcal{X} \subseteq \mathbb{R}^n$, a differentiable function $\psi(\boldsymbol{x}): \mathcal{X} \to \mathbb{R}$ is called $c_0$-strongly convex ($c_0 > 0$) when*

$$\psi(\boldsymbol{x}) \geq \psi(\boldsymbol{x}') + \langle \nabla \psi(\boldsymbol{x}'), \boldsymbol{x} - \boldsymbol{x}' \rangle + \frac{c_0}{2} \left\| \boldsymbol{x} - \boldsymbol{x}' \right\|^2 \tag{L.1}$$

*holds for any $\boldsymbol{x}, \boldsymbol{x}' \in \mathcal{X}$.*

Specifically, if (L.1) holds for $c_0 = 0$, then we call $\psi$ a convex function.

Next, we will introduce the well-known no-regret learning algorithm of Follow-The-Regularized-Leader (Shalev-Shwartz et al., 2012; Hazan et al., 2016).

**Definition L.2** (Follow-The-Regularized-Leader (FTRL)). *For any $T > 0$ and at any timestep $t \in \{0\} \cup [T-1]$, given the utility vectors $\left( \boldsymbol{u}^{(s)} \right)_{s=1}^t$, the strategy at timestep $t+1$, $\pi^{(t+1)}$, is defined as,*

$$\pi^{(t+1)} = \operatorname*{argmax}_{\pi \in \Delta^{\mathcal{A}}} \left( \lambda \sum_{s=1}^t \left\langle \boldsymbol{u}^{(s)}, \pi \right\rangle - \psi(\pi) \right), \tag{FTRL}$$

*for some constant $\lambda > 0$. Typically, $\lambda$ is taken to be $\Theta\left(T^{-r}\right)$ for some constant $r > 0$.*

Next, we will introduce the smoothness of (FTRL).

**Lemma L.3.** *For any $c_0$-strongly convex and differentiable function $\psi: \Delta^{\mathcal{A}} \to \mathbb{R}$, (FTRL) satisfies Assumption 6.1 and Assumption 7.1 with $L = \frac{\lambda}{c_0}$ and $\eta = \frac{\lambda}{c_0}\sqrt{|\mathcal{A}|}$.*

*Proof.* By the first-order optimality, at any timestep $t \in \{0\} \cup [T-1]$ for any two sequences of utility vectors $\left( \boldsymbol{u}^{(s)} \right)_{s=1}^t$ and $\left( \boldsymbol{u}'^{(s)} \right)_{s=1}^t$, let the corresponding strategy generated by (FTRL) be

---

[2]Inspecting the proofs of these theorems shows that one can readily derive an upper bound on $R^{(T),\text{external}}$ that is tighter than the corresponding bound on $R^{(T)}$ in the bandit setting. Consequently, $R_i^{(T),\text{external}}$ can also be bounded in games under the bandit feedback.

$\pi^{(t+1)}$ and $\pi'^{(t+1)}$ respectively, we have

$$\left\langle \lambda \sum_{s=1}^{t} \boldsymbol{u}^{(s)} - \nabla \psi \left( \pi^{(t+1)} \right), \pi'^{(t+1)} - \pi^{(t+1)} \right\rangle \leq 0$$

$$\left\langle \lambda \sum_{s=1}^{t} \boldsymbol{u}'^{(s)} - \nabla \psi \left( \pi'^{(t+1)} \right), \pi^{(t+1)} - \pi'^{(t+1)} \right\rangle \leq 0.$$

By summing them up and rearranging the terms, we have

$$\left\langle \lambda \sum_{s=1}^{t} \boldsymbol{u}'^{(s)} - \lambda \sum_{s=1}^{t} \boldsymbol{u}^{(s)}, \pi'^{(t+1)} - \pi^{(t+1)} \right\rangle \geq \left\langle \nabla \psi \left( \pi'^{(t+1)} \right) - \nabla \psi \left( \pi^{(t+1)} \right), \pi'^{(t+1)} - \pi^{(t+1)} \right\rangle.$$

Since $\psi$ is $c_0$-strongly convex, we have

$$\psi \left( \pi^{(t+1)} \right) \geq \psi \left( \pi'^{(t+1)} \right) + \left\langle \nabla \psi \left( \pi'^{(t+1)} \right), \pi^{(t+1)} - \pi'^{(t+1)} \right\rangle + \frac{c_0}{2} \left\| \pi^{(t+1)} - \pi'^{(t+1)} \right\|^2$$

$$\psi \left( \pi'^{(t+1)} \right) \geq \psi \left( \pi^{(t+1)} \right) + \left\langle \nabla \psi \left( \pi^{(t+1)} \right), \pi'^{(t+1)} - \pi^{(t+1)} \right\rangle + \frac{c_0}{2} \left\| \pi^{(t+1)} - \pi'^{(t+1)} \right\|^2.$$

By summing them up and rearranging the terms, we have

$$\left\langle \nabla \psi \left( \pi'^{(t+1)} \right) - \nabla \psi \left( \pi^{(t+1)} \right), \pi'^{(t+1)} - \pi^{(t+1)} \right\rangle \geq c_0 \left\| \pi^{(t+1)} - \pi'^{(t+1)} \right\|^2.$$

Therefore,

$$c_0 \left\| \pi^{(t+1)} - \pi'^{(t+1)} \right\|^2 \leq \left\langle \lambda \sum_{s=1}^{t} \boldsymbol{u}'^{(s)} - \lambda \sum_{s=1}^{t} \boldsymbol{u}^{(s)}, \pi'^{(t+1)} - \pi^{(t+1)} \right\rangle$$

$$\overset{(i)}{\leq} \left\| \lambda \sum_{s=1}^{t} \boldsymbol{u}'^{(s)} - \lambda \sum_{s=1}^{t} \boldsymbol{u}^{(s)} \right\| \cdot \left\| \pi'^{(t+1)} - \pi^{(t+1)} \right\|,$$

where $(i)$ is by Hölder's Inequality. Then,

$$\left\| \pi^{(t+1)} - \pi'^{(t+1)} \right\| \leq \frac{1}{c_0} \left\| \lambda \sum_{s=1}^{t} \boldsymbol{u}'^{(s)} - \lambda \sum_{s=1}^{t} \boldsymbol{u}^{(s)} \right\|,$$

so that (FTRL) satisfies Assumption 6.1 with $L = \frac{\lambda}{c_0}$. Furthermore, note that the results above also hold for sequences of utility vectors of different lengths (not necessarily equal to length $t$ simultaneously). As a result, we have

$$\left\| \pi^{(t+1)} - \pi^{(t)} \right\| \leq \frac{\lambda}{c_0} \left\| \sum_{s=1}^{t} \boldsymbol{u}^{(s)} - \sum_{s=1}^{t-1} \boldsymbol{u}^{(s)} \right\| = \frac{\lambda}{c_0} \left\| \boldsymbol{u}^{(t)} \right\| \leq \frac{\lambda}{c_0} \sqrt{|\mathcal{A}|},$$

for any $t \in \{0\} \cup [T-1]$, which implies that $\eta = \frac{\lambda}{c_0} \sqrt{|\mathcal{A}|}$ in Assumption 7.1 for (FTRL). $\qquad \square$

## M  CONCLUSION AND LIMITATIONS

In this paper, we studied online learning and equilibrium computation with ranking feedback, which is particularly relevant to application scenarios with humans in the loop. Focusing on the classical (external-)regret metric, we designed novel hardness instances to show that achieving sublinear regret can be hard in general, in a few different ranking models and feedback settings. We then developed new algorithms to achieve sublinear regret under an additional assumption on the sublinear variation of the utility, leading to an equilibrium computation result in the repeated game setting. Finally, we justify the effectiveness of our approach by simulating routing the user's query to the optimal LLM. We believe our work paves the way for promising avenues of future research. For example, it would be interesting to close the gap between the lower-bound and the positive result for **AvgUtil Rank** under bandit feedback, *i.e.*, either show the hardness when $\tau$ is a *constant* or achieve sublinear regret for constant $\tau$ without Assumption 4.2. Moreover, applying our algorithms to real-world datasets with ranking feedback, such as ride-sharing and match-dating, would also be of great interest.

