# OpenReview forum: "Online Learning and Equilibrium Computation with Ranking Feedback"
_ICLR.cc/2026/Conference — ICLR 2026 Oral_

### Official Review · Reviewer_7iCZ · 2025-10-21

**Soundness:** 3
**Presentation:** 3
**Contribution:** 1
**Rating:** 2
**Confidence:** 3

**Summary:**

A common setting for online learning (in the broadest sense) is adapting to feedback where utilities for different actions cannot be computed. Instead, learners may have access to ranking information where the adversary selects one or more actions among a set of actions. This work considers ranking feedback generated by the Plackett-Luce ranking model assuming ranking vectors from either instantaneous or time-averaged utility vectors. The authors formally establish that no-regret learning is not possible when the ranking is deterministic, i.e. actions are ranked in order of utility. When there is some temperature in the rankings, no-regret learning can be achieved in the full-information setting and in the bandit setting under an additional sub-linearity assumption on the utility vectors. This leads to a CCE as is standard in games where all participants employ a no-regret learning algorithm.

**Strengths:**

- The paper motivates from a real phenomenon, giving the examples of RLHF or customers for a service providing a single instance of feedback where utility cannot be easily computed.
- They provide formal definitions of their notions of utilities and the standard no-regret guarantees, as well as guarantees when no-regret cannot be achieved.
- The paper appears theoretically sound.

**Weaknesses:**

The paper motivates their problem from real cases, but then sets up a theoretical framework without demonstrating practical advantages/successes for the real problem they identify as the motivation. In my view, not establishing this connection limits its impact among practitioners in the motivating cases in the short or long term. For example, it is unclear to me how bandits with ranking feedback, as formalized here, applies to the setting of RLHF, which is presented as a motivating example for this work.
The authors provide limited experiments in a synthetic setting that validates their theoretical result is correct and the algorithm is implementable. However, their theoretical results do not seem novel or interesting, given the slew of similar results across various online learning papers, each with slight variations on the bandit / feedback type. Therefore, in my view, the experiments would need to demonstrate practical value for the paper to be accepted.

**Questions:**

None.

---

> ### Author Response · Authors · 2025-11-21
> **Response to Reviewer 7iCZ**
>
> We thank the reviewer for the constructive feedback. We understand that the primary concern lies in the practical demonstration of our algorithms. Despite our focus on the theoretical foundations of online learning, we have tried to address this point below with **new experiments** that are motivated by practical applications.  We sincerely hope the new results may help address your remaining concern. Please do not hesitate to let us know if there are any other suggestions/questions, and we would be more than happy to address them. We again truly appreciate your feedback.
>
> ## Practical Value
>
> > "...it is unclear to me how bandits with ranking feedback, as formalized here, applies to the setting of RLHF... the experiments would need to demonstrate practical value..."
>
> To mitigate your concern, we have added a **new experiment with real-world data** focused on *LLM Routing/Selection via Ranking Feedback*, which directly applies to an RLHF setting when the explicit rewards are unavailable.
>
> ### Experimental Setup:
> We model a scenario where a service provider must route user queries to the most suitable model from a diverse pool of LLMs to maximize user satisfaction. This approach is motivated by varying user preferences across tasks. For instance, choosing between creative writing and coding (*e.g.*, GPT-4o vs. GPT-5) or between latency and reasoning depth (*e.g.*, instant vs. think mode). The problem of selecting the optimal answer from multiple LLMs is well-established in prior literature [1][2][3].
>
> - Candidate Models: We sample responses from four distinct models: Qwen3-32B, Phi-4, GPT-4o, and Llama-3.1-70B.
> - Preference Oracle: To simulate human ranking feedback (the ground truth), we utilize Reward-Model-DeBERTa-v3-large-v2, **an open-source reward model trained on RLHF data**. This allows us to generate realistic utilities based on the quality of the model responses. This is common in the literature to simulate human evaluation, *e.g.*, [4].
> - Task: At each timestep, the API provider receives a query, selects models, observes a ranking produced by the PL model using utilities from the reward model, and then updates its strategy accordingly.
>
> ### New Experimental Results:
>
> The results can be found in Figure 2. Based on Figure 2, our algorithm achieves sublinear regret in this setting. This demonstrates that asymptotically, our method successfully identifies the optimal model for a given **user/query distribution using only ranking feedback, without ever accessing the latent reward scores**. This provides the missing link between our theoretical framework and practical RLHF applications, verifying that our approach is viable for optimizing large-scale model selection systems based on human-ranking preferences.
>
> ## Novelty of Theoretical Results:
>
> > "...theoretical results do not seem novel or interesting, given the slew of similar results... each with slight variations..."
>
> We respectfully disagree with this comment. To the best of our knowledge, the only and most relevant paper to our setting is a recent work [5], which studies learning with ranking feedback in a **stochastic**  online learning framework.
> In contrast, we focus on the **non-stochastic** online learning, a more challenging regime that subsumes the stochastic setting, with applications in equilibrium computation in the game-theoretic setting. Our results contained intrinsically different techniques (for the upper bounds) and counterexamples (for the lower bounds), which delineated the fundamental limits of this problem.
>
> Moreover, [5], as a theoretical paper, also only included small-scale synthetic examples for numerical evaluation. Finally, we have carefully compared with [5] and its related work on preference-based learning.
> We were not aware of any **similar results**. If the reviewer may point out any **specific technical point**  that is similar to/known in/not novel enough compared to these references, we would very much appreciate it and would be more than happy to clarify and carefully contrast.
>
> [1] Yue, Yanwei, et al. "Masrouter: Learning to route llms for multi-agent systems." arXiv preprint arXiv:2502.11133 (2025).
>
> [2] Jiang, Dongfu, Xiang Ren, and Bill Yuchen Lin. "Llm-blender: Ensembling large language models with pairwise ranking and generative fusion." arXiv preprint arXiv:2306.02561 (2023).
>
> [3] Lu, Keming, et al. "Routing to the expert: Efficient reward-guided ensemble of large language models." Proceedings of the 2024 Conference of the North American Chapter of the Association for Computational Linguistics: Human Language Technologies (Volume 1: Long Papers). 2024.
>
> [4] Dwaracherla, Vikranth, et al. "Efficient exploration for llms." arXiv preprint arXiv:2402.00396 (2024).
>
> [5] Maran, Davide, et al. "Bandits with ranking feedback." Advances in Neural Information Processing Systems 37 (2024): 80567-80608.

---

### Official Review · Reviewer_XTgL · 2025-10-31

**Soundness:** 3
**Presentation:** 3
**Contribution:** 3
**Rating:** 6
**Confidence:** 3

**Summary:**

This paper studies an online learning when ranking feedback, where the ranks of a given set of actions is returned instead of the conventional rewards. The authors show that sublinear regret cannot be achieved if the ranking function is highly deterministic, controlled by the parameter $\tau$. On the other than, if the ranking function is more random, sublinear regret is possible. This is an interesting result as in general human rankings are not consistent.  Further, a convergence results in Nash equilibrium is established.

**Strengths:**

The paper is written well and easy to follow. The motivations are clear and ranking model (PL) is standard and popular. Various problem settings are considered, e.g., bandit, full information, and Nash equilibrium. The derivations are clear and results are as expected. Solid and extensive work.

**Weaknesses:**

The ranking model is somewhat limited as having a consistent reward model in practice is very rare, e.g., RLHF in LLM. Is it possible to consider a problem where the reward function itself is sampled from a set of reward functions? This setting might be more realistic. In might be interesting to test inconsistent reward in the simulation.

**Questions:**

Does the theorem apply to the setting where the reward function itself is sampled from a set of reward functions? Is this setting somewhat related to the choice of $\tau$.

---

> ### Author Response · Authors · 2025-11-21
> **Response to Reviewer XTgL**
>
> We thank the reviewer for the positive assessment of our work. Below, we address your specific question regarding the reward functions.
>
> ## Sampled from Reward Functions
>
> > "Does the theorem apply to the setting where the reward function itself is sampled from a set of reward functions?"
>
> Yes, the results can be extended to the setting where rewards are sampled from a set of reward functions. We would like to highlight that, the goal (and whole point) of the **online learning setting** we considered is exactly to handle  **any** reward sequences, as the comparator for regret is the **best-in-hindsight** one. The arbitrary reward sequence may be drawn from ``a set of reward functions'' (as long as they are bounded between $[-1,1]$, as our setup covers).
>
> We have incorporated an online learning experiment in the revised manuscript. In this setup, rewards are sampled from Uniform, Gaussian, and Gamma distributions and subsequently truncated to the range $[-1, 1]$. The detailed experimental setup can be found in Appendix C. As illustrated in Figure 7, our algorithms demonstrate robust performance under these conditions.

---

### Official Review · Reviewer_aUGV · 2025-11-03

**Soundness:** 4
**Presentation:** 4
**Contribution:** 4
**Rating:** 8
**Confidence:** 4

**Summary:**

This paper studies no-regret algorithms under ranking feedback. In the usual no-regret setting, the algorithm sees the utility of the chosen action/ counterfactual utilities of all actions at the end of each round. However, in the ranking setting, the feedback is in the form of relative rankings of the actions, based on the Plankett Luce model, i..e. a softmax over some underlying ground truth utilities (that can be adversarially chosen each round). The paper studies two models of rankings, one based on instantaneous utilities, i.e. with myopic/ short lived agents who only live in each round and the other based on the time average or cumulative utilities over time. The main results of the paper are to characterize under what conditions no-regret learning is possible.

First, the paper observes that without noise in the ranking process (i.e. higher temperatures in the Plankett-Luce model), it is impossible for any algorithm to separate fundamentally different utility functions from each other, leading inevitably to high worst case regret -  this is intuitively true when considering purely ordinal information. Next, they observe that with sufficient, i.e. linear variation in time of the ground truth utility, it is again impossible for the learner in the instantaneous ranking setting to distinguish between different utilities.  These results are matched by constructive results showing that no-regret learning is indeed possible with sufficient (but not too high) randomness in the PL model, and with sublinear variation in the utilities over time, with the latter assumption being dropped in the full information setting.

The algorithmic results are based upon a natural algorithmic procedure that finds a natural approximate utility that best explains the observed rankings - “natural” in the sense that it inverts the PL model based on empirical observations. The best guess utilities thus obtained are fed through any no-regret algorithms from a blackbox class of “consistent” algorithms that behave roughly similarly on similar utility sequences, a property satisfied by popular no-regret algorithms such as FTRL.

**Strengths:**

The main strength of the paper is that it studies a natural and well-motivated problem that was hitherto unsolved. This problem has no shortage of consequential direct applications and is solvable without requiring fundamentally new machinery. That said, the paper is well written and has non-trivial analysis for both the upper and lower bounds results. Further, the assumptions for the positive results are well justified by the lower bounds constructions.

**Weaknesses:**

NA

**Questions:**

How does the paper connect to the offline problem of learning static ground truth utilities from ranking information? Is there a technical connection between algorithms for this problem and the algorithmic techniques in your paper?

---

> ### Author Response · Authors · 2025-11-21
> **Response to Reviewer aUGV**
>
> We sincerely thank the reviewer for their careful reading, positive assessment of our work's novelty and analysis, and for raising the insightful question regarding the connection to offline learning.
>
> ## Connection to Offline Learning
>
> > "How does the paper connect to the offline problem of learning static ground truth utilities from ranking information? Is there a technical connection between algorithms for this problem and the algorithmic techniques in your paper?"
>
> The **offline learning problem** (*e.g.,* that considered in [1]) often assumes the use of **(linear)  function approximation**, where the reward for an action $a$ at state $s$ is $r(s, a) = \mathbf{\theta}^\top \mathbf{\phi}(s, a)$. The goal is to estimate the static parameter $\mathbf{\theta}$ using **Maximum Likelihood Estimation (MLE)** based on the collected ranking feedback.
>
> - Our algorithm for estimating the instantaneous utility vectors $\mathbf{u}_t$ can be viewed as a **special case** of this MLE approach. Specifically, in our setting, the feature vector $\mathbf{\phi}(s, a)$ simplifies to a **one-hot vector** indicating the chosen action $a$. This means that the utility vector $\mathbf{u} _ t$ directly represents the utility of each action (*i.e.*, $\mathbf{u}_{t, i} = r(a_i)$).
>
> This structural simplification facilitates a fine-grained analysis of the utility estimation error, allowing us to extend offline estimation results for static ground-truth utilities to time-varying utilities in the online learning setting. In particular, we can explicitly analyze how the estimation error relates to the variation of the ground truth. Moreover, unlike offline learning, our approach obviates the need for pessimism during estimation, since we can actively control the data collection strategy to ensure sufficient coverage.
>
> [1] Zhu, Banghua, Michael Jordan, and Jiantao Jiao. "Principled reinforcement learning with human feedback from pairwise or k-wise comparisons." International Conference on Machine Learning. PMLR, 2023.

---

### Official Review · Reviewer_xsDY · 2025-11-03

**Soundness:** 3
**Presentation:** 3
**Contribution:** 2
**Rating:** 6
**Confidence:** 3

**Summary:**

This paper studies online learning and equilibrium computation when the learner only observes rankings rather than numeric utilities, in a fully adversarial setting. This connects this to finding coarse correlated equilibria in normal-form games. The authors consider two ranking models, i.e. InstUtil Rank and AvgUtil Rank, under both full-information and bandit feedback. They first prove hardness results: with InstUtil Rank, any algorithm suffers $\mathcal{O}(T)$ regret for constant temperature $\tau$. With AvgUtil Rank, when $\tau$ is very small, sublinear regret is still impossible. They then show that positive results are possible if utilities vary slowly over time, designing estimation-based algorithms that (i) convert ranking feedback into approximate numeric utilities via a Plackett–Luce estimator, and (ii) feed these estimates to a generic numeric-feedback online learner (e.g., FTRL/MWU). With InstUtil Rank and sublinear variation, they obtain sublinear regret in both full-information and bandit settings; with AvgUtil Rank and full information, they can drop the variation assumption and get sublinear regret so long as the underlying numeric-feedback algorithm is stable in cumulative utilities (Assumption 6.1). Finally, they extend the algorithms to multi-player normal-form games and show that if every player follows their no-regret ranking-based learner, the time-averaged play converges to an approximate coarse correlated equilibrium with explicit error bounds.

**Strengths:**

* The paper cleanly formulates adversarial online learning with ranking feedback and explicitly distinguishes instantaneous vs time-averaged ranking models and full-information vs bandit settings.
* The authors provide matching-style hardness results and positive results that reveals the tradeoff between hardness and possibility.
* The paper shows how ranking-based no-regret dynamics imply convergence to approximate coarse correlated equilibria in general normal-form games (Theorem 7.2 and 7.3).

**Weaknesses:**

* Many positive results require the utility sequence to have sublinear variation (e.g. Assumption 4.2), these conditions are quite strong in fully adversarial environments.
* The hardness results for AvgUtil Rank hinge on $\tau$ being extremely small, the paper does not fully clarify how sharp these thresholds are.
* Experiments are only briefly mentioned and relegated to the appendix. From the main text, it’s hard to see how the algorithms behave in practice.

**Questions:**

* Is sublinear variation $P(T)$ information-theoretically necessary for sublinear regret, or is it mainly an artifact of the current estimation + analysis?
* Can you characterize more explicitly how the regret scales with $\tau$ in the intermediate regime between “very small” (hardness) and “\mathcal{O}(1)” (positive results)?
* The results assume a known $\tau$ and exact PL model. How sensitive are the algorithms and regret bounds if $\tau$ is misspecified or the rankings are only approximately PL?

---

> ### Author Response · Authors · 2025-11-21
> **Response to Reviewer xsDY**
>
> We thank the reviewer for the thoughtful review and helpful suggestions. We are grateful that you found the formulation clean, the hardness/possibility results informative, and the application to approximate CCE compelling. We address your points and questions below.
>
> ## Necessity of Sublinear Variation $P(T)$
>
> > "Is sublinear variation $P(T)$ information-theoretically necessary for sublinear regret, or is it mainly an artifact of the current estimation + analysis?"
>
> It is necessary for the InstUtil Rank model, as demonstrated in Table 1. However, it is not required in the AvgUtil Rank model under full information, as shown in Theorem 6.2. Whether it is necessary for AvgUtil Rank with bandit feedback remains an open question.
>
> - **InstUtil Rank:** First, we highlight that for the InstUtil Rank model, sublinear $P(T)$ is information-theoretically necessary for achieving sublinear regret under fully adversarial settings. This has been demonstrated by our hardness results in Theorem 4.1: if the utilities can vary arbitrarily fast, then there exist two utility sequences with different optimal actions in hindsight that generate the same ranking in expectation, making it impossible to distinguish the optimal action.
> - **AvgUtil Rank (Full-Information):** On the other hand, we highlight that for the AvgUtil Rank model with full-information feedback, we do not require the sublinear variation assumption (Theorem 6.2).
> - **AvgUtil Rank (Bandit Feedback):** For this case, the sublinear variation $P(T)$ is indeed required. The possibility of achieving sublinear regret with constant $\tau$ with an unbounded $P(T)$ in the bandit setting is left as an important open question for immediate future work. The requirement for a small $P(T)$ in Theorem 6.3 arises because utilities must evolve slowly enough to allow for sufficient sampling of each action; this ensures that the ranking based on empirical means accurately approximates the ranking of time-average utility vectors. Conversely, in the full-information setting, this constraint is unnecessary because the ranking is derived directly from the time-average utility vectors.
>
>  ## Dependence on $\tau$
>
> > "Can you characterize more explicitly how the regret scales with $\tau$ in the intermediate regime between “very small” (hardness) and “\mathcal{O}(1)” (positive results)?"
>
> The dependence on $\tau$ is primarily governed by the error in the **Plackett-Luce (PL) estimator** we use to convert ranking feedback to numeric utility estimates. As shown in **Theorem 5.1**, the estimation error scales as $O(\tau \exp(2/\tau))$. Note that this holds for any $\tau>0$ (not necessarily large enough, as a constant).
>
> - **Intermediate Regime:** When $\tau$ is larger than $1/\log T$ but still bounded away from any fixed constant, the $\exp(2/\tau)$ term dominates the regret. Consequently, the regret is $O(\exp(2/\tau))$.
>
>  ## Approximate $\tau$ and Inaccurate PL Model
>
> > "The results assume a known $\tau$ and exact PL model. How sensitive are the algorithms and regret bounds if $\tau$ is misspecified or the rankings are only approximately PL?"
>
> ### Inaccurate $\tau$
>
> If the true temperature is $\tau^{\star} \neq \tau$, but we use $\tau$ in our algorithm, the estimate of each underlying utility difference $u(a) - u(b)$ will be scaled by a factor of $\frac{\tau}{\tau^{\star}}$. This scaling introduces an additional term in the final regret:
>
> $$\text{Regret} \approx O\left(\frac{|\tau - \tau^{\star}|}{\tau^{\star}} \cdot T\right) + \text{Original Regret Term}$$
>
>  ### Inaccurate PL Model
>
>  Similar to the case with an inaccurate $\tau$, there will be an additional term in the regret, which is the distance between the inaccurate ranking model and PL times $T$. The exact characterization of the distance depends on the actual misspecified ranking model.
>
> ### Justification for Known $\tau$
>
> Finally, we highlight that the assumption of a known $\tau$ is common in both the theoretical and empirical literature of RLHF:
>
> - Theoretical studies  (e.g., [1][2]) assumed  constant and known $\tau$.
> - Empirical studies (e.g., [3][4]) typically used a fixed $\tau$ (often with $\tau=1$).
>
> This is largely because, given any sequence of observed rankings, there often exist multiple $(\text{utility sequence}, \tau)$ pairs that can generate the same ranking sequence in expectation, making it infeasible to estimate $\tau$ simultaneously with the estimation of the reward model.
>
>  ## Experiments
>
> We understand this concern and thank you for pointing it out. In the revised manuscript, we have moved the most critical experiments into the main body of the paper (while taking into account the 10-page limit). Furthermore, following a suggestion from Reviewer 7iCZ, we have added **new experiments** demonstrating how this online learning problem relates to real-world applications, providing better practical context for our algorithms. The new results can be found at the end of the main text.

---

> > ### Author Response · Authors · 2025-11-21
> >
> > [1] Zhu B, Jordan M, Jiao J. Principled reinforcement learning with human feedback from pairwise or k-wise comparisons[C]//International Conference on Machine Learning. PMLR, 2023: 43037-43067.
> >
> > [2] Wang Y, Liu Q, Jin C. Is rlhf more difficult than standard rl? a theoretical perspective[J]. Advances in Neural Information Processing Systems, 2023, 36: 76006-76032.
> >
> > [3] Ouyang L, Wu J, Jiang X, et al. Training language models to follow instructions with human feedback[J]. Advances in Neural Information Processing Systems, 2022, 35: 27730-27744.
> >
> > [4] Christiano P F, Leike J, Brown T, et al. Deep reinforcement learning from human preferences[J]. Advances in Neural Information Processing Systems, 2017, 30.

---

### Meta-Review · Area_Chair_o7e4 · 2026-01-05

**Summary:**

The paper provides new theoretical results in the problem of online learning with ranking feedback. The reviewers mostly agree that the results are solid and potentially exciting progress in the research direction.

**Reviewer Concerns:**

There are questions about the strength and the implications of the technical assumptions that are clarified in the rebuttal; the authors can consider adding some of the clarifications in the final version. There is also a rejection based on the concern that the paper does not have adequate experiments despite being motivated by a practical problem. First of all, the paper is primarily a theory paper and the results can be accepted as-is without experiments. Second, the authors have added new empirical results in the revision.

**Reviewer Scores:**

The current scores are 8/6/6/2. I'd expect reviewer xsDY to potentially increase their score (6 -> 8) given the detailed response to the questions on the technical assumptions. The review with score 2 purely focuses on the empirical side of the work and is not considered in the final decision.

---

### Decision · Program_Chairs · 2026-01-26

Accept (Oral)